# Genomic and microenvironmental heterogeneity shaping epithelial-to-mesenchymal trajectories in cancer

Guidantonio Malagoli Tagliazucchi[1,2], Anna J. Wiecek[1,2], Eloise Withnell[1] & Maria Secrier [1]✉

The epithelial to mesenchymal transition (EMT) is a key cellular process underlying cancer progression, with multiple intermediate states whose molecular hallmarks remain poorly characterised. To fill this gap, we present a method to robustly evaluate EMT transformation in individual tumours based on transcriptomic signals. We apply this approach to explore EMT trajectories in 7180 tumours of epithelial origin and identify three macro-states with prognostic and therapeutic value, attributable to epithelial, hybrid E/M and mesenchymal phenotypes. We show that the hybrid state is relatively stable and linked with increased aneuploidy. We further employ spatial transcriptomics and single cell datasets to explore the spatial heterogeneity of EMT transformation and distinct interaction patterns with cytotoxic, NK cells and fibroblasts in the tumour microenvironment. Additionally, we provide a catalogue of genomic events underlying distinct evolutionary constraints on EMT transformation. This study sheds light on the aetiology of distinct stages along the EMT trajectory, and highlights broader genomic and environmental hallmarks shaping the mesenchymal transformation of primary tumours.

The epithelial-to-mesenchymal transition (EMT) is a cellular process in which polarised epithelial cells undergo multiple molecular and biochemical changes and lose their identity in order to acquire a mesenchymal phenotype[1]. EMT occurs during normal embryonic development, tissue regeneration, and wound healing, but also in the context of disease[1,2]. In cancer, it promotes tumour progression with metastatic expansion[3]. Recent studies have uncovered that EMT is not a binary switch but rather a continuum of phenotypes, whereby multiple hybrid EMT states underly and drive the transition from fully epithelial to fully mesenchymal transformation[3–5]. Elucidating the evolutionary trajectories that cells take to progress through these states is key to understanding metastatic spread and predicting cancer evolution.

The transcriptional changes accompanying EMT in cancer have been widely characterised and are governed by several transcription factors, including Snail, Slug, Twist and zinc fingers *ZEB1* and *ZEB2*[6–8].

EMT appears driven by waves of gene regulation underpinned by checkpoints, such as *KRAS* signalling driving the exit from an epithelial state, dependent upon *EGFR* and *MET* activation[9].

However, EMT progression is not only characterised by transcriptional alterations of regulatory circuits; the genetic background of the cell can also impact its capacity to undergo this transformation. Gain or loss of function mutations in a variety of genes, including *KRAS*[10], *STAG2*[11], *TP53*[12], as well as amplifications of chromosomes 5, 7 and 13 have been shown to promote EMT[13]. Several pan-cancer studies have also linked copy number alterations, miRNAs and immune checkpoints with EMT on a broader level[14,15]. Mathematical models have been developed to describe the switches between epithelial and mesenchymal states[4] but without considering any genomic dependencies.

Despite extensive efforts to study the dynamics of EMT, some aspects of this process remain poorly characterised. In particular, most

[1]UCL Genetics Institute, Department of Genetics, Evolution and Environment, University College London, London WC1E 6BT, UK. [2]These authors contributed equally: Guidantonio Malagoli Tagliazucchi, Anna J Wiecek. ✉e-mail: m.secrier@ucl.ac.uk

of the studies mentioned considered EMT as a binary switch and failed to capture intrinsic and local microenvironment constraints that may change along the continuum of EMT transformation. Single-cell matched DNA- and RNA-seq datasets would ideally be needed for this purpose, but they are scarce.

In this work, we integrate data from the Cancer Genome Atlas (TCGA), MET500[16], MetMap[17], GDSC[18], POG570[19], as well as orthogonal spatial transcriptomics and single-cell datasets to characterise the EMT continuum, its spatial context and interactions established with cells in the tumour microenvironment. By mapping 7,180 tumours of epithelial origin onto a timeline of epithelial-to-mesenchymal transformation, we identify discrete EMT macro-states and derive a catalogue of genomic hallmarks underlying evolutionary constraints of these states. These genomic events shed light on the aetiology of hybrid E/M and fully mesenchymal phenotypes, and could potentially act as early biomarkers of invasive cancer.

## Results

### Pan-cancer reconstruction and validation of EMT trajectories from transcriptomics data

We hypothesised that a pan-cancer survey of EMT phenotypes across bulk-sequenced samples should capture a broad spectrum of the phenotypic variation one may expect to observe at single-cell level, and this could be linked with genomic changes accompanying EMT transformation. To explore the EMT process within bulk tumour samples, we employed a cohort of primary tumours of epithelial origin ($n = 7180$) spanning 25 cancer types from TCGA. The bulk RNA-seq data from these tumours underlie multiple transcriptional programmes reflecting different biological processes, including EMT. To quantify the level of EMT transformation within a given tumour sample, we should compare its expression profile against a clean template for this programme at different steps of the EMT transition, which is best obtained in single-cell data (Fig. 1a). Inspired by McFaline-Figueroa et al.[9], we surveyed multiple single-cell RNA-sequencing (scRNA-seq) datasets where immortalised non-malignant epithelial cells or cancer cells from various tissues have been profiled at different times during the epithelial to mesenchymal transition in vitro (see Methods). These single-cell profiles can act as a reference for EMT transformation in a healthy or cancer setting, and in the following tissues: breast, lung, ovary and prostate. Hence, these data allowed us to reconstruct tissue-specific pseudo-timelines of spontaneous EMT transformation (Fig. 1b). These pseudo-timelines captured a broader or narrower range of EMT transformation, with lesser transformed (i.e., closer to 0) or more transformed (closer to 100) states depending on the tissue. Given the inter-tissue variation of the EMT programme and the limited tissue diversity in the available single-cell datasets profiled in this manner, we reasoned that a consensus reference timeline of EMT combining all these datasets would reflect a broader and more generalisable range of EMT phenotypes that we could then use as a baseline for quantifying EMT in any cancer sample (see Methods). The resulting consensus EMT reference indeed yielded a pseudo-timeline that was more specifically centred on mid-transformed and higher-transformed EMT states (Fig. 1b), which would be more often expected in random cancer samples across various stages of the disease. Furthermore, the reconstructed pseudo-timeline closely matched increasing levels of EMT transformation in independent cell line experiments from a variety of systems (Fig. 1c, Supplementary Figure 1a), validating our approach experimentally.

Next, we projected the bulk-sequenced samples from TCGA on this consensus EMT template, positioning them within the continuum of EMT states (Fig. 1a, d). To account for signals from non-tumour cells in the microenvironment, which have been recently shown by Tyler and Tirosh[20] to confound the EMT state inference in bulk data, we adjusted the expression of all genes based on the tumour purity inferred from matched DNA-sequencing (see Methods). These

corrected expression profiles were then mapped to the single-cell consensus reference trajectory and an EMT pseudo-timeline in bulk was reconstructed that accounted for potential tumour contamination. Indeed, after this correction there was no notable correlation observed between the reconstructed EMT pseudotime and the sample purity or the amount of fibroblast infiltration in the tumour (Supplementary Figure 1b). Moreover, the confounding effects highlighted by Tyler and Tirosh are prominent when using specific mesenchymal signatures that overlap with markers of cancer-associated fibroblasts (CAFs), but our approach should also be generally less prone to such biases as we employ a whole-transcriptome reference of single cells progressing through EMT rather than selected markers. Thus, the resulting signal should more reliably capture the transformation of epithelial cells rather than immune/stromal programmes, and is expected to reflect the average EMT state across the entire tumour cell population.

Using this approach, we reconstructed the EMT pseudotime trajectory across multiple cancer tissues (Fig. 1d, Supplementary Data 1). The expression of canonical epithelial and mesenchymal markers was consistent with that observed in the scRNA-seq data and expectations from the literature (Supplementary Figure 1c). Along the pseudotime, we observed frequent co-expression of such markers, which could reflect a hybrid E/M state[21] (Supplementary Figure 1d). Importantly, when analysing cancer types individually by aligning against breast, lung and prostate reference cell lines rather than to a consensus reference, the pseudotime reconstruction and EMT scores obtained were strongly correlated with those from the pan-cancer analysis (Supplementary Figure 1e), even in the case of the non-transformed breast cell line MCF10 (Supplementary Figure 1f). This suggests that the pan-cancer methodology can broadly recapitulate phenotypes identified in individual cancers, but also a classical EMT transformation outside the cancer context—thus a core, generalisable EMT programme.

### EMT macro-states underlie the continuum of mesenchymal transformation in cancer

To characterise the dominant EMT states governing the continuum of transcriptional activity described above, we discretised the pseudotime trajectory based on expression values of canonical EMT markers using a Hidden Markov Model approach and uncovered three macro-states: epithelial (EPI), hybrid EMT (hEMT) and mesenchymal (MES) (Fig. 1e, f, Supplementary Figure 1d). These states were robust to varying levels of gene-expression noise (Supplementary Figure 2a), and were not influenced by sample purity, as expected due to the correction step before pseudotime reconstruction (Supplementary Figure 2b). Furthermore, the EMT macro-state assignment employing the consensus template was similar to that obtained when using most of the tissue-specific templates, with expected small disagreements in phenotype (i.e. from EPI to hEMT, or hEMT to MES) occasionally observed while large disagreements (EPI to MES) were rare (Supplementary Figure 2c). The discrepancies were mostly observed against the MCF10 reference, which may be explained by the fact that these cells generally show a rather intermediate phenotype between epithelial and mesenchymal at baseline[22], and hence this template may not capture the entire EMT spectrum. This analysis confirmed that the uncovered macro-states are relatively stable.

As expected, the probability for the cancer cells to switch from the epithelial to the hEMT (0.32) state was higher than the probability to passage directly into the mesenchymal state (0.09) (Fig. 1e). The hEMT tumours tended to remain in the same state 42% of the times, suggesting this state could be more stable than anticipated—as previously stipulated[23] and consistent with observations that a fully mesenchymal state is not always observed[24]. Intriguingly, the EPI and hEMT states appear to convert at roughly the same rates, which could possibly reflect the dynamic effects of specific cytokines or chemokines either

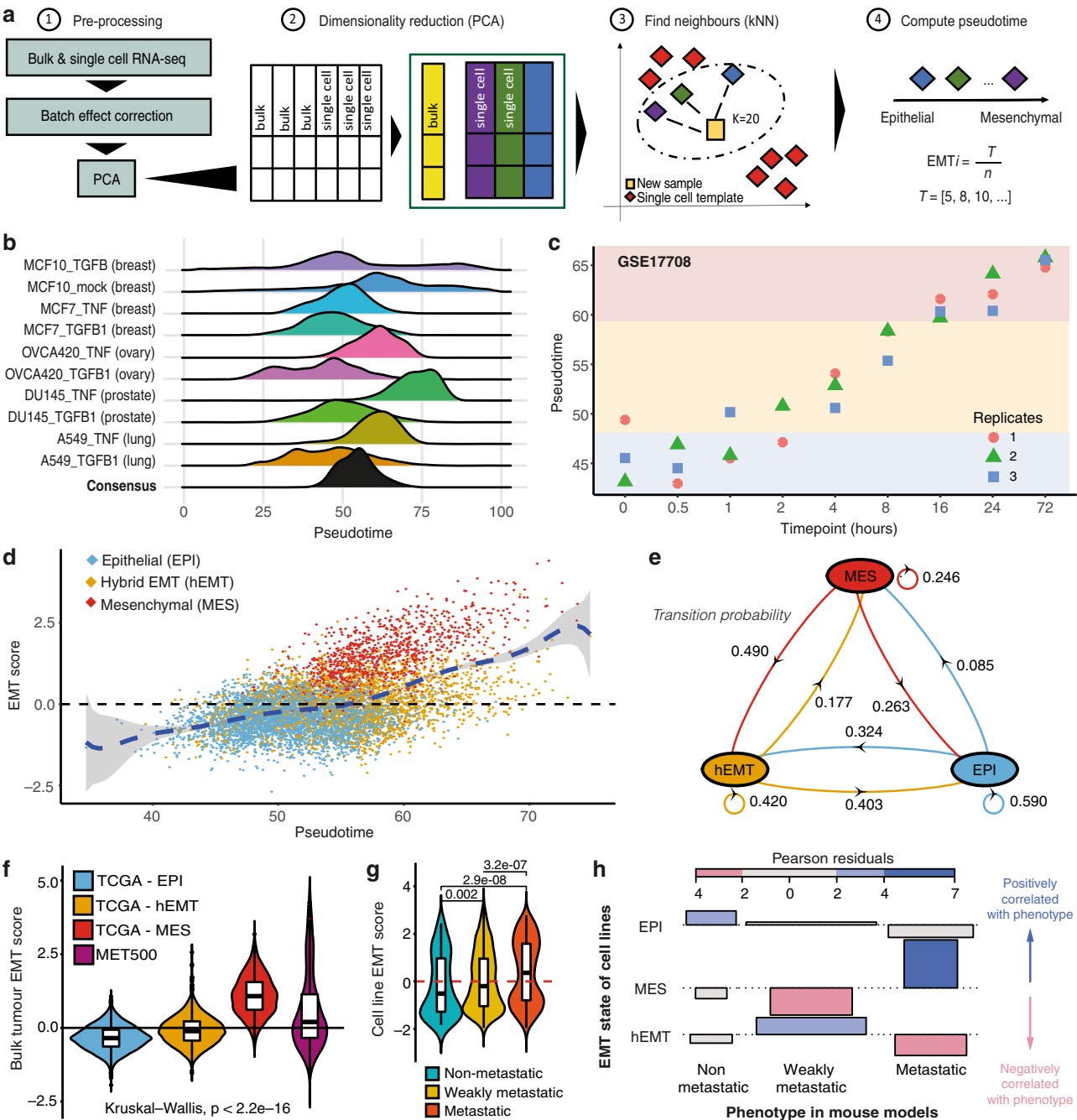

**Fig. 1 | Reconstruction and validation of EMT trajectories from transcriptomics data. a** Workflow for reconstructing the EMT trajectories from bulk/single-cell RNA-seq data. 1: Bulk and single-cell datasets are processed together to remove batch effects. 2: Dimensionality reduction using PCA is performed. 3: A k-nearest neighbours (kNN) algorithm is used to map new samples onto a reference EMT trajectory derived from scRNA-seq data. 4: Tumours are sorted by mesenchymal potential along an EMT pseudotime axis. $T$ = EMT value at the specific time point, $n$ = number of neighbours for sample $i$. **b** Distribution of EMT pseudotime values inferred using different single-cell RNA-seq templates. The consensus template combines all 10 datasets. **c** Application of the EMT trajectory reconstruction method to a time course experiment of A549 lung adenocarcinoma lines treated with TGF-beta. The pseudotime estimate increases with time as expected for gradually transforming cells. Replicates are depicted in different colours. **d** Scatter plot of EMT scores along the pseudotime across TCGA cancers. Each dot corresponds to a sample, coloured by its designated state. **e** Diagram of the transition probabilities for switching from one EMT state to another, as estimated by the HMM model. **f** EMT scores differ significantly across biologically independent samples from TCGA in the epithelial ($n$ = 3388), hEMT ($n$ = 2764), mesenchymal ($n$ = 1028) category, and the MET500 cohort ($n$ = 496) (Kruskal–Wallis test $p$ < 2.2e-16). The box centerlines depict the medians, and the edges depict the first/third quartiles. **g** EMT scores compared between cell lines from CCLE classified as non-metastatic ($n$ = 116), weakly metastatic ($n$ = 249), metastatic ($n$ = 111) according to MetMap500. The box centerlines depict the medians, and the edges depict the first/third quartiles. Two-sided Wilcoxon rank-sum test $p$ values are displayed. **h** Association plot between the HMM-derived cell line states (rows) and their experimentally measured metastatic potential (columns) (conditional independence test $p$ = 2.2e-16). Source data are provided as a Source Data file.

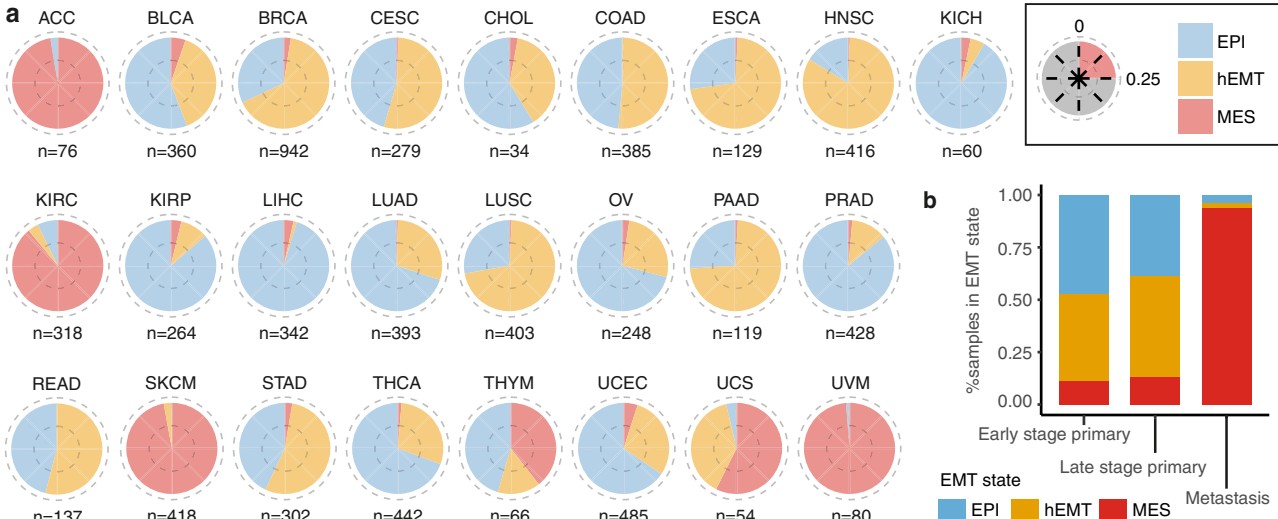

**Fig. 2 | Pan-cancer distribution of EMT macro-states. a** Distribution of the EMT states across different cancer tissues. The fraction of EPI, MES and hEMT samples are indicated in different colours. Each quarter of the pie corresponds to 25% of the data. The number of samples analysed is indicated for each tissue. Tissue abbreviations are explained in Source Data 2a. **b** EMT score distribution compared between early, late-stage primaries and metastatic samples. The colours indicate the assigned EMT state by our method. Source data are provided as a Source Data file.

starting or ceasing to be expressed in the tumour environment. For instance, Bidarra et al.[25] have shown that the removal of TGFβ1 from the medium of a 3D in vitro mesenchymal cell culture leads to a reversion to an epithelial-like state. Whether stimuli that drive EMT switches are as likely to disappear as they are to appear during the course of cancer progression is difficult to assess here and should be validated in other model systems.

The EMT scores progressively increased between the EPI, hEMT and MES states, as expected (Fig. 1f). Reassuringly, in an independent cohort of metastatic samples (MET500), EMT levels were relatively elevated along the transformation timeline compared to TCGA samples and were most abundantly falling within the hEMT state (Fig. 1f, Supplementary Figure 3a). Interestingly, we also observed possible cases of a reversion to an epithelial state in metastatic samples, which is to be expected to happen when colonising a new environmental niche via the mesenchymal to epithelial transition programme (MET), as amply described in the literature[26]. This could also explain the relatively high conversion rate (26%) from MES to EPI predicted by our HMM model (Fig. 1e).

We also applied our EMT scoring methodology to the MetMap resource, which has catalogued the metastatic potential of 500 cancer cell lines across 21 cancer types. The invasion potential of these cell lines increased with the EMT score as expected (Fig. 1g, Supplementary Figure 3b). Cell lines classified as MES by our HMM model were predominantly metastatic, while hEMT cases had a weak invasion potential (Fig. 1h), further confirming that our methodology reflects expected cellular behaviour at different EMT macro-states.

### Cancer type and stage distribution of EMT macro-states
At tissue level, the proportion of samples in each EMT state was variable (Fig. 2a), with hEMT dominating in head and neck, oesophageal, lung squamous and pancreatic carcinomas, while adenoid cystic, kidney carcinomas and melanomas were highly mesenchymal. When investigating molecular subtypes already described for a variety of cancers, most of them did not show distinct distributions by EMT state (Supplementary Figure 3c). Nevertheless, the ovarian mesenchymal subtype was reassuringly enriched in hEMT and MES cancers, and the same could be observed for genomically stable gastric cancers, which have been linked with diffuse histology and enhanced invasiveness[27].

The EMT classification was significantly correlated with the clinical cancer stage (Chi-square test statistic (df = 2) = 35.255, $p$ = 2.21e-08, Supplementary Figure 3d), with transformed samples (hEMT/MES) found to be 1.3-fold and 1.2-fold enriched in late-stage tumours (Fisher's exact test $p$ = 6.83e-06, 95% CI = [1.16,1.46]; and $p$ = 0.03, 95% CI = [1.02,1.45]), respectively, while the epithelial state was 1.4-fold over-represented (Fisher's exact test $p$ = 2.95e-09, 95% CI = [1.26,1.60]) in early-stage cancers (Fig. 2b). In primary tumours, 12% of the profiled samples were classified as fully transformed (MES), with the majority of them (60%) annotated as late-stage tumours (Supplementary Table 1). Notably, metastatic samples available from TCGA ($n$ = 343) were overwhelmingly classed as MES (94%, Fig. 2b), suggesting that the transformed phenotype is more pronounced in metastases than in primary tumours, as expected. While the correlation between cancer stage and EMT state does not appear as strong as potentially anticipated in primary tumours, the proportion of observed late-stage cancers increases as we move from EPI to hEMT and MES cancers in most cancer tissues, with mesenchymal cholangiocarcinomas, oesophageal and kidney chromophobe cancers being entirely late-stage (Supplementary Fig S3e). The fact that some early-stage cancers are classified as fully mesenchymal (5%) may suggest early evidence for the phenotypic transformation required for metastasis. Indeed, multiple studies have demonstrated the activation of the EMT transcriptional programme in the early stages of cancer[10,28]. Even the hEMT phenotype was hypothesised to be sufficient for promoting metastatic dissemination[29], although this is likely tissue-dependent.

### Tumour cell intrinsic hallmarks of EMT
While lacking the granularity of single cells, large bulk-sequenced datasets like the one from TCGA provide a great advantage through their matched genomics and transcriptomics measurements, as these allow us to glance into potential genomic determinants of cellular plasticity. We thus sought to explore the genomic background that underlies EMT transformation. Intrinsic cell properties such as increased proliferation, mutational and copy number burden, as well as aneuploidy would be expected along the EMT trajectory. Across distinct tissues, these changes were most pronounced in the hEMT state (Fig. 3a). While the clonality of tumours in the three states did not differ significantly (Kruskal–Wallis chi-squared (df = 2) = 2.3045, $p$ = 0.32), the number

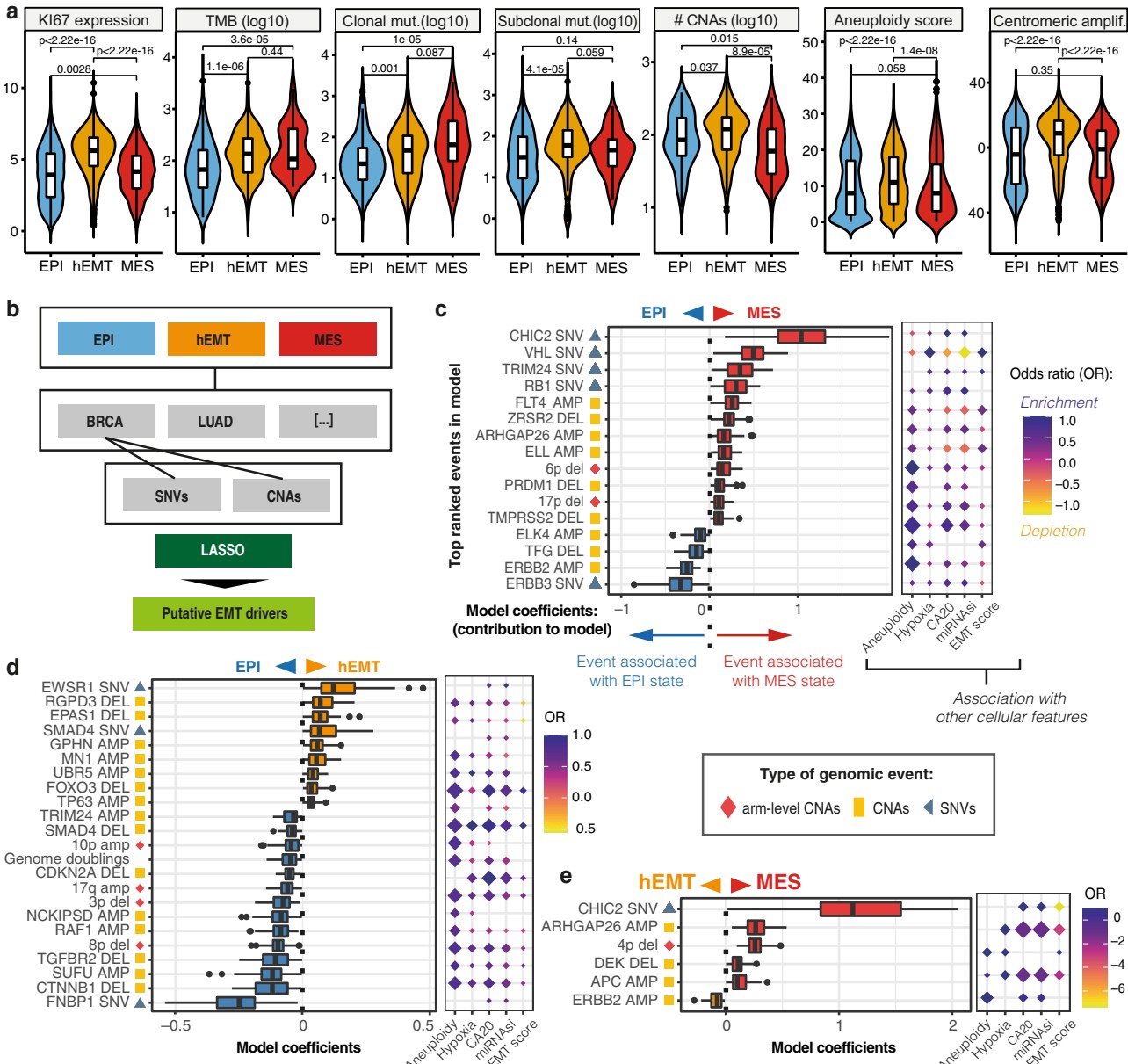

**Fig. 3 | Genomic hallmarks of EMT. a** Expression of the proliferation marker Ki67, tumour mutational burden (TMB), number of clonal/subclonal mutations (mut.), copy number aberration (CNA) burden, aneuploidy and centromeric amplification (amplif.) levels compared across biologically independent samples from TCGA in the epithelial (n = 3080), hEMT (n = 2856), mesenchymal (n = 1006) states. The centerline of boxes depicts the median values; the bottom and top box edges correspond to the first and third quartiles. Two-sided Wilcoxon signed-rank test p values are displayed. **b** The analytical workflow used to detect genomic events linked with EMT. For each state and cancer type, we used dNdScv, single nucleotide variants (SNVs) and copy number alterations (CNAs) enrichment to prioritise mutated genes and copy number events, respectively. These genomic events were then employed as input for lasso modelling to classify EMT states. **c** Top-ranked genomic markers distinguishing the mesenchymal (n = 822) from the epithelial

(n = 2710) state. The box plots depict the estimated contributions of each marker to the model across 1000 model iterations. The centerline of boxes depicts the median values; the bottom and top box edges correspond to the first and third quartiles. The balloon chart on the right illustrates the association between each marker and aneuploidy, hypoxia, centromeric amplification (CA20), stemness index (mRNAsi) and EMT score. The size of the diamonds is proportional to the significance of association, the colour gradient reports the odds ratios (OR). **d** List of the top-ranked genomic markers distinguishing the hEMT (n = 2211) from the EPI (n = 2710) state and their associated hallmarks. The annotations are as described in **c**. **e** List of the top-ranked genomic markers distinguishing the hEMT (n = 2211) from the MES (n = 822) state and their associated hallmarks. The annotations are as described in **c**. Source data are provided as a Source Data file.

of clonal and subclonal mutations increased with the state of EMT transformation. Interestingly, the hEMT group also presented higher levels of centrosome amplification, which have been linked with increased genomic instability[30,31] and poor prognosis[32].

Such alterations to the genomic integrity of the cells result from multiple mutational processes. These processes leave recognisable patterns in the genome termed mutational

signatures, which in their simplest form constitute of trinucleotide substitutions and have been broadly characterised across cancers[33]. However, their involvement in EMT transformation is poorly understood. To investigate whether any neoplastic process introducing mutations in the genomes was conditioned by EMT, we modelled the associations between mutational signatures and EMT using linear mixed-effects models (Methods, Figure S4a). The

mismatch repair deficiency signature SBS6 and the smoking-linked SBS4 signature were significantly increased in hEMT tumours, while SBS39, of unknown aetiology, was most elevated in fully transformed tumours (Figure S4b). The APOBEC mutagenesis signatures SBS2 and SBS13 also appeared elevated in hEMT tumours, in line with observations that inflammation-induced upregulation of the activation-induced cytidine deaminase (AID) enzyme, a component of the APOBEC family, triggers EMT[34]. However, when taking the tissue effect into account in the modelling procedure, no pan-cancer tissue agnostic associations between mutational processes and EMT were identified—suggesting that the previously captured associations are likely tissue-restricted. Thus, while some influence may exist on EMT from tissue-specific mutational processes, there was no evidence of an overarching mutagen that might induce EMT.

### Genomic driver events underlying the EMT transformation pan-cancer

Beyond the broader hallmarks discussed above, we sought to identify specific genomic changes creating a favourable environment for EMT transformation or imposing evolutionary constraints on its progression. We observed that subclonal diversification followed distinct routes according to the pattern of EMT transformation for several genes, including *BRAF*, *PMS1* and *FNBP1* (Figure S4c). The fraction of cancer cells harbouring *BRAF* mutations, frequently acquired in melanoma, was markedly increased in mesenchymal samples, suggesting that a clonal fixation of this event may be key for the establishment of a fully mesenchymal state, which is in line with the observed dominance of this phenotype in skin cancers (Figure S4c). Mutations in the mismatch repair gene *PMS1* and the actin cytoskeleton remodelling gene *FNBP1* were subclonally fixed in hEMT cancers, potentially suggesting that acquiring such alterations later during tumour evolution may benefit the establishment of a hybrid phenotype.

To further investigate such associations, we prioritised cancer driver mutations, focal and arm-level copy number changes that may be linked with EMT, and implemented a lasso-based machine learning framework to identify those drivers able to discriminate between EPI, hEMT and MES states across cancers, while accounting for tissue-specific effects (Methods, Fig. 3b). The developed models were validated using several other machine learning approaches and demonstrated accuracies of 92–97% in distinguishing the fully transformed state from either the hybrid or the epithelial one (Supplementary Figures S5a, b, d, e, g, h). Lower performance was obtained for the model discriminating between hEMT and EPI (~62–73%, Supplementary Figures S5c, f, i), which is not surprising due to the intermediate, hybrid nature of the former, but is still useful in understanding weaker effects on EMT transformation.

Among the genomic biomarkers able to discriminate transformed tumours (hEMT, MES) from the epithelial state, we identified genes that have been previously linked with cell migration, invasion and EMT, such as *RB1*, *VHL*, *ERBB2*, *ARHGAP26*, *PRDM1*, *APC* (Fig. 3c, d, Supplementary Data 2, Supplementary Table 2). *RB1*, a key cell cycle regulator, has been shown to promote EMT in conjunction with p53 in triple negative breast cancer[35], while *VHL* alterations contribute to EMT via regulation of hypoxia[36]. Larger scale events included deletions of the 4p, 6p and 17p chromosomal arms, all of which harboured cancer drivers which have been previously linked with EMT, e.g. *FGFR3* on 4p[37], *DAXX* and *TRIM27* on 6p[38,39], *TP53* on 17p[12] (Supplementary Data 2). Deletions of the 4p arm appeared in the majority of lung squamous cell and oesophageal carcinomas (58% and 50%, respectively), while 6p arm deletions were most frequent in pancreatic, oesophageal cancers and adrenocortical carcinomas (>20% in each). 17p arm deletions were the most abundant, especially in ovarian (76%) and kidney chromophobe cancers (76%), with an average of 37% of cases affected per tissue. Therefore, no

strong bias in terms of cancer type was observed for these large-scale alterations. In addition to these, events less strongly linked with metastatic transformation were also uncovered, such as mutations in *CHIC2*, encoding for a protein with a cysteine-rich hydrophobic domain occasionally implicated in leukaemia, or amplifications of the *ELL* gene, an elongation factor for polymerase II. While the genomic hallmarks distinguishing the extremes of mesenchymal transformation (MES versus EPI) were predominantly classical cancer drivers involved in the most fundamental processes (e.g. cell cycle) (Supplementary Figure 5j), the ones distinguishing fully transformed from hybrid phenotypes were more clearly linked with cell migration, including processes of cytoskeletal regulation, cell adhesion and T cell signalling (Supplementary Figure 5k).

The hEMT state-specific markers were mostly enriched in cell fate commitment and metabolic pathways (Fig. 3e, Supplementary Figure 5l, Supplementary Data 2). Among the top events distinguishing this phenotype from the epithelial one was the disruption of *EPAS1* (*HIF2A*), a well-known hypoxia regulator which has been previously implicated in EMT[40]. *SMAD4*, a suppressor of cell proliferation, was clearly linked with the switch between hEMT and EPI, with activating mutations contributing to an hEMT phenotype while deletions were prevalent in epithelial cancers. Indeed, *SMAD4* mutations have been shown to induce invasion and EMT marker upregulation in colorectal cancer[41]. Deletions of *FOXO3*, a gene involved in cell death and implicated in EMT[42], were specifically linked with high levels of aneuploidy, stemness and centrosome amplification.

### Validation of genomic associations

To gain further insight into the role of the putative genomic markers proposed by our pan-cancer model on EMT transformation, we validated some of these candidates and their effect on cell migration using several siRNA screens. There was a significant enrichment of our candidate genes among the siRNA hits from Koedoot et al.[43] (hypergeometric test $p = 3.33e{-}16$). Specifically, we found that knocking down 31 of the 61 targets (Supplementary Data 3) resulted in significant changes in the surface area, perimeter and elongation/roundness of the cells in Hs578T and MDA-MBA-231 breast cancer cell lines, suggesting either an impairment or an enhancement of migratory properties (Fig. 4). *ETV6*, linked to EPI-hEMT transformation in our models, was shown in Koedoot et al. to produce a big round cellular phenotype upon knockdown, with effects on cellular migration in line with expectations from the model. Indeed, *ETV6* disruption has been shown to promote TWIST1-dependent tumour progression[44], confirming our observations. Several other genes also showed significant phenotypic effects upon knockdown, albeit to a lesser extent, and many of them, including *RB1*, *ELL* and *NCKIPSD* (involved in signal transduction) were confirmed in both cell lines. *RB1* also showed a low penetrance EMT microscopy phenotype upon knockdown in an independent transcription factor-focused siRNA screen from Meyer-Schaller et al.[45], further confirming it as a mesenchymal marker.

Another gene with effects in the Hs578T cell line, *PRDM1*, a repressor of interferon activity which our model linked with the MES state, was also shown to alter multiple cellular properties associated with migration in an independent screen from Penalosa-Ruiz et al.[46] (Supplementary Figure 6a). In particular, *PRDM1* knockdown increased the E-cadherin expression area and intensity, as did *SETD2* knockdown. Among other MES-linked candidates from our models, knockdowns of the transcriptional regulators *CDC73* and *TRIM24* showed weaker phenotypes linked with migration, mostly related to homogeneity of textures observed under the microscope, again potentially related to a less transformed state. Overall, these analyses recapitulate many of the already described markers of EMT transformation, and also suggest that *ELL* and *NCKIPSD* mutations may affect the cancer cell's ability to undergo EMT transformation. Further experimental studies will be needed to clarify the mechanism by which this may occur.

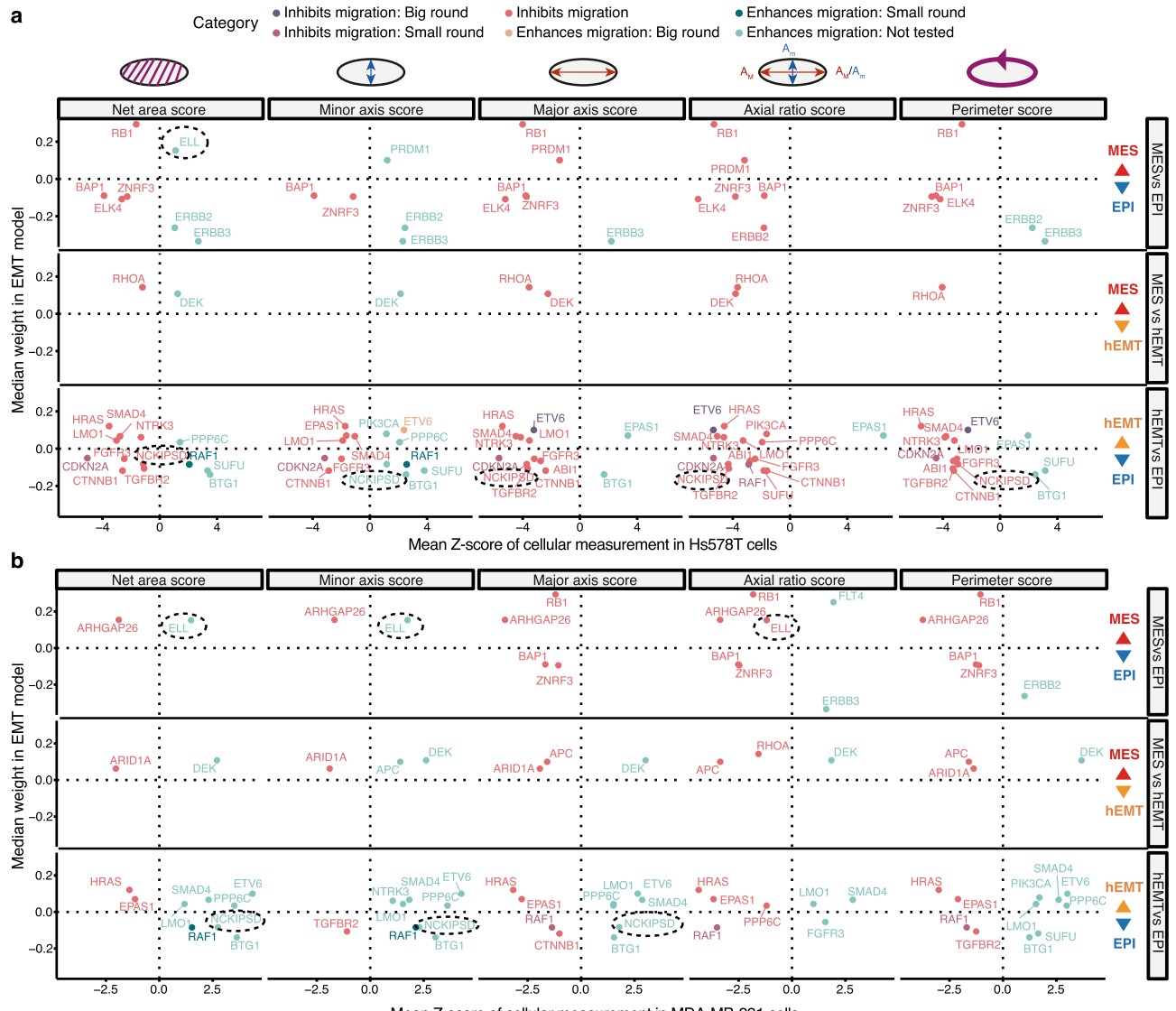

**Fig. 4 | Validation of genomic associations with EMT using siRNA screens.**
a Gene knockdown effects on cell migration abilities in Hs578T **a** and MDA-MB-231
**b** cell lines (data from Koedoot et al.[43]). The *x* axis depicts a change in the following
measurements in the cells upon the knockdown: net surface area, length of minor
($A_m$) and major ($A_M$) axes, axis ratio (large/small: elongated cells, close to 1: round
cells), perimeter score (larger—more migration). The *y* axis depicts the median
weight of the gene in the model distinguishing two different EMT states. Larger
absolute weights indicate more confident associations with EMT. The genes are
coloured according to the suggested phenotype by the respective cellular mea-
surement. A few of the genes highlighted have undergone further phenotypic tests
and this is indicated by the confirmed phenotype (big/small round). The rest of the
genes were not further tested in the study (Not tested). Only candidates with a *Z*
score value of cellular measurement >1 or <−1 are shown. The genes ELL and
NCKIPSD are highlighted with dotted ovals as they are less well characterised in the
context of EMT. Source data are provided as a Source Data file.

A good fraction of the reported alterations (36%) were also con-
firmed to be linked with the metastatic potential of cancer cell lines at
pan-cancer or tissue-specific level (Supplementary Figures S6b–d).
Among these *DEK*, a splicing regulator and putative hEMT biomarker,
showed a particularly strong correlation. Suppression of several of
these genes also strongly impacted cell viability (Supplementary
Figure 6e–f), but *RB1, DEK, RGPD3, MN1, LMO1* and *ARHGAP26* were
deemed non-essential and thus more likely to be promising targets for
EMT manipulation.

### Tumour cell extrinsic hallmarks of EMT
Beyond certain tumour cell intrinsic changes, multiple microenviron-
mental factors including tumour associated macrophages, secreted
molecules (IL-1, TNF-α) or hypoxia have been described to promote
EMT[47–49]. However, their macro-state specificity is less well char-
acterised. Investigating such associations in bulk tissue is complicated

by the previously highlighted potential bias stemming from the EMT
signal. Our tumour purity correction step should at least partially
alleviate this bias, as previously shown (Supplementary Figure 2b).
Hence, whichever associations we might still identify between the EMT
states and TME composition are likely attributable to real biological
effects rather than confounding signals. We observed that cytotoxic,
γδ T and endothelial cells were progressively enriched with increased
stages of EMT transformation (Fig. 5a, b, Supplementary Figure 7a),
suggesting that the fully mesenchymal state is most often associated
with immune hot tumours. In line with this hypothesis, these tumours
also showed the highest exhaustion levels (Fig. 5c). This links well with
our previous observations of increased clonal mutations in the MES
tumours (Fig. 3a), which could generate higher neoantigen loads
thereby triggering immune evasion[50].

The hEMT samples still displayed a relatively higher level of
fibroblasts pan-cancer, despite the tumour purity correction (Fig. 5a,

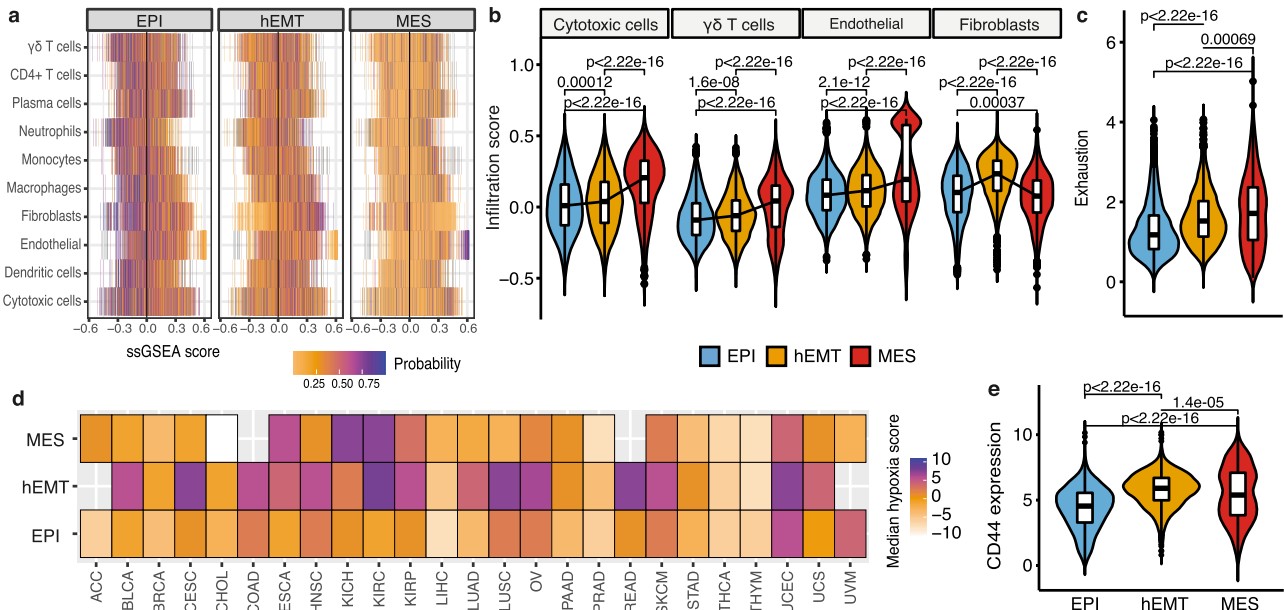

**Fig. 5 | Tumour extrinsic hallmarks of EMT. a** Heat map showcasing the results of a multinomial logistic regression model trained to predict EMT states based on cell infiltration in the microenvironment. Each row corresponds to a cell type and the corresponding per-sample infiltration is highlighted via single-sample Gene Set Enrichment Analysis (ssGSEA) scores reported on the *x* axis. The values reported in the heat map are the probabilities that a sample should fall into the epithelial, hEMT or mesenchymal categories in relation to the ssGSEA score of a certain cell type. **b** Cell abundance compared across biologically independent samples in the epithelial (*n* = 3388), hEMT (*n* = 2764) and mesenchymal (*n* = 1028) states for selected cell types. The centerline of boxes depicts the median values; the bottom and top box edges correspond to the first and third quartiles. Two-sided Wilcoxon rank-sum test *p* values are displayed. **c** Levels of exhaustion quantified across biologically independent samples in the epithelial (*n* = 3388), hEMT (*n* = 2764) and mesenchymal (*n* = 1028) states. The centerline of boxes depicts the median values; the bottom and top box edges correspond to the first and third quartiles. Two-sided Wilcoxon rank-sum test p-values are displayed. **d** Median hypoxia values in the three different EMT states across tissues are indicated by the colour gradient. **e** Gene-expression levels of the stemness marker *CD44* compared across the biologically independent samples in the epithelial (*n* = 3388), hEMT (*n* = 2764) and mesenchymal (*n* = 1028) states. The centerline of boxes depicts the median values; the bottom and top box edges correspond to the first and third quartiles. Two-sided Wilcoxon rank-sum test *p* values are displayed. Source data are provided as a Source Data file.

b). When examining these associations by cancer tissue, we noticed that active fibroblasts were often similarly enriched in hEMT and MES samples compared to epithelial ones (Supplementary Figure 7b), which would be expected with increased tumour progression. However, due to the confounding effects between fibroblast and hEMT markers as highlighted by Tyler and Tirosh[20], we acknowledge that part of the signal recovered may still not be unambiguously attributed to either cancer or microenvironmental component despite our best efforts to correct for this.

Samples with a transformed phenotype (MES, hEMT) presented significantly elevated hypoxia levels in several cancer types (Fig. 5d). Hypoxia has previously been shown to promote EMT by modulating stemness properties[51], but our analyses indicate this may be tissue and context-specific. Indeed, when examining distinct tissue sections from prostate tumours from Berglund et al.[52], only one of two tissue sections showed a marked correlation between hypoxia and hEMT within the cancer areas, and not in normal or prostatic intraepithelial neoplasia (PIN) (Supplementary Figure 8a–b). Furthermore, hypoxia was more strongly associated with high rather than moderate levels of EMT transformation in a 3D micro-tumour breast model where collective migration had been induced[53] (Supplementary Figure 8c). Thus, while hypoxia may create a favourable environment for EMT transformation in certain contexts, the analysed datasets provide no evidence for it being an obligatory condition.

We found that CD44, an established cancer stem cell marker known to promote EMT[54,55], was most highly expressed in the hEMT state across TCGA cancers (Fig. 5e), and elevated levels of several other stemness signatures most often accompanied the hEMT and MES macro-states (Figure S7c, d). Unlike mesenchymal samples, the majority of hEMT tumours (20%) were characterised by both hypoxia

and CD44 expression (Supplementary Table 3). Thus, the interplay between hypoxia and stemness may play a greater role during the intermediate rather than advanced stages of EMT transformation, although this is likely tissue-specific and requires further mechanistic investigation.

## Spatial transcriptomics reveals heterogeneous EMT patterning within the tissue

The associations identified between EMT and tumour microenvironmental features are interesting—but could potentially be confounded by averaged signals in bulk data. Indeed, bulk data is not able to capture the diversity of EMT states that may be comprised within an entire tumour, and may miss spatial effects on EMT transformation. To shed further light into these associations, we employed spatial transcriptomics data from three breast cancer slides from 10x genomics generated with the Visium platform, along with multi-region profiling of eight breast tumours generated using ST2K as described by Andersson et al.[56] to explore the spatial heterogeneity of EMT and links with other phenotypes within the cancer tissue. We observed a broad heterogeneity of EMT transformation across the tissue, with occasional clustering of EMT states within epithelial pockets (Fig. 6a, d, g). The most striking spatial pattern was that of fibroblasts, which surrounded the epithelial neoplastic areas proportionally to the increase in EMT state, appearing more strongly linked with highly transformed areas of the tumour (Fig. 6b, e, h).

We used clustering to identify areas within the tissue that present more homogeneous patterns of expression (see Methods, Supplementary Figure 9a, b) and investigated the tumour microenvironment composition within these clusters in relation to EMT states. We confirm the associations between the MES state and CD8+/CD4+ T cell

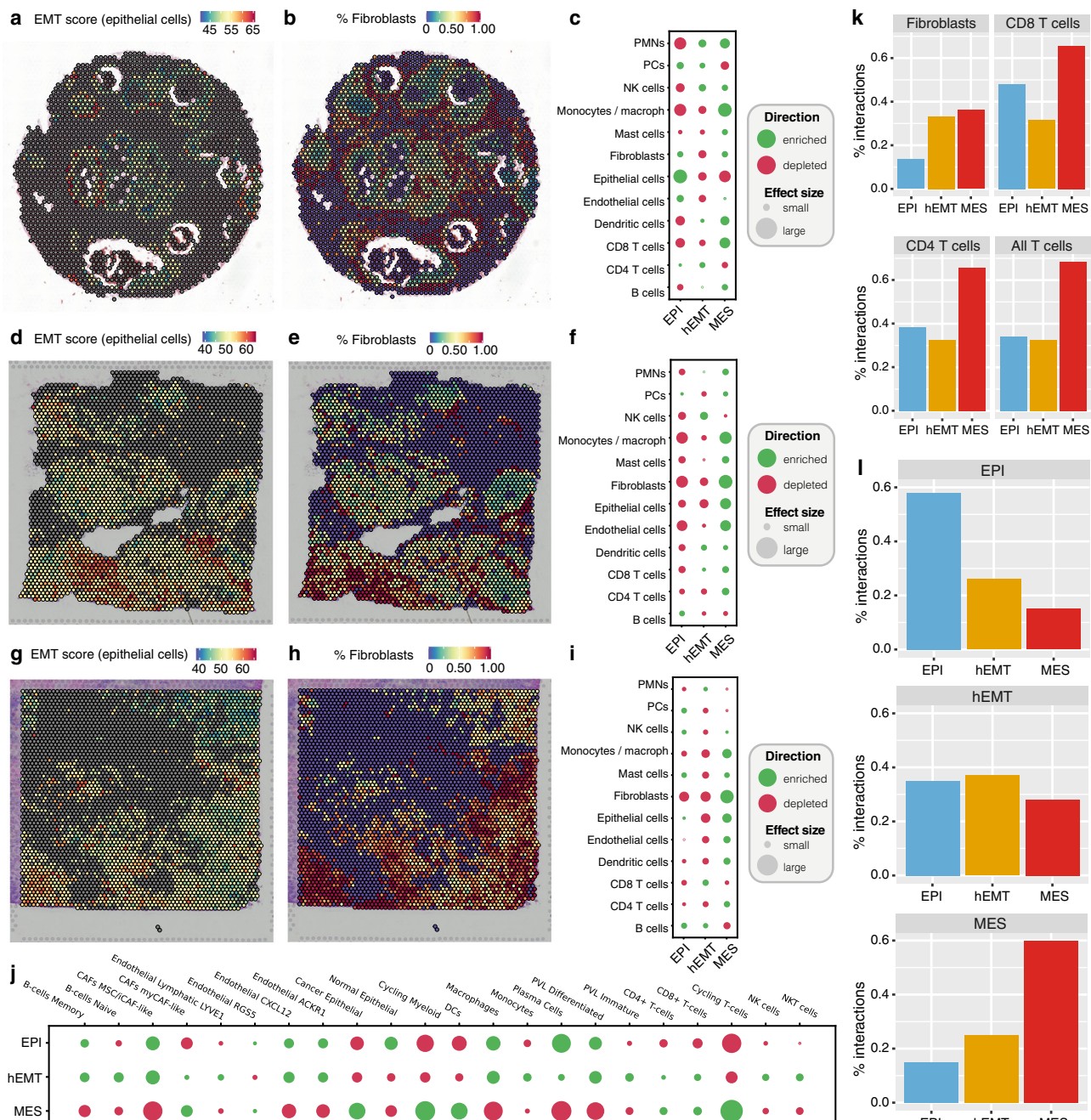

**Fig. 6 | Spatial patterns of EMT. a, b** EMT scores and the fraction of fibroblasts are visualised across within individual spots profiled across the tissue in a selected breast cancer slide, derived from spatial transcriptomics data from Patient 1 of the Visium dataset. The colour gradient reflects the expression of markers of the specific cell state (for EMT) or the fraction of cell types (for fibroblasts). **c** Enrichment and depletion of cell types in each EMT-based cluster from Patient 1. The plots represent the difference between the average cell type proportion value per region, compared to a permuted spot value (calculated 10,000 times). The plot marker size corresponds to the absolute enrichment score, and the colour represents the enrichment sign. PMN polymorphonuclear neutrophils, PC plasma cells, NK natural killer, macroph macrophages. **d–f** The same annotations as above for a breast cancer sample from Patient 2 of the Visium dataset. **g–i** The same annotations as above for a breast cancer sample from Patient 3 of the Visium dataset. **j** Enrichment and depletion of cell types in EMT-based clusters derived from multi-region spatial transcriptomics slides from the ST2K cohort. Annotation as in **c**. CAF cancer-associated fibroblasts, myCAF myofibroblastic CAF, DC dendritic cells, PVC perivascular cells, NKT natural killer T cells. **k** Fraction of interactions established between tumour cells in the three EMT macro-states and fibroblasts or T cells in the Visium dataset. **l** Fraction of interactions established among cancer cells in different EMT macro-states in the Visium dataset. Source data are provided as a Source Data file.

infiltration, monocytes and macrophages observed in bulk data (Fig. 6c, f, i). We further uncover associations between transformed (hEMT/MES) areas and dendritic cells and polymorphonuclear leucocytes (PMNs).

The hEMT state shows a relative depletion of fibroblasts compared to the MES areas in the Visium slides, which contrasts the signals seen in bulk data. However, within a larger dataset of multi-region spatial transcriptomics slides from multiple patients profiled using ST2K, we found that intermediate levels of transformation (hEMT) uniquely associated both with MSC/iCAF-like and myCAF-like cells, whereas the EPI state was only linked with the former and the MES state with the latter (Fig. 6j). Thus, the heterogeneity of hEMT-CAF

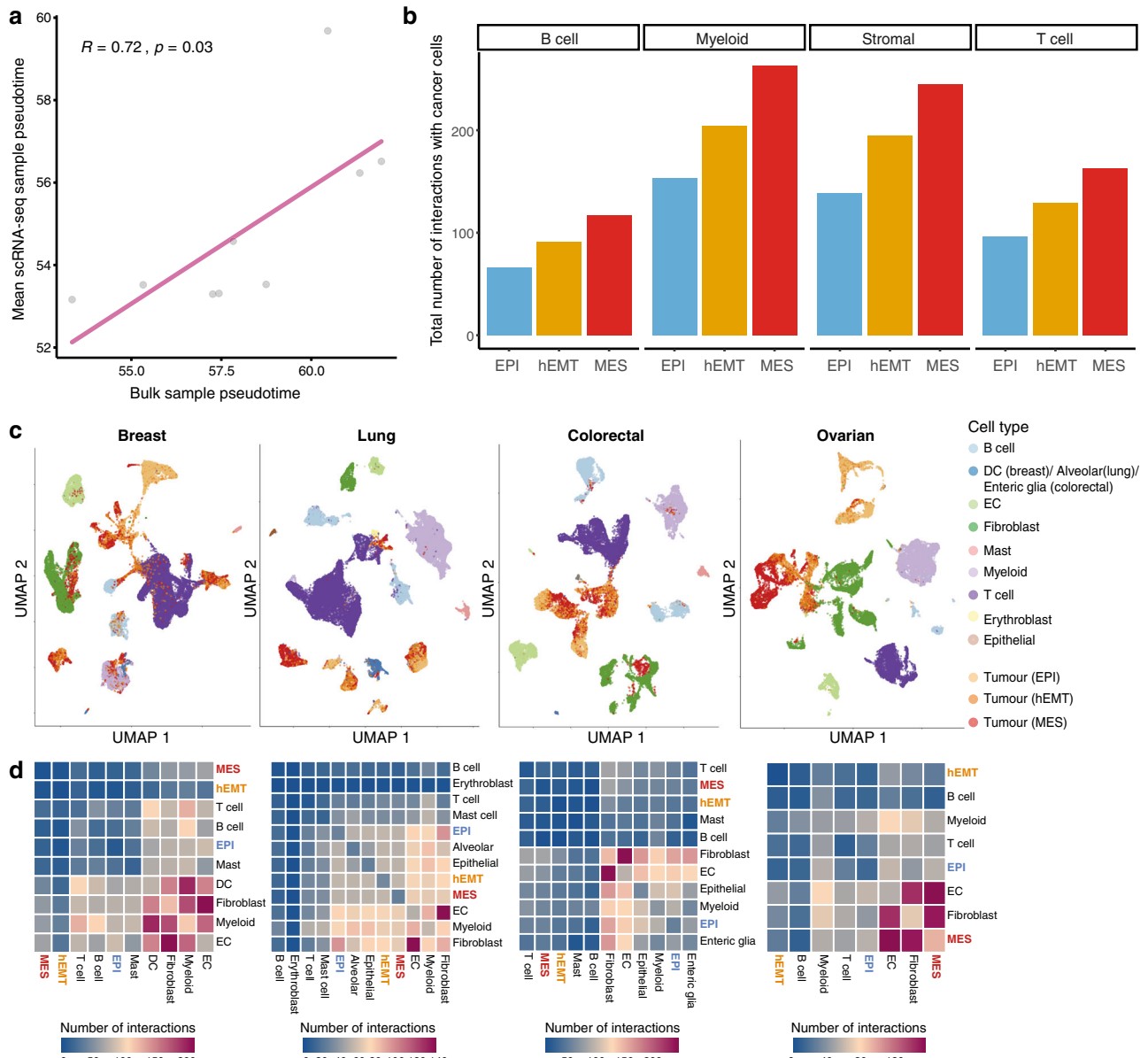

**Fig. 7 | EMT diversity in single-cell data. a** Comparison between EMT pseudotime estimates in matched bulk and single-cell samples from the same individuals (Pearson correlation coefficient R and *p*-value are shown). **b** Number of interactions established between tumour cells found in an EPI, hEMT, or MES state and other cells in the tumour microenvironment in the Chung et al.[57] dataset. **c** Uniform Manifold Approximation and Projection (UMAP) reconstruction of single-cell expression profiles depicting the tumour and microenvironment landscape of breast, lung, colorectal and ovarian tumours from Qian et al.[58]. Tumour cells are coloured according to their assigned EMT state (EPI/hEMT/MES). All other cells in the microenvironment are also depicted in different colours. DC dendritic cells, EC endothelial cells. **d** Heat maps depicting the total number of interactions established among all cell types in the same breast, lung, colorectal and ovarian datasets. The tumour cells are denoted by their EPI, hEMT and MES states. Source data are provided as a Source Data file.

associations may be explained by different subtypes of CAFs present in the context of hEMT and MES samples. Furthermore, natural killer (NK) cells were the only cell type to solely associate with hEMT spots, potentially suggesting NK activation strategies may be effective against tumour cells in this hybrid state. It is worth noting that other cell types, such as endothelial cells, showed more heterogeneous associations from sample to sample and recapitulated to a lesser extent the observations obtained in bulk tissue.

Beyond inspecting enrichment of cell populations within the spatial transcriptomics spots, we also inferred cell-cell interactions within the spatially profiled slides based on signal co-localisation. We find that cells in the fully mesenchymal state tend to interact more frequently with fibroblasts, CD8+ and CD4+ T cells compared to the ones in a hybrid or epithelial state, the latter two showing no marked

differences in the fraction of such interactions established (Fig. 7k, Supplementary Figure 9c). This further confirms the associations between transformed tumour cells and an immunogenic environment, which may be supressed via CAFs.

Finally, we also inspected how the tumour cells themselves interact with each other (Fig. 7l). Overall, we see that cancer cells at the extremes of the EMT transformation (either epithelial or fully mesenchymal) tend to interact more with cells in the same state. In contrast, there was no marked difference between the fraction of interactions established between hEMT cells and tumour cells in any state—implying that the hybrid state may be more mobile or more easily reached from any other state, as our HMM model had also suggested. The patterns were highly similar in the Visium and ST2K profiled slides (Fig. 7l, Supplementary Figure 9d).

Overall, this analysis recapitulates some of the features observed in bulk tumours, while uncovering a fine-grained heterogeneous landscape of cell states and associations. Although some of the patterns are recurrent, there is a high degree of spatial and patient-to-patient variation in EMT and TME composition, suggesting that local spatial effects are likely important determinants of EMT progression. The associations in this experiment may nevertheless be overshadowed by the fact that early stages of transformation from ductal carcinoma in situ to invasive ductal carcinoma are being investigated. Stronger patterns may be observed in more advanced cancers with hEMT or MES phenotypes, which is something we could not capture in these analyses as all tumours originated from stage I or II cancers.

Despite the large spatial variability, the continuum of EMT transformation is abundantly clear in spatially profiled slides, and stresses the importance of examining local effects to understand tumour progression and responses to treatment.

### EMT diversity in single-cell data

The analyses performed so far have been focused on datasets where the EMT signal is either measured in bulk across the entire tumour, or via spatial techniques within finer grained spots but still comprising multiple cells. This of course limits our ability to comprehend the true EMT heterogeneity of a tissue, as we lack single-cell resolution of phenotypes. To further investigate this, we employed matched bulk and single-cell data from the same cancer patients to test whether the EMT profiles estimated in bulk tissue might capture similar states as those seen at single-cell level. Using breast cancer data from Chung et al.[57], we were able to confirm a good correlation between the estimated EMT pseudo-timeline in bulk and the average EMT signal captured from single-cell data (Fig. 7a). This provides some further reassurance that the bulk estimates, while fairly generic, do approximate the average signal across the tumour. Moreover, we investigated the interactions established between cells in different EMT states and other cell populations in the tumour microenvironment (Fig. 7b). Within this dataset, the number of interactions with non-tumour cells increased with increasing EMT transformation, closely reflecting the observations in bulk tumours and spatial transcriptomics, particularly with fibroblasts and T cells.

To further explore this in multiple cancer types, we investigated single-cell data from breast, lung, colorectal and ovarian tumours as described by Qian et al.[58]. We found that tumour cells in the EPI, hEMT and MES state formed distinct clusters that often reflected an EMT progression and were well separated from clusters of other cells in the microenvironment (Fig. 4c). The majority of tumour cell clusters were clearly distanced from fibroblast clusters, confirming our premise that a whole-transcriptome reference would be better able to distinguish true malignant cells on the course of mesenchymal transformation from CAFs. Nevertheless, a minority of cells appear more similar to T cells (Fig. 4c breast and lung panels) or fibroblasts (Fig. 7c breast and colorectal panels), although they are not dispersed throughout these clusters but rather grouped at the extremity.

The cell-cell interaction landscape was quite diverse, with hEMT cells generally showing fewer interactions with the TME amongst the three states, while epithelial-fibroblast interactions were enhanced in lung and colon cancers, and mesenchymal-fibroblast interactions in ovarian cancers (Fig. 7d). This stark contrast to the observations in bulk and, to a lesser extent, also in the spatially profiled datasets could potentially imply that while immune recognition may initially happen at more advanced stages of EMT within localised areas of the tissue that become gradually transformed, the physical interactions established with these immune cells are rather volatile and the transformed cells might quickly find ways to escape immune detection, possibly with the help of CAFs for the cells in a mesenchymal state and via other mechanisms for hybrid cells. These observations, along with the spatial transcriptomics data, suggest that the relation between EMT transformation of tumour cells and their interactions with the TME is likely a complex one, highly tissue-specific and driven by local spatial effects. Ideally, single-cell, spatially resolved longitudinal datasets would be needed to fully resolve such heterogeneity.

### Clinical relevance of EMT

Finally, we show that the defined EMT states have potential clinical utility. As expected, patients with a partially or fully transformed phenotype had worse overall survival outcomes (Fig. 8a, Supplementary Table 4). Furthermore, the EMT macro-state progression reflected a step-wise decrease in progression-free intervals (Fig. 8b).

Among the driver events that have been linked with EMT in this study, alterations in genes *ERBB2*, *PRDM1*, *FLT4* and *TMPRSS2* associated with a mesenchymal phenotype, and ten other events associated with hEMT (including genome doubling, 3p/8p deletions, *EPAS1*, *NCKIPSD* mutations) were linked with worse prognosis (Fig. 8c, d, Supplementary Table 5). Cases with mutations in *FNBP1* and *CHIC2* displayed better prognosis.

To further explore potential links between EMT and therapy responses, we investigated whether EMT progression might confer different levels of sensitivity to individual cancer drugs using cell line data from GDSC[18]. We found 22 compounds whose IC50 values were significantly correlated with the EMT score (Fig. 8e). The strongest associations were observed with Sapitinib, an inhibitor of ErbB1/2/3[59], Osimertinib, a lung cancer *EGFR* inhibitor, and Acetalax, a drug used in the treatment of triple-negative breast cancers. These observations reiterate the reported genomic links between events in the tyrosine kinase pathway and EMT transformation.

Finally, we investigated whether EMT transformation may be linked with different treatment outcomes in the clinic. Within TCGA, patients with higher EMT levels in the pre-treatment tumour showed progressively worse outcomes upon oxaliplatin treatment (Fig. 8f), with complete responders significantly distinguished from patients with progressive disease. In fact, there was a twofold enrichment in complete responders to chemotherapy among patients with epithelial and hybrid tumours compared to mesenchymal ones (Fisher's exact test $p = 1.5e{-}05$, 95% CI [1.44, 2.75]). We also linked post-treatment EMT phenotypes with therapy responses using the POG570 dataset (Supplementary Figure 10a). The EMT levels increased significantly in samples treated with temozolomide over progressively longer time frames, suggesting this drug may induce EMT transformation in cancer (Supplementary Figure 10b). The opposite effect was observed for rituximab, with tumours becoming more epithelial over the treatment course.

Overall, these analyses suggest that the level of EMT transformation may play a role in determining responses to some chemotherapies as well as targeted therapies. However, our insights into the exact context in which EMT matters are limited by the lack of longitudinal, spatially and microenvironmentally resolved datasets.

## Discussion

Previous studies of the EMT process have suggested the existence of a phenotypic continuum characterised by multiple intermediate states[60]. We have developed a robust method to capture this EMT continuum in bulk, single-cell and spatially profiled datasets which uses a consensus template for EMT transformation in multiple tissues. This method can be flexibly employed to study EMT in a variety of scenarios, although we would recommend adjusting the template based on the tissue that is being studied if a pan-cancer analysis is not envisioned and if suitable single-cell data are available for the specific tissue. More information on how to achieve this is provided in the 'Reconstructing EMT using one or more single-cell references' section at our linked GitHub repository[61] (see Code Availability).

We have shown that distinct EMT trajectories in cancer are underpinned by three macro-states, reflecting both tumour cell

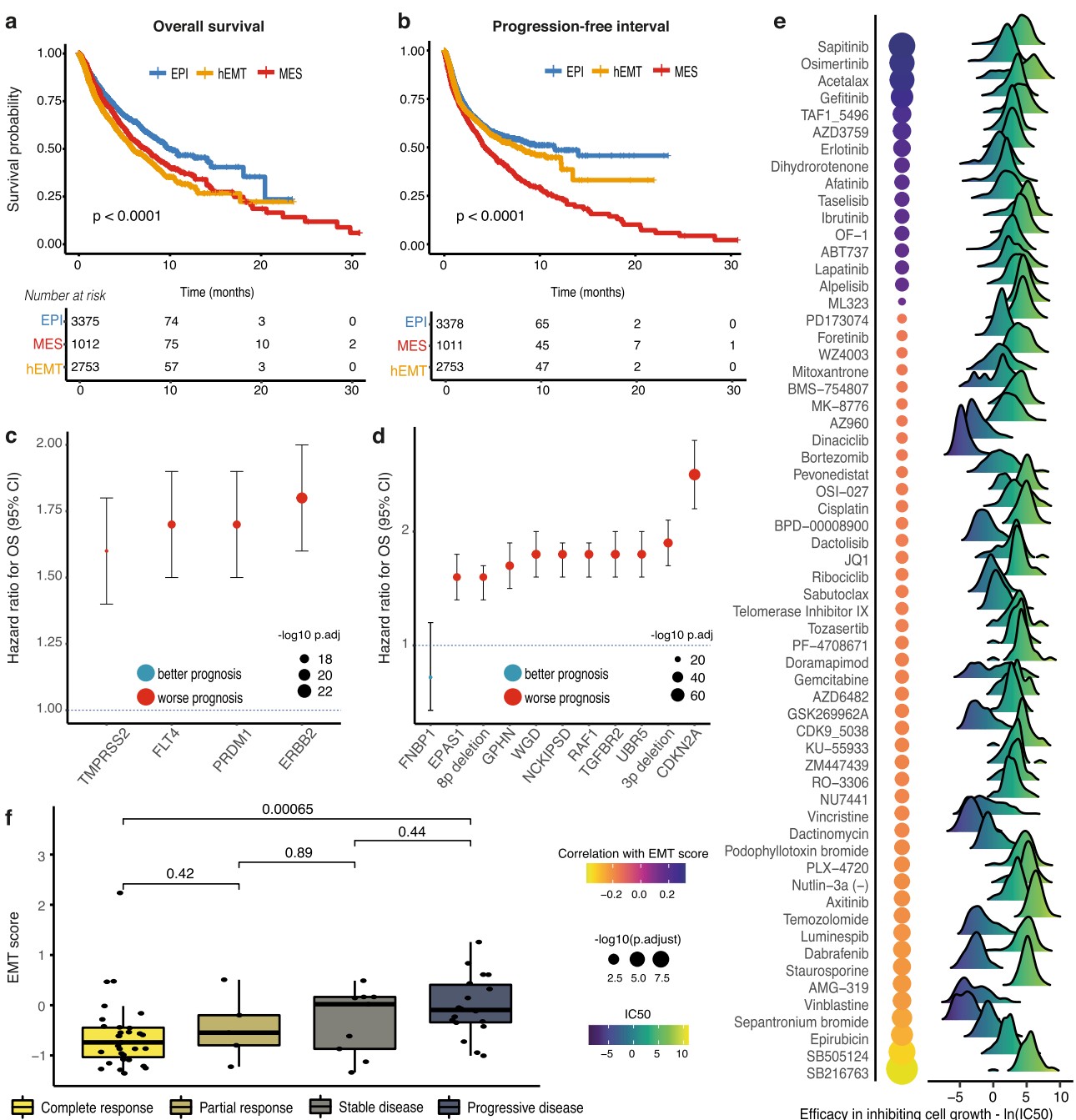

**Fig. 8 | Clinical relevance of the EMT states. a** Overall survival compared between MES, hEMT and EPI samples (Cox proportional hazards analysis). Every curve corresponds to patients whose tumours fall in a specific EMT category, depicted using different colours. **b** Progression-free interval compared between the three groups (Cox proportional hazards analysis). **c** Genomic markers distinguishing between mesenchymal and epithelial states with a significantly worse or improved outcome (q < 0.001). The mean hazard ratios for overall survival and corresponding confidence intervals from the Cox proportional hazards analysis are indicated for each marker. The colour of the dot indicates whether the marker is linked with better or worse prognosis. **d** Genomic markers distinguishing between hybrid and epithelial states with a significantly worse or improved outcome. The mean hazard ratios for overall survival and corresponding confidence intervals from the Cox proportional hazards analysis are indicated for each marker. The colour of the dot indicates

whether the marker is linked with better or worse prognosis. WGD = whole-genome doubling. **e** Correlation between the EMT scores and IC50 values in cell lines treated with various drugs. The balloon chart on the left illustrates the association between the IC50 for each compound and EMT. The size of the circles is proportional to the significance of association, and the colour corresponds to the Pearson correlation coefficient. The IC50 ranges for all cell lines are depicted by the density charts and their colour gradient. **f** EMT scores compared between responders (complete, n = 31; partial, n = 5) and non-responders (stable, n = 9; progressive, n = 18) to treatment with oxaliplatin. A gradual increase in EMT levels is observed with progressively worse outcomes. Groups are depicted using different colours and compared using two-sided Wilcoxon rank-sum tests. The centerline of boxes depicts the median, and the bottom and top box edges the first and third quartiles. Source data are provided as a Source Data file.

intrinsic as well as tumour microenvironment-associated changes. The hybrid E/M state, characterised by the co-expression of epithelial and mesenchymal markers, was surprisingly frequent (39%). It is clear that this state is distinct from epithelial tumours, presenting higher CAF infiltration and occasionally enhanced hypoxia and stemness. While it is likely this is an intermediate state in cancer progression along the EMT continuum, as suggested by the longitudinal datasets analysed and the intermediate progression-free intervals, it is also clearly heterogeneous and less genomically influenced than the extreme epithelial and mesenchymal states. Furthermore, the extent to which it is intrinsically rather than environmentally distinct cannot be determined in bulk datasets. It has been reported that cells with hybrid EMT features give rise to daughter cells that are either mesenchymal or epithelial and are more prone to migrate[54], which could explain some of the heterogeneity observed for this state. Undoubtedly, the hEMT state can be further subdivided into sub-states, as shown by Pastushenko et al.[5], Goetz et al.[4] and Brown et al.[62]. The true number of EMT intermediate states is just beginning to be explored, and the frequency and context of such switches needs to be better understood. However, the noisy bulk sequencing data are limiting our ability to capture them, highlighting the need to complement these studies with spatially resolved and single-cell data. Indeed, we show that the different EMT states are generally well distinguished from immune cells and the stroma in single-cell datasets, with a minority of cells requiring improvement in discrimination methods. Thus, our method could form a good baseline for more detailed studies of EMT heterogeneity in the single-cell space in the future.

Our study confirmed previously established molecular hallmarks of EMT, including increased mutational burden, chromosomal instability and immune exhaustion[15], along with several genomic dependencies of this process. While the exploration of EMT biomarkers is not new, most of the studies in this area have been reliant on gene-expression activity rather than mutational dependencies and they are generally tissue-specific[15,48]. Pan-cancer studies generally consider EMT as a binary switch[14,15,48]. In contrast, our study identified genomic hallmarks of three EMT macro-states, providing further granularity into how genome-driven cancer evolution shapes EMT trajectories in a state-specific manner. Indeed, we show that distinct genes contribute to the establishment of a fully mesenchymal phenotype, e.g. *RB1* or *DEK*, while others such as *EPAS1*, *FNBP1* or *SMAD4* modulate switches between epithelial and hybrid phenotypes. Furthermore, the genomic distinction in the latter case was less strong than between the extremes of EMT transformation, suggesting that transcriptional or epigenetic alterations may play an increased role in the earlier stages of EMT, while genomic events may further promote and help establish a fully transformed phenotype, which was accurately predicted based solely on genomic alterations. A causal relationship between the acquisition of any of these genomic changes and EMT should be further experimentally tested in the future.

Our spatial transcriptomics analysis demonstrates a heterogeneous EMT landscape, delineating clear spatial effects of the continuum of EMT transformation within the tissue. Thus, we gain a broader appreciation of the diversity of EMT states within a single tumour, which needs to be accounted for in future studies. However, this discovery does not detract from our analyses in bulk tumours: while considering a single sample as having a unique state was clearly a simplification, the signal captured from the tumour still reflects the average state of the cells, likely captures well the extremes of the EMT distribution when scanned across thousands of TCGA samples and is the only setting that makes genomic associations possible currently. In the future, it will be important to verify these findings in matched DNA/RNA datasets from spatially or single-cell sequenced samples.

In the spatially profiled samples that we analysed, fibroblasts and cytotoxic T cells often surrounded more mesenchymal neoplastic areas, and more frequent interactions with these cells were observed in

this context. There was evidence for initial immune recognition as suggested by the co-localisation of MES with CD8/CD4+ T cell signals and hEMT with NK cell signals, and this was further backed up by the increased mutational burden of tumours in hEMT and MES states, which could lead to higher neoantigen presentation and subsequent exhaustion of T cells[50]. Thus, these data consolidate evidence for a link between EMT progression and immune evasion which was already well documented in the literature[63,64]. Nevertheless, this analysis is limited by the small sample size and our ability to capture a broad spectrum along the EMT transformation as the data are only sourced from early-stage cancers. Larger spatial datasets and a finer-grained resolution of EMT states, especially hybrid ones, will be required in order to understand the more complex relationships established at intermediate EMT stages, which are less clear than for the fully mesenchymal states. The combined data from spatial and single-cell sequencing seem to indicate that hEMT cells are even more successful in avoiding immune detection than MES cells, as suggested by previous studies[65,66], but this hypothesis needs further investigation. This will also require new methods to identify localised, context-specific effects within the tissue which may not generalise throughout the tumour.

Our study was limited by the availability of spatial and single-cell datasets, which are only sourced from a few patients with breast, lung, ovarian and colon cancer, and hence it is not surprising that the TME associations are not always generalisable in these datasets. Furthermore, the hybrid state was treated as a single state, but we acknowledge that more than one such state can occur with different properties, which may confound the results. Intra- and intertumour heterogeneity are likely to create complex EMT-TME landscapes that are tissue and patient-specific, and resolving these is beyond the scope of our study. Despite these limitations, our analyses do serve as a proof of concept for the ability to survey EMT spatially and lay out a framework for future studies in this space. These should ideally integrate spatial and single-cell transcriptomics for a better comprehension of the complex interplay between EMT and the tumour microenvironment.

The EMT process was also linked to responses to several targeted therapies as well as some chemotherapy drugs, with an expected reduction in response in more mesenchymal cancers. EMT could thus potentially be exploited for therapeutic benefit in certain contexts.

Overall, the results of this study demonstrate the complex intrinsic and microenvironmental mechanisms that shape the landscape of EMT transformation during cancer. We have not considered the role of chromosomal rearrangements or epigenetic changes in EMT, which could provide further explanations to the maintenance of an hEMT phenotype. Additional research is required to understand the biological role and spatial constraints of the identified biomarkers, their importance in a clinical setting, and to identify additional mechanisms that may promote EMT.

## Methods
### Bulk sequencing data sources
Bulk RNA-sequencing, copy number (segment file and focal alterations), somatic variants (MuTect[67]), molecular subtypes and clinical data were retrieved for 8778 primary tumours of epithelial origin from the harmonised version of TCGA using the TCGAbiolinks R package[68]. Based on tumour purity estimates reported by Hoadley et al.[69] samples with purity lower than 30% were removed leaving 7180 samples. All other data sources employed for validation are described below.

### Reconstruction of EMT trajectories in bulk data
The reconstruction of the EMT trajectory of the TCGA samples was performed using a procedure that allows the mapping of bulk-sequenced samples to single-cell-derived expression programmes inspired from McFaline-Figueroa et al.[9]. The workflow of the analysis

consists of several steps. The first step of the analysis requires two gene-expression matrices as input, namely one bulk-sequenced data-set, for which the EMT trajectory is to be determined, and one single-cell reference dataset, for which the associated trajectory (*P*) of indi-vidual cells is known. In the first step of the analysis the matrices were merged; then, in order to remove the batch effects originated by the two different platforms, a correction was applied using the ComBat[70] function from the sva R package. In the second step, principal com-ponent analysis (PCA) was performed on the merged matrix. The single-cell-derived EMT trajectory was then mapped onto the bulk data using an iterative process and a mapping strategy based on k-nearest neighbours (k-NN). The number of iterations (i) is equal to the number of bulk samples. During each i-th step of iteration, a single bulk-sequenced sample and the reference scRNA-seq data were used as input for the k-NN algorithm. The procedure computed the mean of the pseudotime values of the single-cell samples that have been detected by the k-NN algorithm to be associated with the i-th bulk sample. The implementation of the k-NN algorithm is based on get.knnx() function from the FNN R package. In our case, we used as input the bulk RNA-seq data from TCGA samples. scRNA-seq datasets from McFaline-Figueroa et al.[9], as well as, Cook et al.[71] were used as references. Overall, 10 different scRNA-seq datasets were used including A549, MCF7, DU145 and OVCA420 cell lines treated with TGF-beta 1 or TNF. A spontaneous, as well as TGF-beta 1 driven EMT model in MCF10 cell lines was also used. The procedure described above was repeated with each of the 10 reference datasets as input along with the TCGA bulk expression data. This resulted in 10 separate pseudotime estimates for each TCGA bulk-sequenced sample, one based on each one of the reference single-cell datasets. The average of the 10 pseudotimes was used to obtain the final pseudotime estimate. Because samples are projected individually along the consensus reference single-cell data points, the pseudotime estimate only depends on the reference used and not on the specific cohort the sample is part of. Thus, the pseudotime estimates are cohort-independent.

## Segmentation of the EMT trajectory and robustness evaluation

We used a Hidden Markov Model approach to identify of a discrete number of EMT states, with the assumption that the observed expression profiles of samples that are ordered along an EMT timeline underlie multiple unobserved states. The input of this analysis was an expression matrix (M) where the rows were the TCGA samples (*N*) and the columns the gene markers (G) of EMT (see the section *Computation of the EMT scores* below for the list of genes). The original *N* columns were sorted for the *t* values of the pseudotime (*P*). This matrix and *P* were provided as input for a lasso penalised regression. *P* was used as response variable, the genes as the independent variables. The non-zero coefficients obtained from this analysis were selected to create a sub-matrix of M that was used as input for a Hidden Markov Model.

Different HMM models were tested while changing the number of states. After this tuning, and through manual inspection, we deter-mined that 3 states were most in line with biological expectations. Each HMM state was assigned to a biological group (i.e. epithelial, hybrid EMT, mesenchymal) by exploring the expression levels of known epi-thelial and mesenchymal markers in each HMM state. The selection of the coefficients was performed with the R package glmnet. The iden-tification of the EMT states was done using the depmixS4 R package.

To evaluate the robustness of the EMT states we applied the same procedure described above while increasing levels of expression noise in the original dataset. We used the *jitter* function in R to introduce a random amount of noise to the expression values of the genes (from the default parameter of the *jitter* function to noise levels of 5500). For each noise level, we repeated the analysis 100 times. We considered several metrics to measure the stability of the HMM-derived EMT

states. We reasoned that increasing noise could result in classification mismatches of the samples compared to their originally assigned EMT state. Therefore, we evaluated two metrics to assess the correct assignment of the samples to the original EMT states. Firstly, for each level of noise added and at each iteration, we computed the change in number of samples categorised in the new states compared to the original EMT states. Second, we measured the assignment accuracy for the samples to the original EMT states.

## EMT pseudotime reconstruction with adjustment for TME contamination

To account for confounding expression signals coming from non-tumour cells in the microenvironment, we regressed the expression data on the tumour purity estimates obtained from matched DNA-sequencing using the MOFA R package. The purity-adjusted expres-sion values were used as bulk input to the PCA projection for the pseudotime reconstruction.

## Computation of the EMT scores

A list of epithelial and mesenchymal markers was compiled through manual curation of the literature[6,9,48], as follows:

- epithelial genes: *CDH1, DSP, OCLN, CRB3*
- mesenchymal genes: *VIM, CDH2, FOXC2, SNAI1, SNAI2, TWIST1, FN1, ITGB6, MMP2, MMP3, MMP9, SOX10, GSC, ZEB1, ZEB2, TWIST2*

EMT scores for each TCGA sample were computed by subtracting the average RNA-seq z-scores of the epithelial marker genes from the average RNA-seq z-scores the mesenchymal marker genes, similar to Chae et al.[72]. Briefly, the average z-score transformed expression levels of the mesenchymal markers were subtracted from the average z-score transformed expression levels of the epithelial markers. To segment the EMT trajectory, along with the epithelial and mesenchymal mar-kers we have also considered markers of hybrid EMT:[6,73] *PDPN, ITGA5, ITGA6, TGFBI, LAMC2, MMP10, LAMA3, CDH13, SERPINE1, P4HA2, TNC, MMP1*.

## Tissue-specific EMT trajectory derivation

Using a similar bulk-to-single-cell mapping approach as described above, we mapped the RNA-seq data of BRCA, LUAD and PRAD tumours onto the trajectories derived from the single-cell data (including batch effect removal using ComBat, PCA on 25 dimensions and k-NN clustering). For the BRCA tumours, the final pseudotime estimates were averaged using values calculated from the MCF10 and MCF7 scRNA-seq reference datasets only. Similarly, for LUAD and PRAD bulk-sequenced samples only scRNA-seq references from A549 and DU145 cell lines were used respectively.

## Longitudinal datasets of EMT transformation

Longitudinal datasets for the validation of the EMT reconstruction method were obtained from the Gene Expression Omnibus (GEO) database as follows: GSE17708, a time course experiment of A549 lung adenocarcinoma lines treated with TGF-beta; GSE84135, a time course EMT transition experiment in hSAEC airway epithelial cells; and GSE75487, a 7 day EMT transformation experiment in H358 non-small cell lung cancer cells under doxycycline treatment to induce Zeb1. EMT pseudotime inference in these datasets was performed as described above.

## EMT trajectory reconstruction of CCLE data and inference of the metastatic potential

The RSEM gene-expression values of the Cancer Cell Line Encyclopedia[74] project were retrieved from the CCLE Data Portal. We used the same procedure described above to map the CCLE data onto the 10 reference single-cell dataset EMT trajectories. This allowed for

the pseudotime to be quantified for each CCLE sample. A segmentation using an HMM model was performed to identify a discrete number of EMT states ($n = 3$). The EMT scores were also computed for each cell line. These results were referenced against the metastatic potential scores from MetMap500[17]. The association between HMM states and experimentally measured metastatic potential groups in cell lines (non-metastatic, weakly metastatic and metastatic) was assessed using the vcd R package.

## Genomic hallmark quantification

To characterise the aneuploidy and the centromeric amplification levels of the samples in each EMT state we used the pre-computed values for TCGA from previous works[32,75]. Copy number alterations and clonality estimates based on PhyloWGS were obtained from Raynaud et al.[76]. The hypoxia levels were quantified as the number of hypoxia marker genes with expression greater than the median across TCGA samples minus the number of hypoxia marker genes with an expression less than the median, as previously proposed by Bhandari et al.[77]. Several hypoxia gene signatures from Bhandari et al.[77] were considered, yielding similar results. Only the results obtained using the genes from Buffa et al.[76] were reported. The validation of hypoxia associations with EMT was performed using the spatial transcriptomics dataset from Berglund et al.[52] and the Affymetrix profiled dataset GSE166211[53] downloaded from the GEO database using GEOquery. The EMT levels in these datasets were quantified via expression Z-scores using the GSVA package[78].

Finally, to estimate the levels of stemness in each EMT state, we considered a catalogue of stemness gene sets[79] and used them as input for gene set enrichment analysis via the GSVA R package.

## Mutational signature analysis

The identification of the mutational spectrum of the samples in each EMT state was performed using a custom approach based on SigProfilerExtractor[33] and deconstructSigs[80]. SigProfilerExtractor was used for a de-novo identification of the mutational signatures. We selected the solutions in which the minimal stability was greater than 0.4 and the sum of the minimal stabilities across signatures was greater than 1. The cosine similarity with mutational signatures catalogued in the COSMIC database was computed, and only the solutions with non-redundant signatures were selected. Next, we independently ran deconstructSigs. To ensure consistency with Alexandrov et al.[33], we evaluated the presence of the ageing-linked SBS1 and SBS5, which have been identified in all cancers. We employed the following steps to obtain a final list of signatures and their exposures for each tissue individually:

(1)  Considering the results obtained from deconstructSigs, the signatures with average contribution (across all samples) greater than 5% were taken forward in the analysis.

(2)  We combined the signatures obtained in (1) and by SigProfiler to obtain a final list of signatures for the given tissue. If SBS1 and SB5 were not present, we added these signatures manually.

To identify EMT-associated mutational processes we used a similar approach to the one described in Bhandari et al.[77], based on linear mixed-effect models. Cancer type was incorporated as a random effect in each model. An FDR adjustment was applied to the $p$ values obtained from the analysis. The full model for a specific signature (SBS) is as follows:

$$EMT\_score \sim SBS + (1|cancer)$$

## Prioritisation of genomic alterations in TCGA

Single nucleotide variants were obtained from TCGA using the TCGAbiolinks R package and the Mutect pipeline. Cancer driver events

harbouring nonsynonymous mutations were selected for further analysis. To identify putative driver events that are positively selected in association with an EMT state, we employed dNdScv[81], which quantified the ratio of nonsynonymous and synonymous mutations (dN/dS) in each gene and state, by tissue. All the somatic driver events with a $q$ value <0.10 were considered for downstream analysis.

Copy number events were obtained using the TCGAbiolinks R package. Chromosomal arm-level data were obtained from Taylor et al.[75].

## Identification of genomic events linked with EMT

To search for genomic events linked with the described EMT macro-states, we considered all somatic mutations, focal and arm-level copy number events in driver genes from the COSMIC database that were obtained in the previous steps. Two parallel methodological approaches based on lasso and random forest were used to identify events that could be predictive of EMT transitions in a two-step process. First, feature selection was performed using a stability selection approach. We used the function createDataPartition() from the caret R package to generate an ensemble of vectors representing 1000 randomly sampled training models. This is an iterative approach, in which at each iteration a lasso analysis is performed, and the non-negative coefficients computed by lasso are saved. This step was performed using the cv.glmnet() function from glmnet. The tissue source was included as potential confounder in the lasso model. The models were trained on 80% of the data. At the end of this stage, the variables that were selected in at least 80% of the iterations were taken forward and employed as predictors. Features selected in at least 50% of the iterations were also considered for downstream validation. A similar approach was employed for feature selection and model building with random forest.

In the second step, ROC curves were generated on the test dataset (20% of the data). In addition, the predictors obtained from the two pipelines were also used as input for random forest (ranger implementation), gradient boosting (gbm) and Naive Bayes models. In these cases, the trainControl() function (from the caret R package) was used in a fivefold cross-validation repeated 10 times. The function evalm() (from MLeval R package) was used to compare the different machine learning methods. Only the features selected via the lasso procedure were carried forward for downstream analysis.

## Cancer cell fraction estimates

The cancer cell fraction (CCF) of selected mutations was calculated using the following formula:

$$CCF_i = (2 + \frac{[\text{purity*}(CN_i - 2)]}{\text{purity}}) \cdot VAF_i \quad (2)$$

where $CN_i$ stands for the absolute copy number of the segment spanning mutation $i$ and $VAF_i$ is the variant allele frequency of the respective mutation. The purities of the TCGA samples were obtained from Hoadley et al.[69].

## Validation of genomic events linked with EMT

Three large-scale public siRNA screens were employed for experimental validation of the proposed genomics associations with EMT. The first dataset from Koedoot et al.[43] looked at gene knockdown effects on cell migration abilities in the Hs578T (top panel) and MDA-MB-231 breast cancer cell lines. The data were obtained from the associated publication and contained detailed measurements of effects on cellular phenotype upon knockdown, quantified as changes in cell net surface area, length of minor and major axes, axis ratio (large/small: elongated cells, close to 1: round cells), perimeter score (larger—more migration). Data from further phenotypic tests containing confirmed morphology (big/small round cells) were also

available on a subset of the genes. The hypergeometric test performed to assess enrichment of 31 out of 61 targets in the screen was performed using the dhyper function in R by calculating the probability of getting 31 or more targets among the 217 genes which showed a phenotype in the screen out of 3906 tested in total, given that we started with a total of 61 candidate genes derived from the various genomic models.

The second siRNA screen from Penalosa-Ruiz et al.[46] quantified migration-related cell integrity in mouse embryonic fibroblasts through a variety of microscopic measurements of the cells upon gene knockdown across multiple replicates. The data were obtained from the corresponding publication.

The third screen from Meyer-Schaller et al.[45] focused on the effect of transcription factor knockdown on cell migration in normal murine mammary gland epithelial cells.

To understand the relevance of the hypothesised biomarkers to the metastatic dissemination of various cancer cell lines, we downloaded the experimentally measured metastatic potential levels for cancer cell lines from MetMap[19]. We compared metastatic potential between samples with and without a specific EMT marker event (mutations or copy number alterations), pan-cancer and by tissue. Only the markers that were linked with the hEMT or MES states and that showed a statistically significant difference ($p < 0.05$) in metastatic potential between the two groups (with and without alteration) have been considered.

The viability of the cancer cell lines harbouring putative EMT biomarkers was evaluated based on CRISPR screening data[82] conducted on 990 cell lines. CERES scores denoting gene essentiality were downloaded from Project Achilles. Negative values of these scores indicate that the depletion of a gene influences negatively the viability of a cell line. We only considered genomic markers linked with the hEMT and MES states from our analysis and assessed CERES scores for individual genes both pan-cancer and at tissue level.

## Tumour microenvironment quantification

The tumour purity values of TCGA samples were retrieved from Hoadley et al.[69]. Immune deconvolution was performed using the ConsensusTME R package[83] and the ssGSEA method for cell enrichment analysis.

The results of ConsesusTME were used as input for a multinomial logistic regression model. The function multinom() (from the nnet R package) was used to determine the probability of each sample belonging to a macro-EMT state based on the cellular content of the sample.

## Spatial transcriptomics data analysis

Three breast cancer patient samples were downloaded from 10x genomics (https://support.10xgenomics.com/spatial-gene-expression/datasets). Patient 1 was AJCC Stage Group I, ER positive, PR positive and HER2 negative. Patient 2 was AJCC Stage Group IIA, ER positive, PR negative and Her2 positive. Patient 3 did not have molecular details described. The output from the Space Ranger Visium pipeline was used for analysis. The SCTransform R package was used to normalise the data based on a regularised negative binomial regression method. Cell type and state proportions for each spot were estimated using EcoTyper[84] which was run using Docker. The cell types consisted of B cells, CD4+ T cells, CD8+ T cells, dendritic cells, endothelial cells, epithelial cells, fibroblasts, mast cells, monocytes/macrophages, NK cells, plasma cells and neutrophils.

ST2K (ST second generation, 2000 spots/array) datasets (9 patients with 3–5 repeats each) were downloaded from https://github.com/almaan/her2st. All samples were stained positive for HER2. The same pre-processing steps were employed as in Andersson et al.[56]. Briefly, this consisted of using SCTransform for normalisation and Non-Negative Matrix Factorisation (NMF) for dimensionality reduction. The factors that contained consistent patterns across the tissue replicates were kept for analysis. The Stereoscope[85] (v.0.2) R package was used for cell-type deconvolution. The deconvolution data was downloaded from https://github.com/almaan/her2st. The major class consists of myeloid cells, T cells, B cells, epithelial cells, plasma cells, endothelial cells, CAFs, and perivascular-like cells (PVL cells). The minor tier contains finer partitioning of the major cell types, e.g., macrophages and CD8+ T cells. Further description of the deconvolution method is described by the authors[56].

The Seurat[86] R package was used for storing, manipulating and visualising the spatial transcriptomic data.

**Gene module scores.** An EMT score was calculated per spot by adapting the method used to assign a score to the TCGA samples, using only the breast cancer cell line scRNA-seq data. The EMT scRNA-seq trajectory was mapped onto each spot within the spatial transcriptomic slide, and the mean of the pseudotime values of the single-cell samples detected by the k-NN algorithm was used. This was performed on multiple breast cancer cell lines and the average pseudotime across the cell lines was used to calculate the EMT score. The pseudotime was split into three intervals to define an epithelial-like, hybrid-like and mesenchymal-like state. The SpatialFeaturePlot Seurat R function was used to visualise the scores. Correlations for the EMT scores were calculated by filtering for the spots containing epithelial cells and using the STUtility[87] R package to calculate the 12 nearest neighbours for each epithelial spot. The proportions of cells within each spot were summed across the neighbours.

**Cluster identification from spatial transcriptomics data.** The spatial transcriptomic data was subsetted to include solely the epithelial, hybrid and mesenchymal genes. The FindClusters Seurat R package then clustered the gene-expression data and assigned a cluster value to each barcode spot. This identified clusters by calculating the k-nearest neighbours (k-NN) and constructing a shared nearest neighbour graph. The EMT scores were averaged across the clusters. The results for each cluster were then binned so that 'low', 'medium' and 'high' groups (corresponding to EPI, hEMT, MES) were created. The cell type enrichment scores calculated per region were plotted using the enriched-region.py Python file from https://github.com/almaan/her2st.

**Inference of interaction networks.** The ScanPy[88] (Single-Cell Analysis in Python) and SquidPy[89] (Spatial Single-Cell Analysis in Python) packages were used for graph analysis on the Visium spatial slides. This included graph visualisation and graph metric algorithms. The STUtility[56] package was adapted and used to create graphs from the ST2K spatial slides. The deconvolved spot results were used to assign node labels. Edges were assigned based on the spot neighbours. NetworkX[90] was used for querying and further analysis of the networks.

## Single-cell data processing and analysis

Matched bulk and single-cell RNA-sequencing data from breast tumours described in Chung et al.[57] were retrieved from the Gene Expression Omnibus using the GSE75688 accession code. Single-cell sequencing data from breast, lung, colorectal and ovarian tumours as described by Qian et al.[58] were obtained from an interactive web server provided by the authors (https://lambrechtslab.sites.vib.be/en/pan-cancer-blueprint-tumour-microenvironment-0). Quality control analysis and normalisation of the raw gene-expression matrices provided by Qian et al.[58] was performed using the Seurat R package[91]. Matrices were filtered by removing cells with <200 and >6000 expressed genes, as well as cells with >15% of reads mapping to mitochondrial RNA. EMT pseudotime estimates were calculated for tumour cells only as described above using the scRNA-seq data references from McFaline-

Figueroa et al.[9] as well as Cook et al.[71]. For each dataset, the cells were sorted according to their pseudotime and split into 3 equally-sized groups with low, medium or high mean pseudotime estimates, corresponding to EPI, hEMT and MES states. Cell-cell interaction analysis was performed using CellPhoneDB[92] using the normalised gene-expression matrices as input, along with cell type and tumour cell pseudotime group annotation.

### Drug response datasets

Cell line drug sensitivity data were obtained from the Genomics of Drug Sensitivity in Cancer database (GDSC)[22]. The treatment information for TCGA cancers was retrieved using the TCGAbiolinks R package. The POG570[23] dataset was used to study the relation between the EMT states and the duration and effects of given cancer treatments. The EMT states in this dataset were inferred similarly as described above using the k-NN approach.

### Gene ontology analysis

The characterisation of the biological processes associated with the reported lists of genes was performed using the R package pathfindR[93].

### Survival analysis

Standardised clinical information for the TCGA cohort was obtained from Liu et al.[94]. Cox proportional hazard models were used to model survival based on variables of interest and to adjust for the following potential confounders: tumour stage, age at diagnosis, gender and body mass index (BMI). Patients in clinical stages I–II were denoted as having early-stage tumours, while stages III–IV corresponded to late-stage tumours. The R packages survival and survminer were used for data analysis and visualisation.

### Data visualisation and basic statistics

All analyses and visualisation of bulk and single-cell data were performed in R. All spatial transcriptomics analyses and visualisation were performed in R and Python v.3.7. Graphs were generated using the ggplot2, ggpubr and diagram R packages. Groups were compared using the Student's $t$ test, Wilcoxon rank-sum test or ANOVA, as appropriate.

### Reporting summary

Further information on research design is available in the Nature Portfolio Reporting Summary linked to this article.

## Data availability

The results published here are based upon publicly available data generated by the TCGA Research Network (https://www.cancer.gov/tcga), MET500 (https://met500.path.med.umich.edu/), MetMap (https://depmap.org/metmap/), GDSC (https://www.cancerrxgene.org/), POG570, CCLE (https://sites.broadinstitute.org/ccle/) and GDSC (https://www.cancerrxgene.org/). The following expression datasets from the Gene Expression Omnibus (GEO) have also been employed: GSE17708, GSE84135, GSE75487, GSE166211[53], GSE75688[58] accession code. The processed single-cell data from Qian et al.[58] were obtained from https://lambrechtslab.sites.vib.be/en/pan-cancer-blueprint-tumour-microenvironment-0. The spatial transcriptomics data employed in the study were downloaded from https://support.10xgenomics.com/spatial-gene-expression/datasets (10x Genomics Visium slides) and from https://github.com/almaan/her2st (ST2K slides). All data comply with ethical regulations, with approval and informed consent for collection and sharing already obtained by the relevant consortia. The EMT pseudotime information, macro-state labels, genomic, TME, clinical, and drug response associations in bulk data, as well as state identification and TME associations in a single-cell and spatial transcriptomics data generated in this study, are provided in the Supplementary Information and Source Data files. Source data are provided with this paper.

## Code availability

All code developed for the purpose of this analysis can be found at the following GitHub repository: https://github.com/secrierlab/EMT (Zenodo https://doi.org/10.5281/zenodo.7565397[61]).

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

## Acknowledgements

This research was supported by a Wellcome Trust Seed Award in Science (215296/Z/19/Z to G.M.T.), an MRC DTP grant (MR/N013867/1 to A.J.W.), a studentship award from the Health Data Research UK-The Alan Turing Institute Wellcome PhD Programme in Health Data Science (218529/Z/19/Z to E.W.), a UKRI Future Leaders Fellowship (MR/T042184/1 to M.S.), an Academy of Medical Science Springboard award (SBF004\1042 to M.S.), a BBSRC equipment grant (BB/R01356X/1 to M.S.) and a Wellcome Institutional Strategic Support Fund (204841/Z/16/Z to M.S.). The results published here are in part based upon data generated by the TCGA Research Network: https://www.cancer.gov/tcga.

## Author contributions

M.S. designed the study, supervised the analyses and performed the validation of EMT genomic markers using public siRNA screens. G.M.T. and A.J.W. conducted the EMT reconstruction in bulk data, and clinical and drug response correlations. G.M.T. and M.S. correlated EMT states with genomic markers and tumour intrinsic and extrinsic features. A.J.W. performed the single-cell analysis. E.W. performed the analysis of the spatial transcriptomics data. M.S. and G.M.T. wrote the manuscript. All authors read and approved the manuscript.

## Competing interests

The authors declare no competing interests.
