## [Peer review file · Nature Communications]

REVIEWER COMMENTS

Reviewer #1 (Remarks to the Author): Expert in computational cancer genomics and bioinformatics

In the current study, Tagliacruzchi and Secrier sought to identify the genetic alterations that are responsible for the epithelial to mesenchymal transition (EMT) and explore the potential clinical impact. They first established a reference of EMT score using in vitro system, which allowed them to quantitatively assess the levels of EMT in individual tumors. By analyzing TCGA samples of epithelial origin, they classified EMT status into three states: epithelial, hybrid EMT, and mesenchymal, and identified large numbers of CNVs, mutations, and patterns of mutations associated with each transformation. The findings were further validated in additional data sets including MET500, MSK-IMPACT, GENIE and MetMap, and correlated with clinical outcomes. Overall, this is a very large multi-omics study focusing on EMT as an important phenotype of tumorigenesis. The findings are scientifically interesting and clinically relevant. Some major and minor questions could be better addressed.

Major:

1. There is limited analysis of tumor stage. Since EMT is a marker for malignant transformation, it is important to determine how well the calculated EMT scores correlate with the tumor stage. The current study only provided a summary of the overall cohort. This analysis identified 18% of tumors are MES, but only 8% are from late-stage tumors. This remarkable difference warrants further investigation to explain the potential source of discrepancy, and determine if there are any systematic differences between tumor types.
2. Tumor subtypes are not taken into consideration. Many cancers have multiple subtypes with distinct clinical behaviors (e.x. hormone-driven vs HER2 vs triple-negative breast cancers). Is there any difference in the measurements, EMT score etc, between different subtypes?
3. One of the interesting findings is the association between high EMT scores (MES status) and UV mutational signature SBS7a/b, as shown in Figure S3b-c. However, the current interpretation that UV as a carcinogen may facilitate EMT is not well supported. As shown in Figure 1f, UV-related EMS tumors were mostly found in cutaneous melanoma. However, head and neck cancers, many of which are also caused by UV, do not have any MES. On the other hand, uveal melanoma, which are less related to UV, were 100% MES. Based on these patterns, the observed association between high EMT scores and UV mutations might be caused by a melanoma phenotype or a cellular property of melanocytes.

4. One major question is if the identified hEMT state represents a stable end-stage with intermediate morphological and molecular alterations levels or a transitional stage that is progressing towards MES? The named "pseudo timeline" seems to suggest the latter, but it may need to be supported with more convincing evidence. Ideally, this needs to be tested in a longitudinal setting. With the current available data sets, could the authors demonstrate more evidence of progressive patterns within individual diseases to support this important point? (again, related to tumor stage)

Minor:

1. Figure 2h. Could the higher mutation burden in hEMT be explained by more heterogeneous cellular populations or clonal compositions, especially if hEMT is considered a transitional stage?

2. Line 106, comma should be a period.

3. There are many large, integrative figures but the legends are not well documented in general. For instance, in figure 3b, does the direction of the bar indicate the type with a relatively higher fraction, or the opposite? It looks like TP53, KRAS and PIK3CA are always in the same direction in 3 pair-wise comparisons. Is it because they are co-mutated in the same cancer type with consistently higher or lower EMT scores? And it seems EPI is in-between MES and hEMT for these 3 genes. Any explanation if we consider hEMT as the intermediate state?

4. The observed association between copy numbers and EMT status, including 16q in EPI and 5q in hEMT is quite interesting. Are they enriched in specific cancer types?

5. Line 258-273, it seems there are not many overlapping EMT-driver genes between the lung and breast cancer cells, except for NF1. Does it mean the EMT-driver genes are disease-specific?

6. It seems the hEMT state had worse overall survival but better progression-free interval compared with the MES state. Any explanation?

7. There are some potential over-interpretations. For example, line302, using "confer" seems to suggest a causal relationship, but the findings only indicate an association.

8. Figure 6f, why does chemo treatment have a distinct effect on the EMT score in EPI/hEMT and MES?

Reviewer #2 (Remarks to the Author): Expert in epithelial-to-mesenchymal transition

The analysis of EMT states and trajectories is very timely and many studies related this issue have been recently published. This manuscript contains massive amount of data which prompted the authors to make many predictions which could be relevant to cancer progression. Unfortunately, the study falls very short of validation analyses, and given the varied aspects addressed and the many predictions included, it does not seem reasonable to ask for a full validation. Thus, this manuscript does not seem appropriate for Nat Comm and would be more suited, once revised and without the need to reach so many conclusions, to be a resource paper. The most interesting part is the integrative analysis of DNA mutations and EMT states, but again they lack in vivo validation.

Specific Comments:

1- Using available scRNA seq from MCF10 cells undergoing EMT, the authors build a pseudo-time template to study EMT states scored from transcriptional signatures of TCGA epithelial cancer samples. They identify macro EMT states expected to follow a phenotypic evolution from Epithelial (E) to Mesenchymal (M) over the pseudotime, but a large fraction of TCGA samples at E and hybrid EMT states are found all over the trajectory (Fig1C). The largest fraction of the M macrostate overlaps with both hybrid and E macro states. This maybe due to the fact that bulk RNA seq was noisy or may indicate that the pseudotime template modelled based on MCF10 EMT is different from that occurring in the cancer types used.

2- The application of the described EMT template should be validated in TCGA samples from individual cancer types. The pseudotime should (i) faithfully predict the in vivo EMT trajectories already described, and (ii) fit with the behaviour of cancer cells (morphology, invasiveness, motility, etc...).

3- Using a different EMT inducing protocol (cytokines treatment) in MCF7 and A549 cells, they found that TCGA samples segregated into in 5 rather than 3 states and with different outcomes, indicative of a low consistency with the EMT states predictions. An improved EMT pseudotime template based on the TCGA available data and guided by a consensus of EMT trajectory from different in vitro data sets could be generated.

4- The Hidden Markov Model predicts high stability of the E and hybrid EMT macrostates and an equal transition rate from both E and hybrid EMT to M. Does this prediction hold when cancer samples are analyzed individually?

5- Is not clear how EMT states distribution has been measured at tissue level(Figure 1F).

6- The authors present a very interesting correlation between the programme in the microenvironment and EMT status in cancer cells. The analysis is clear from the side of the immune microenvironment but misleading for the cancer associated fibroblasts (CAFs). The authors allude to the difficulty in identifying unique markers that discriminate between CAFs and cancer cells undergoing EMT. Indeed, the majority of genes used to discriminate CAFs in Figure 2E have been also described in cancer cells. A way out could be to rely on single-cell OMICs data from cancer patients to avoid mixing of cell types.

7- The higher enrichment in Hypoxia transcripts in hybrid EMT states compared to the M state is very interesting but would need to be validated functionally. Along the same lines, several cancers contain transcripts for leakage/impaired vasculature and hypoxia genes at advanced stages. mRNA in situ analysis would clarify the location of these transcripts and may help to identify the type of microenvironment that could favour the hybrid EMT over the M phenotype.

8- DNA analysis identified some mutational signatures that were more frequent and/or specific for hybrid EMT or M states. Do these mutations show specific enrichment in particular cancer types? To what extent they represent a pan-cancer specific EMT state signature? Again, this is promising but requires the characterization of EMT states. This may help to solve the contradiction found for some genes as RB1 being enriched in the M state in Figure 5E (MCF7 EMT-template) and in the hybrid EMT state in Figure 5F (A549 cells EMT-template).

9- Genomic screening and correlation with EMT states need to be validated at least for some genes to prove the functional relationship. CRISPR/Cas9 editing can be used to analyze the impact of the corresponding genes on specific EMT states.

10- In Figure 4, what is the main difference between a and b, and between c and d? In the analysis of correlation between EMT states/specific mutations and metastatic potential, many genes (i.e. ERBB4 and ERBB3) found in the pan-cancer analysis (Figure 4e) are specific of one type of cancer (Figure 4f). Others, such as CDKN2A and ARID1A, are associated with higher or lower metastatic potential depending on the type of cancer. Thus, the pan-cancer analysis does not appear to result in conserved features. Please clarify.

Reviewer #3 (Remarks to the Author): Expert in computational cancer genomics, epithelial-to-mesenchymal transition, and single-cell data analysis

The authors provide a pancancer analysis in the context of three EMT macro-states, namely: epithelial, hybrid and mesenchymal. The hybrid state, denoted hEMT, has both epithelial and mesenchymal features. While the concept of a hybrid EMT is becoming more widely accepted, the study offers a unique perspective by analyzing the hEMT state the context of driving mutations through a pancancer perspective using bulk expression datasets such as TCGA. However, the study has several major weaknesses:

First, study does not account for the stromal component of bulk expression signatures contributing to the mesenchymal component. The authors admit to this limitation, by stating “To avoid confounding effects between fibroblast and hEMT markers as highlighted by Tyler and Tirosh [ref. 34], we excluded EMT markers from the employed fibroblast gene sets, but we acknowledge that part of the signal recovered may still not be unambiguously attributed to either the cancer or microenvironmental component.” What the authors fail to discuss is the main conclusion of Tyler and Tirosh, who state that the hybrid EMT program is highly context-specific. Tyler and Tirosh reach this conclusion with a careful analysis of CAF vs malignant cell components of EMT-related genes. They add that their work “serves as a warning against using any single EMT signature for all cancer types.” Hence the authors’ work appears to directly contradict the findings of Tyler and Tirosh. These contradictions need to be more directly addressed.

Second, the authors derive their EMT macro-state signature from a single cell analysis of MCF10 breast cancer cells that have been profiled at different times during the epithelial to mesenchymal transition in vitro and apply it in a pancancer context. This is not a convincing model system for an analysis of bulk human tumors. What is the rationale that the genomic background of the MCF10 breast cancer cell line serves as a framework for a pancancer analysis? Only in the last section of their paper do they acknowledge a prior study [reference 49] that finds EMT signatures derived from cell lines to be highly context specific. The authors need to have a stronger reason for a pancancer analysis based on the MCF10-derived signature. What would happen if they derived the genomic drivers starting from another cell line?

Third, the authors do not put their work in the context of increasing amount of multi-cancer single cell RNAseq datasets in order to provide a more convincing analysis that their analysis of bulk profiles is truly specific to malignant cells derived from bulk tumors.

Revision for manuscript:

“Genomic events shaping epithelial-to-mesenchymal trajectories in cancer” (Wiecek et al)

Summary of major revisions

We are very grateful for the reviewers’ suggestions, which have helped us refine our approach to EMT analysis and supplement it with further orthogonal evidence, creating a much more nuanced landscape of EMT transformation in cancer. In particular, following the comments of Reviewers 2 and 3, we have revised the reference EMT trajectory that we employ in inferring EMT states in cancer, which was the first step in our analysis. As a result, we have had to update all the analysis of the manuscript, which means that we have redone all the figures of the manuscript. This has allowed us to also clean up and better integrate some of the analyses so that the story is easier to follow. We have made the following major modifications to the analysis and manuscript:

1. Built a consensus dataset from multiple cancer and non-transformed cell lines to act as reference for EMT reconstruction
2. Performed tumour purity correction of expression data to account for potential confounding signals from the tumour microenvironment (TME)
3. Re-inferred the EMT pseudotime and states pan-cancer and in selected cancer types
4. Analysed several longitudinal EMT datasets to further validate the method
5. Added spatial transcriptomics profiling to better explore the relation between tumour cells in different EMT states and their microenvironment, as well as the relation to hypoxia
6. Added single cell transcriptomics data to get further insights into EMT-TME cell interactions
7. Validated genomic associations using public siRNA screens

Thus, in addition to revising and strengthening our story, we also now demonstrate remarkable spatial effects in EMT-tumour microenvironment organisation and further dissect such interactions in single cells.

A detailed, point-by-point description of the changes implemented in order to address the Reviewers’ comments is included in the next pages. The resubmitted manuscript contains tracked changes and Figures 3, 4 and 6, S4 and S5 are entirely new, depicting the spatial transcriptomics, single cell and siRNA analyses, as well as further validation. The former Figure 4 has now been moved to the Supplementary Material (now Supplementary Figure S7) to avoid cluttering the main manuscript with unnecessary detail given that this validation has been superseded by the RNAi-based one in the new Figure 6. The analysis of genomic associations in breast and lung cancers has been removed as it is now redundant with the pan-cancer analysis and to simplify the story. Two new authors, Anna Wiecek and Eloise Withnell, have been added to the manuscript as they have helped address multiple points of the revision. We have also slightly altered the title to reflect the new insights from spatial transcriptomics and single cell datasets: “Genomic and local microenvironment effects shaping epithelial-to-mesenchymal trajectories in cancer”.

Reviewer #1

Major:

1. There is limited analysis of tumor stage. Since EMT is a marker for malignant transformation, it is important to determine how well the calculated EMT scores correlate with the tumor stage. The current study only provided a summary of the overall cohort. This analysis identified 18% of tumors are MES, but only 8% are from late-stage tumors. This remarkable difference warrants further investigation to explain the potential source of discrepancy, and determine if there are any systematic differences between tumor types.

We thank the reviewer for raising this valid point, which was indeed a surprise to us and needs further investigation. We have updated the entire analysis using a consensus pan-cancer EMT reference as suggested by Reviewers 2 and 3 (see initial comments and Review #3 point 2 below), so the numbers have changed slightly, but the trends remain the same. We would like to emphasise that 60% of MES samples are late-stage, and the numbers were similar before, but the phrasing was confusing, which we have now corrected. Furthermore, we show that metastatic tumours are overwhelmingly classed as mesenchymal (Supplementary

Figure 2e). We have also extended our analysis to investigate the correlation between EMT states and cancer stage by tissue, and we see an increase in the proportion of late stage cancers with increasing EMT stage in most tissues (Supplementary Figure 2f). Furthermore, we acknowledge that the discrepancy between EMT transformation and late-stage disease was not sufficiently discussed, and we have updated the manuscript text to further elaborate on this point (lines 142-156):

“In primary tumours, 12% of the profiled samples were classified as fully transformed (MES), with the majority of them (60%) annotated as late-stage tumours (Supplementary Table S2). Notably, metastatic samples available from TCGA (n=343) were overwhelmingly classed as MES (94%, Supplementary Figure 2e), suggesting that the transformed phenotype is more pronounced in metastases than in primary tumours, as expected. While the correlation between cancer stage and EMT state does not appear as strong as potentially anticipated in primary tumours, the proportion of observed late-stage cancers increases as we move from EPI to hEMT and MES cancers in most cancer tissues, with mesenchymal cholangiocarcinomas, esophageal and kidney chromophobe cancers being entirely late stage (Supplementary Fig 2f). The fact that some early stage cancers are classified as fully mesenchymal (5%) may suggest early evidence for the phenotypic transformation required for metastasis. Indeed, multiple studies have demonstrated the activation of the EMT transcriptional programme in the early stages of cancer^{10,25}. Even the hEMT phenotype was hypothesised to be sufficient for promoting metastatic dissemination²⁶, although this is likely tissue-dependent.”

2. Tumor subtypes are not taken into consideration. Many cancers have multiple subtypes with distinct clinical behaviors (e.x. hormone-driven vs HER2 vs triple-negative breast cancers). Is there any difference in the measurements, EMT score etc, between different subtypes?

The reviewer is right in pointing out that the EMT progression may vary by cancer subtype. We have now extended our analysis to survey the EMT state distribution by cancer subtype (Supplementary Figure S2c). While we do not observe any drastic differences in most cancers, we do highlight some notable variations that are in line with what would be expected for the respective molecular subtypes (lines 134-138):

“When investigating molecular subtypes already described for a variety of cancers, most of them did not show distinct distributions by EMT state (Supplementary Figure S2c). Nevertheless, the ovarian mesenchymal subtype was reassuringly enriched in hEMT and MES cancers, and the same could be observed for genomically stable gastric cancers, which have been linked with diffuse histology and enhanced invasiveness²⁴.

3. One of the interesting findings is the association between high EMT scores (MES status) and UV mutational signature SBS7a/b, as shown in Figure S3b-c. However, the current interpretation that UV as a carcinogen may facilitate EMT is not well supported. As shown in Figure 1f, UV-related EMS tumors were mostly found in cutaneous melanoma. However, head and neck cancers, many of which are also caused by UV, do not have any MES. On the other hand, uveal melanoma, which are less related to UV, were 100% MES. Based on these patterns, the observed association between high EMT scores and UV mutations might be caused by a melanoma phenotype or a cellular property of melanocytes.

Following a recalculation of the EMT scores and states based on a consensus EMT reference,, we do not see the previously reported association between MES and UV signature anymore. Even previously, that association was only appearing when not correcting for tissue-specific effects, and this is even clearer now that we have redone the analysis. While we do see a few mutational processes more pronounced in hEMT or MES, these associations are not significant anymore when adjusting for tissue effects. We have therefore reduced the claim about links with mutational processes in the manuscript and updated the text and figures accordingly (lines 295-306, Figures 5b-c):

“The mismatch repair deficiency signature SBS6 and the smoking-linked SBS4 signature were significantly increased in hEMT tumours, while SBS39, of unknown aetiology, was most elevated in fully transformed tumours (Figure 5c). The APOBEC mutagenesis signatures SBS2 and SBS13 also appeared elevated in hEMT tumours, in line with observations that inflammation-induced upregulation of the activation-induced cytidine deaminase (AID) enzyme, a component of the APOBEC family, triggers EMT⁴². However, when taking the tissue effect into account in the modelling procedure, no pan-cancer tissue agnostic associations between mutational processes and EMT were identified – suggesting that the previously captured associations are likely tissue-restricted. Thus, while some influence may exist on EMT from tissue-specific mutational processes, there was no evidence of an overarching mutagen that might induce EMT.”

4. One major question is if the identified hEMT state represents a stable end-stage with intermediate morphological and molecular alterations levels or a transitional stage that is progressing towards MES? The named "pseudo timeline" seems to suggest the latter, but it may need to be supported with more convincing evidence. Ideally, this needs to be tested in a longitudinal setting. With the current available data sets, could the authors demonstrate more evidence of progressive patterns within individual diseases to support this important point? (again, related to tumor stage)

This is an excellent point and we thank the reviewer for these suggestions. Firstly, as already discussed in point 1 above, we have further detailed the stage breakdown by cancer type and the hEMT state generally seems to present a distribution of cancer stages that is somewhat in-between the EPI and MES states (Supplementary Figure 2e). Secondly, to shed some further clarity into this issue, we have explored our EMT pseudotimeline reconstruction in three longitudinal datasets where EMT transformation has been induced and tracked over multiple time points in various cell lines: GSE17708, a time course experiment of A549 lung adenocarcinoma lines treated with TGF-beta; GSE84135, a time course EMT transition experiment in hSAEC airway epithelial cells; and GSE75487, a 7 day EMT transformation experiment in H358 non-small cell lung cancer cells under doxycycline treatment to induce Zeb1. These are ideal time course scenarios where we would expect that our EMT estimates would reflect an increase in the pseudotimeline as the experiment progresses. We have reconstructed the EMT pseudotimeline in these datasets using the same approach as in the rest of the study, and indeed we find it is very well correlated with the timeline of experiment progression through EMT (Supplementary Figure S1d and lines 102-104):

“Furthermore, the reconstructed pseudotimeline closely matched increasing levels of EMT transformation in independent cell line experiments from a variety of systems (Supplementary Figure S1d), further validating our approach experimentally.”

While these cell line experiments fall short from recapitulating the EMT transformation in human cancers, they do suggest that our EMT methodology is generalisable to a range of systems and is able to capture expected EMT transitions. Unfortunately, the limited datasets do not allow us to assess whether hEMT is always a temporary state or can be a final phenotype as well. However, most evidence, including association with intermediate levels of progression-free intervals (Figure 7b), now most likely points to the former. We expand on this briefly in the discussion (lines 440-445):

“It is clear that this state is distinct from epithelial tumours, presenting higher CAF infiltration and occasionally enhanced hypoxia and stemness. While it is likely this is an intermediate state in cancer progression along the EMT continuum, as suggested by the longitudinal datasets analysed and the intermediate progression-free intervals, it is also clearly heterogeneous and less genomically influenced than the extreme epithelial and mesenchymal states.”

Minor:

1. Figure 2h. Could the higher mutation burden in hEMT be explained by more heterogeneous cellular populations or clonal compositions, especially if hEMT is considered a transitional stage?

Following the reviewer's suggestion, we have investigated the clonality of the different TCGA cancers analysed in relation to the EMT state. We have used the annotation provided by Raynaud et al (Plos Genetics 2018), who dissect the clonal architecture of TCGA tumours based on PhyloWGS. We do not find evidence for differences in tumour clonality based on EMT state. However, we see an increased accumulation of CNAs in hEMT tumours, and higher abundance of clonal and subclonal SNVs in transformed tumours. We have updated these results in Figure 5a and Results section lines 284-286:

“While the clonality of tumours in the three states did not differ significantly (Kruskal-Wallis $p > 0.05$), the number of clonal and subclonal mutations increased with the state of EMT transformation.”

We do not wish to draw further conclusions from this analysis due to the limitations of reconstructing tumour clonality from a single bulk sequenced sample.

2. Line 106, comma should be a period.

Thank you for the correction, we have amended this now.

3. There are many large, integrative figures but the legends are not well documented in general. For instance, in figure 3b, does the direction of the bar indicate the type with a relatively higher fraction, or the opposite? It looks like TP53, KRAS and PIK3CA are always in the same direction in 3 pair-wise comparisons. Is it because they are co-mutated in the same cancer type with consistently higher or lower EMT scores? And it seems EPI is in-between MES and hEMT for these 3 genes. Any explanation if we consider hEMT as the intermediate state?

Following the re-calculation of the EMT pseudotime and states with a consensus pan-cancer reference, we have updated the genomic association analysis and have identified new associations that are much more state-specific than previously, with only a few genes shared between states. Therefore, an extensive comparison of the results from the different models is not needed anymore, and we have removed that panel from the new figure, as it was indeed confusing. The new results are shown in Figure 5f-h and described in lines 332-364. We have not noticed any striking discrepancies in the new models. We have also removed unnecessary panels that were not adding much to the story. Furthermore, we have revisited all other figures in the manuscript and have simplified many of the analyses and results so as to give a clearer and more consistent message.

4. The observed association between copy numbers and EMT status, including 16q in EPI and 5q in hEMT is quite interesting. Are they enriched in specific cancer types?

Following the updated EMT reconstruction, the associated chromosomal arm changes have now been revised and are distinct to the previous ones. However, we have looked at their prevalence across cancer types according to the Reviewer's suggestion and comment on this in the text (lines 337-346):

“Larger scale events included deletions of the 4p, 6p and 17p chromosomal arms, all of which harboured cancer drivers which have been previously linked with EMT, e.g. *FGFR3* on 4p⁴⁵, *DAXX* and *TRIM27* on 6p^{46,47}, *TP53* on 17p¹² (Supplementary Table S4). Deletions of the 4p arm appeared in the majority of lung squamous cell and esophageal carcinomas (58% and 50%, respectively), while 6p arm deletions were most frequent in pancreatic, esophageal cancers and adrenocortical carcinomas (>20% in each). 17p arm deletions were the most abundant, especially in ovarian (76%) and kidney chromophobe cancers (76%), with an average of 37% of cases affected per tissue. Therefore, no strong bias in terms of cancer type was observed for these large-scale alterations.”

5. Line 258-273, it seems there are not many overlapping EMT-driver genes between the lung and breast cancer cells, except for NF1. Does it mean the EMT-driver genes are disease-specific?

In the updated analysis, our EMT trajectories inferred pan-cancer and in individual tissues (including breast and lung) were very similar (see the new Supplementary Figure 1c), and therefore we have not recalculated the associations in a tissue-specific manner as the analysis would have been redundant and less powered. Previously, we were using a normal epithelial cell line undergoing spontaneous EMT as pan-cancer reference, and cancer cell lines where EMT was induced via different stimuli, e.g. TGFbeta, for the tissue-specific analysis – so some differences would not have been surprising. Tissue-specificity of EMT drivers is not unexpected, given that general drivers of cancer have quite a lot of tissue specificity too (and we are only examining driver gene effects in this manuscript). In the new analysis we do not see strong evidence for tissue specificity though, as highlighted above for chromosomal arm data. If we performed the analysis tissue by tissue, we would also often be underpowered to find an association with EMT states for less common drivers – but these can better be captured through the pan-cancer analysis.

6. It seems the hEMT state had worse overall survival but better progression-free interval compared with the MES state. Any explanation?

This has now changed with the updated EMT reconstruction method, and the overall survival is not significantly different between hEMT and MES, while progression-free intervals are worse for MES than hEMT cases. This is much more in line with expectations and assumptions that hEMT is an intermediate state. The new results are presented in Figure 7a-b and lines 401-404:

“As expected, patients with a partially or fully transformed phenotype had worse overall survival outcomes (Figure 7a, Supplementary Table S6). Furthermore, the EMT macro-state progression reflected a step-wise decrease in progression-free intervals (Figure 7b). “

7. There are some potential over-interpretations. For example, line302, using "confer" seems to suggest a causal relationship, but the findings only indicate an association.

The reviewer is right to point this out – we have now corrected this in the corresponding parts of the text to suggest a link rather than a causal effect.

8. Figure 6f, why does chemo treatment have a distinct effect on the EMT score in EPI/hEMT and MES?

While this observation is indeed puzzling, it comes from analysis of unmatched patient cohorts, where pre-treatment samples are from different patients than those from samples after treatment. Upon further reflection, we have decided that this severely limits the interpretation of the findings and have removed these results from the manuscript as no confident conclusions can be drawn for this. However, we do show now that the response to the chemotherapy drug oxaliplatin is progressively worse with increased EMT levels in pre-treatment samples – which is also more in line with what one might expect as cells become gradually more transformed (Figure 7f and lines 419-423):

“Within TCGA, patients with higher EMT levels in the pre-treatment tumour showed progressively worse outcomes upon oxaliplatin treatment (Figure 7f), with complete responders significantly distinguished from patients with progressive disease. In fact, there was a two-fold enrichment in complete responders among patients with epithelial and hybrid tumours compared to mesenchymal ones (Fisher’s exact test $p=1.5e-05$).”

Reviewer #2

The analysis of EMT states and trajectories is very timely and many studies related this issue have been recently published. This manuscript contains massive amount of data which prompted the authors to make many predictions which could be relevant to cancer progression. Unfortunately, the study falls very short of validation analyses, and given the varied aspects addressed and the many predictions included, it does not seem reasonable to ask for a full validation. Thus, this manuscript does not seem appropriate for Nat Comm and would be more suited, once revised and without the need to reach so many conclusions, to be a resource paper. The most interesting part is the integrative analysis of DNA mutations and EMT states, but again they lack *in vivo* validation.

We acknowledge that our insights were limited by the lack of experimental validation. We have now revised this and added several spatial transcriptomics, single cell and siRNA validation datasets that help us further consolidate and expand the findings presented. These are described in detail in the introduction of this letter and in the points below.

Specific Comments:

1- Using available scRNA seq from MCF10 cells undergoing EMT, the authors build a pseudo-time template to study EMT states scored from transcriptional signatures of TCGA epithelial cancer samples. They identify macro EMT states expected to follow a phenotypic evolution from Epithelial (E) to Mesenchymal (M) over the pseudotime, but a large fraction of TCGA samples at E and hybrid EMT states are found all over the trajectory (Fig1C). The largest fraction of the M macrostate overlaps with both hybrid and E macro states. This maybe due to the fact that bulk RNA seq was noisy or may indicate that the pseudotime template modelled based on MCF10 EMT is different from that occurring in the cancer types used.

The Reviewer is right in questioning the validity of the MCF10 single cell dataset as a template for all cancer types. Following this comment, point 3 below and comments from Reviewer #3, we have now revised our strategy to reconstructing an EMT timeline pan-cancer. Specifically, we have built a consensus EMT reference from both the non-transformed epithelial cell line MCF10 as well as several cancer cell lines where EMT has been induced via different stimuli, including breast, lung, prostate and ovarian cancer (Methods lines 522-529 and Results lines 72-79):

“Inspired by McFaline-Figueroa et al⁹, we quantified the levels of EMT in these bulk tumours against a consensus reference single cell RNA sequencing (scRNA-seq) dataset derived from non-transformed epithelial cells as well as cancer cells from multiple tissues that have been profiled at different times during the epithelial to mesenchymal transition *in vitro*. These data allowed us to reconstruct a generic “pseudo-timeline” of spontaneous EMT transformation onto which we projected the bulk sequenced samples from TCGA, positioning them within the continuum of EMT states (Figure 1a).”

This has allowed us to have a reference that is much less biased in terms of tissue of origin, and indeed the associations between the newly identified hEMT and MES states with clinical outcomes (Figures 7a-b), as well as validations of metastatic potential in cancer cell lines (Figure 1e) have improved. Nonetheless, the noise and convolution of the bulk sequencing will always be a limiting factor to our inference of EMT states and we openly acknowledge this in lines 90-91:

“Thus, the resulting signal ... is expected to reflect the average EMT state across the entire tumour cell population. “

This is also why we define only 3 states (rather than multiple intermediate states). The bulk data are indeed noisy and this is a limitation which we now further comment on in the discussion (lines 450-452):

“The true number of EMT intermediate states is just beginning to be explored. However, the noisy bulk sequencing data are limiting our ability to capture them, highlighting the need to complement these studies with spatially-resolved and single cell data.”

Finally, in this revised version we have complemented these analyses with new insights from spatial transcriptomics and single cell datasets, to obtain a much more refined picture of EMT transformation and its spatial and microenvironmental effects. These new results complement the insights from bulk tumours and are presented in Figures 3 and 4 of the manuscript (described in detail in Results sections “Spatially-resolved EMT patterning reveals local microenvironmental effects” and “EMT diversity in single cell data”).

2- The application of the described EMT template should be validated in TCGA samples from individual cancer types. The pseudotime should (i) faithfully predict the *in vivo* EMT trajectories already described, and (ii) fit with the behaviour of cancer cells (morphology, invasiveness, motility, etc...).

Following the Reviewer’s suggestion, we have compared the EMT reconstruction performance when using a pan-cancer (consensus) reference versus tissue-specific references and we see very good correlations between pseudotime and EMT score estimates (Supplementary Figure S1c and lines 97-101):

“Importantly, when analysing cancer types individually by aligning against breast, lung and prostate reference cell lines rather than to a consensus reference, the pseudotime reconstruction and EMT scores obtained were strongly correlated with those from the pan-cancer analysis (Supplementary Figure S1c), thereby demonstrating that the pan-cancer methodology can broadly recapitulate phenotypes identified in individual cancers.”

Secondly, to shed some further clarity into this issue, we have explored our EMT pseudotime reconstruction in three longitudinal datasets where EMT transformation has been induced and tracked over multiple time points in various cell lines: GSE17708, a time course experiment of A549 lung adenocarcinoma lines treated with TGF-beta; GSE84135, a time course EMT transition experiment in hSAEC airway epithelial cells; and GSE75487, a 7 day EMT transformation experiment in H358 non-small cell lung cancer cells under doxycycline treatment to induce Zeb1. These are ideal time course scenarios where we would expect that our EMT estimates would reflect an increase in the pseudotime as the experiment progresses. We have reconstructed the EMT pseudotime in these datasets using the same approach as in the rest of the study, and indeed we find it is very well correlated with the timeline of experiment progression through EMT (Supplementary Figure S1d and lines 102-104):

“Furthermore, the reconstructed pseudotime closely matched increasing levels of EMT transformation in independent cell line experiments from a variety of systems (Supplementary Figure S1d), further validating our approach experimentally.”

While these cell line experiments fall short from recapitulating the EMT transformation in human cancers, they do suggest that our EMT methodology is generalisable to a range of systems and is able to capture expected EMT transitions.

3- Using a different EMT inducing protocol (cytokines treatment) in MCF7 and A549 cells, they found that TCGA samples segregated into 5 rather than 3 states and with different outcomes, indicative of a low consistency with the EMT states predictions. An improved EMT pseudotime template based on the TCGA available data and guided by a consensus of EMT trajectory from different *in vitro* data sets could be generated.

We thank the reviewer for this excellent suggestion. We have now implemented the suggested consensus trajectory approach and obtain a much better agreement with both cell line metastatic potential measurements (see Figure 1e, where we see hEMT correlates with a weakly metastatic phenotype, when previously it did not), as well as better validation in longitudinal datasets (Supplementary Figure 1c), as described in points 1 and 2 above.

4- The Hidden Markov Model predicts high stability of the E and hybrid EMT macrostates and an equal transition rate from both E and hybrid EMT to M. Does this prediction hold when cancer samples are analyzed individually?

We cannot apply the HMM model to a single sample, as by definition it requires multiple data points. However, we show that the HMM state inference is robust to variable levels of noise in the system (Supplementary Figure 1e, lines 112-113). Furthermore, the pseudotime value for each sample will always be the same regardless of the cohort within which the sample is analysed, as each sample is projected individually on the reference EMT timeline and the inference only depends on the overall measured transcriptome of that sample. Thus, the pseudotime for each point is independent of other points in the data. The HMM state assignment may show slight variations depending on the cohort e.g. if there are no MES sample in one cancer tissue, but we ask the algorithm for 3 states, some samples will be forced into a mesenchymal label. This is why a pan-cancer approach is best suited to capture a broad range of phenotypes, so that as much of the variation of EMT profiles can be captured and states can be more accurately inferred.

5- Is not clear how EMT states distribution has been measured at tissue level(Figure 1F).

The EMT states distribution is derived based on the pan-cancer pseudotime reconstruction, which is then segregated into 3 discrete states EPI, hEMT and MES, using the HMM model. We then take the samples in individual cancer types and check with state they had been assigned to, so we can summarise all states found in individual cancer types. Therefore, the state of each sample was derived pan-cancer and not individually for each cancer tissue. However, the pseudotime estimation would not change if we performed this on a cancer-by-cancer basis using the same consensus reference, as every bulk sample is independently projected on the PCA containing the reference single cell dataset reflecting the baseline EMT transformation profiles. Because every sample is independently projected, whether we use the entire pan-cancer dataset or a single cancer type will not make a difference to the result. We have now clarified this in the Methods (lines 529-531):

“Because samples are projected individually along the consensus reference single cell data points, the pseudotime estimate only depends on the reference used and not on the specific cohort the sample is part of. Thus, the pseudotime estimates are cohort-independent.”

Furthermore, the EMT pseudotime shows good agreement when using consensus and tissue-specific templates (Supplementary Figure S1c).

6- The authors present a very interesting correlation between the programme in the microenvironment and EMT status in cancer cells. The analysis is clear from the side of the immune microenvironment but misleading for the cancer associated fibroblasts (CAFs). The authors allude to the difficulty in identifying unique markers that discriminate between CAFs and cancer cells undergoing EMT. Indeed, the majority of genes used to discriminate CAFs in Figure 2E have been also described in cancer cells. A way out could be to rely on single-cell OMICs data from cancer patients to avoid mixing of cell types.

The Reviewer makes an excellent point, which was also highlighted by Reviewer #3. While we did acknowledge the possibility of the data being confounded by the CAF signals in our original manuscript, we should have further emphasised that this imposes a major limitation to the analysis. Please see Reviewer#3 point 1 for a more detailed comment on how we have expanded this analysis to address such limitations more comprehensively. Briefly, we have now performed a tumour purity correction to account for signals coming from non-tumour cells in the microenvironment in bulk data (lines 79-89):

“To account for signals from non-tumour cells in the microenvironment, which have been recently shown by Tyler and Tirosh²⁰ to confound the EMT state inference in bulk data, we adjusted the expression of all genes based on the tumour purity inferred from matched DNA-sequencing (see Methods). These corrected expression profiles were then mapped to the single cell reference trajectories and an EMT pseudotime was reconstructed that accounted for potential tumour contamination. Moreover, the confounding effects highlighted by Tyler and Tirosh are prominent when using specific mesenchymal signatures that overlap with markers of cancer-associated fibroblasts (CAFs), but our approach should also be generally less prone to such

biases as we employ a whole-transcriptome reference of single cells progressing through EMT rather than selected markers. Thus, the resulting signal should more reliably capture the transformation of epithelial cells rather than immune/stromal programmes”

This has now enabled us to identify an hEMT state that does not differ in purity compared to EPI and MES samples (Figure S3a), which provides confidence that we are capturing tumour rather than microenvironmental signals:

“The three EMT macro-states we have described within TCGA cancers displayed no significant difference in tumour purity, confirming that non-tumour cell content did not play a significant part in assigning these states (Supplementary Figure S3a).”

CAFs still appear elevated in this state, which we acknowledge could still be due to imperfect correction or it could be a real association (lines 170-178).

We have therefore followed the reviewer’s advice and **have additionally explored single cell as well as spatial transcriptomics datasets** to gain further insights into the interactions established by cells in different EMT states and their microenvironment. These new results complement the insights from bulk tumours and are presented in Figures 3 and 4 of the manuscript (described in detail in Results sections “Spatially-resolved EMT patterning reveals local microenvironmental effects” and “EMT diversity in single cell data”). The results are summarised in the discussion (lines 453-466):

“Our spatial transcriptomics and single cell analyses demonstrate a heterogeneous EMT landscape, delineating clear spatial effects of the continuum of EMT transformation within the tissue. Fibroblasts and cytotoxic T cells often surround more mesenchymal neoplastic areas, and these are occasionally accompanied by hypoxia. While some differential immune recognition is evidenced by co-localisation of MES with CD8/CD4+ T cell signals and hEMT with NK cell signals, partially or fully mesenchymal cells appear to interact less with the microenvironment, potentially due to evasion caused by neoantigen presentation in these more mutated cells⁵⁸. The tumour cells in different EMT states are generally well distinguished from immune cells and the stroma in single cell datasets, with a minority of cells requiring improvement in discrimination methods. While this analysis is limited by our ability to capture a broad spectrum along the EMT transformation as the data are only sourced from early stage cancers, it does lay out a framework for future studies in this space. These should ideally integrate spatial and single cell transcriptomics for a better comprehension of the complex interplay between EMT and the tumour microenvironment.”

7- The higher enrichment in Hypoxia transcripts in hybrid EMT states compared to the M state is very interesting but would need to be validated functionally. Along the same lines, several cancers contain transcripts for leakage/impaired vasculature and hypoxia genes at advanced stages. mRNA in situ analysis would clarify the location of these transcripts and may help to identify the type of microenvironment that could favour the hybrid EMT over the M phenotype.

We have attempted to validate this using spatial transcriptomics data and a 3D microtumour breast cancer model of migration under hypoxia. However, we have observed that the hypoxia landscape across an entire tissue is much more varied depending on the sample and spatial distribution, and thus our original inferences may have been confounded by the bulk nature of the data analysed. We have now altered the text to tone down this claim and instead highlight the more heterogeneous hypoxia associations, which we believe are equally interesting. See Figures 3 and S4 for these new analyses, along with lines 203-206 and 222-237:

“Furthermore, hypoxia was generally increased within areas presenting more advanced mesenchymal transformation, with strongest correlations observed either in hybrid or fully mesenchymal spots (Figures 3d,h,i,p, Supplementary Figure S4d,f). “

“Although some of the patterns are recurrent, there is a high degree of spatial and patient-to-patient variation in EMT and TME composition, suggesting that local spatial effects are likely important determinants of EMT progression. While we found a moderate association between hypoxia and EMT transformed cells in breast cancer tissues examined, the inter-patient and tissue-specific heterogeneity became clearer when examining spatial maps from prostate transcriptomes from Berglund et al³⁴ (Supplementary Figure S5a-b). Here, only one of two tissue sections showed a marked correlation between hypoxia and hEMT within the cancer areas, and not in normal or prostatic intraepithelial neoplasia (PIN). Furthermore, hypoxia was more strongly associated with high rather than moderate levels of EMT transformation in a 3D micro-tumour breast model where collective migration had been induced³⁵ (Supplementary Figure 5c). The associations in this experiment may nevertheless be overshadowed by the fact that early stages of transformation from ductal carcinoma in situ to invasive ductal carcinoma are being investigated. Hypoxia may be a clearer phenotype in more advanced cancers with hEMT or MES phenotypes, which is something we could not capture in these analyses as all tumours originated from stage I or II cancers.“

8- DNA analysis identified some mutational signatures that were more frequent and/or specific for hybrid EMT or M states. Do these mutations show specific enrichment in particular cancer types? To what extent they represent a pan-cancer specific EMT state signature? Again, this is promising but requires the characterization of EMT states. This may help to solve the contradiction found for some genes as RB1 being enriched in the M state in Figure 5E (MCF7 EMT-template) and in the hybrid EMT state in Figure 5F (A549 cells EMT-template).

It is unclear from the reviewer’s comment whether this question refers to mutations in individual cancer drivers or to mutational signatures. However, in either case we can indeed confirm that both cancer drivers as well as mutational signatures could be more frequently observed in some cancers compared to others (e.g. BRAF is very frequent in melanoma, but not so much in other cancers; the UV light signature S7 is also predominant in melanoma, and occasionally appearing in head&neck cancers). However, all our analyses of associations between mutations/mutational signatures and EMT states have been performed while taking into account this tissue-specificity of different drivers and signatures, which we correct for in our models. This is performed in both cases by including the tissue as an additional variable into the model. Therefore, the genomic associations we identify should therefore be more generic, and not just due to a confounding effect. Following the updated EMT methodology, our new results identify different mutational signatures linked with hEMT or MES states, but we see no significant associations after correcting for cancer tissue anymore. We have now clarified this in the text (lines 298-306):

“The mismatch repair deficiency signature SBS6 and the smoking-linked SBS4 signature were significantly increased in hEMT tumours, while SBS39, of unknown aetiology, was most elevated in fully transformed tumours (Figure 5c). The APOBEC mutagenesis signatures SBS2 and SBS13 also appeared elevated in hEMT tumours, in line with observations that inflammation-induced upregulation of the activation-induced cytidine deaminase (AID) enzyme, a component of the APOBEC family, triggers EMT⁴². However, when taking the tissue effect into account in the modelling procedure, no pan-cancer tissue agnostic associations between mutational processes and EMT were identified – suggesting that the previously captured associations are likely

tissue-restricted. Thus, while some influence may exist on EMT from tissue-specific mutational processes, there was no evidence of an overarching mutagen that might induce EMT.”

Similarly, we only present pan-cancer associations between genomic driver events and we do not see strong evidence for tissue specificity, as we have corrected for it, e.g. lines 337-346:

“Larger scale events included deletions of the 4p, 6p and 17p chromosomal arms, all of which harboured cancer drivers which have been previously linked with EMT, e.g. *FGFR3* on 4p⁴⁵, *DAXX* and *TRIM27* on 6p^{46,47}, *TP53* on 17p¹² (Supplementary Table S4). Deletions of the 4p arm appeared in the majority of lung squamous cell and esophageal carcinomas (58% and 50%, respectively), while 6p arm deletions were most frequent in pancreatic, esophageal cancers and adrenocortical carcinomas (>20% in each). 17p arm deletions were the most abundant, especially in ovarian (76%) and kidney chromophobe cancers (76%), with an average of 37% of cases affected per tissue. Therefore, no strong bias in terms of cancer type was observed for these large-scale alterations.”

9- Genomic screening and correlation with EMT states need to be validated at least for some genes to prove the functional relationship. CRISPR/Cas9 editing can be used to analyze the impact of the corresponding genes on specific EMT states.

We thank the reviewer for this important suggestion. We have now validated the association between EMT states and several genomic events using three RNA interference screens (Methods lines 752-766):

“Three large scale public siRNA screens were employed for experimental validation of the proposed genomics associations with EMT. The first dataset from Koedoot et al⁵¹ looked at gene knockdown effects on cell migration abilities in the Hs578T (top panel) and MDA-MB-231 breast cancer cell lines. The data were obtained from the associated publication and contained detailed measurements of effects on cellular phenotype upon knockdown, quantified as changes in cell net surface area, length of minor and major axes, axis ratio (large/small: elongated cells, close to 1: round cells), perimeter score (larger – more migration). Data from further phenotypic tests containing confirmed morphology (big/small round cells) were also available on a subset of the genes.

The second siRNA screen from Penalosa-Ruiz et al⁵⁴ quantified migration-related cell integrity in mouse embryonic fibroblasts through a variety of microscopic measurements of the cells upon gene knockdown across multiple replicates. The data were obtained from the corresponding publication.

The third screen from Meyer-Schaller et al⁵³ focused on the effect of transcription factor knockdown on cell migration in normal murine mammary gland epithelial cells.”

This analysis is presented in the new Figure 6 and the section “Validation of genomic associations” (lines 367-392). We show that 31 or 61 of our identified putative modulators are linked with changes in cellular migration and when knocked down they will display phenotypes of migration inhibition/enhancement. While we acknowledge this analysis does not fully demonstrate a preferential link with either an hEMT or MES phenotype (difficult to test this formally due to the scarcity of hEMT models), it does contribute to the better understanding of the role of these genes in the process of cellular transformation and in their capacity to modulate the migratory capacity of the cells. We also supplement these findings with investigations of metastatic potential changes and essentiality in cancer cell lines, as we had done before (Supplementary Figure

S7 and lines 393-399). Additionally, we have removed any strong claims about links between specific genomic events and hEMT from the abstract and discussion, and highlight that the main value of our analysis is showing how the genome evolution in cancer may drive different EMT trajectories, and that the fully mesenchymal state appears much more strongly determined by these alterations than intermediate states (lines 472-483):

“In contrast, our study identified genomic hallmarks of three EMT macro-states, providing further granularity into how genome-driven cancer evolution shapes EMT trajectories in a state-specific manner. Indeed, we show that distinct genes contribute to the establishment of a fully mesenchymal phenotype, e.g. *RBI* or *DEK*, while others such as *EPAS1*, *FNBPI* or *SMAD4* modulate switches between epithelial and hybrid phenotypes. Furthermore, the genomic distinction in the latter case was less strong than between the extremes of EMT transformation, suggesting that transcriptional or epigenetic alterations may play an increased role in the earlier stages of EMT, while genomic events may further promote and help establish a fully transformed phenotype, which was accurately predicted based solely on genomic alterations. A causal relationship between the acquisition of any of these genomic changes and EMT should be further experimentally tested in the future.”

10- In Figure 4, what is the main difference between a and b, and between c and d? In the analysis of correlation between EMT states/specific mutations and metastatic potential, many genes (i.e. *ERRB4* and *ERRB3*) found in the pan-cancer analysis (Figure 4e) are specific of one type of cancer (Figure 4f). Others, such as *CDKN2A* and *ARID1A*, are associated with higher or lower metastatic potential depending on the type of cancer. Thus, the pan-cancer analysis does not appear to result in conserved features. Please clarify.

We acknowledge that these panels were confusing and indeed not very informative, as they were only checking whether specific genomic alterations identified in our analysis also appeared in other cohorts in metastatic samples. Since it is quite a leap from primary tumours to metastasis and this does not confirm the actual associations with EMT due to lack of matched expression data, we have now removed this from the analysis to simplify the story and avoid any overstatements.

Reviewer #3

1. First, study does not account for the stromal component of bulk expression signatures contributing to the mesenchymal component. The authors admit to this limitation, by stating “To avoid confounding effects between fibroblast and hEMT markers as highlighted by Tyler and Tirosh [ref. 34], we excluded EMT markers from the employed fibroblast gene sets, but we acknowledge that part of the signal recovered may still not be unambiguously attributed to either the cancer or microenvironmental component.”

We had indeed acknowledged this limitation, but following the Reviewer’s comments we took the opportunity to reflect on this very important point further and have decided to revise our analysis to better address this issue. To account for signals coming from non-tumour cells in the microenvironment in bulk data, we have now performed a tumour purity correction of the gene expression signals that are used as input to our method. Specifically, we regress the original expression of every gene in a bulk sample on the purity estimate from the matched DNA-sequencing data (which we believe is the most reliable estimate of tumour purity in bulk tissue). This gives us an adjusted gene expression for every gene where the signal from the tumour microenvironment has been removed/corrected for. We then apply our EMT reconstruction framework on these purity-adjusted gene expression profiles (lines 79-89):

“To account for signals from non-tumour cells in the microenvironment, which have been recently shown by Tyler and Tirosh²⁰ to confound the EMT state inference in bulk data, we adjusted the expression of all genes based on the tumour purity inferred from matched DNA-sequencing (see Methods). These corrected

expression profiles were then mapped to the single cell reference trajectories and an EMT pseudotime was reconstructed that accounted for potential tumour contamination. Moreover, the confounding effects highlighted by Tyler and Tirosh are prominent when using specific mesenchymal signatures that overlap with markers of cancer-associated fibroblasts (CAFs), but our approach should also be generally less prone to such biases as we employ a whole-transcriptome reference of single cells progressing through EMT rather than selected markers. Thus, the resulting signal should more reliably capture the transformation of epithelial cells rather than immune/stromal programmes”

This has now enabled us to identify an hEMT state that does not differ in purity compared to EPI and MES samples (Figure S3a), which provides confidence that we are capturing tumour rather than microenvironmental signals (lines 161-163):

“The three EMT macro-states we have described within TCGA cancers displayed no significant difference in tumour purity, confirming that non-tumour cell content did not play a significant part in assigning these states (Supplementary Figure S3a).”

We would also like to emphasise, as pointed out in the manuscript text above, that we believe our method is less prone to CAF-linked confusion of EMT programmes because we are not using any specific hEMT gene signature. Instead, we are projecting the expression profile of the bulk sample onto a reference full transcriptome profile of epithelial cells going through spontaneous EMT transformation. Thus, we are not reliant on individual markers, but rather compare to entire expression programmes of epithelial cells, not fibroblasts. Nevertheless, we acknowledge this system is not perfect: we indeed see that hEMT samples still have higher fibroblast composition despite our best efforts to correct for tumour purity. This may be due to confounding signals, or to a genuine association between hEMT and higher immune/stromal infiltrations in bulk data. We have now altered the text to acknowledge this limitation more thoroughly (lines 170-178 and 440-446):

“The hEMT samples still displayed a relatively higher level of fibroblasts pan-cancer, despite the tumour purity correction, potentially suggesting a real biological association (Figures 2a-b). In fact, when examining these associations by cancer tissue, we noticed that active fibroblasts were often similarly enriched in hEMT and MES samples compared to epithelial ones (Supplementary Figure S3c), which would be expected with increased tumour progression. However, due to the confounding effects between fibroblast and hEMT markers as highlighted by Tyler and Tirosh²⁰, we acknowledge that part of the signal recovered may still not be unambiguously attributed to either the cancer or microenvironmental component despite our best efforts to correct for this.”

“It is clear that this state is distinct from epithelial tumours, presenting higher CAF infiltration and occasionally enhanced hypoxia and stemness. While it is likely this is an intermediate state in cancer progression along the EMT continuum, as suggested by the longitudinal datasets analysed and the intermediate progression-free intervals, it is also clearly heterogeneous and less genomically influenced than the extreme epithelial and mesenchymal states. Furthermore, the extent to which it is intrinsically rather than environmentally distinct cannot be determined in bulk datasets.”

Nonetheless, there is a limit to the resolution obtained in bulk data. This is why we have now followed suggestions from Reviewers 2 and 3 and have additionally explored single cell as well as spatial transcriptomics dataset to gain further insights into the interactions established by cells in different EMT states and their microenvironment. These new results complement the insights from bulk tumours and are presented in Figures 3 and 4 of the manuscript (described in detail in Results sections “Spatially-resolved EMT patterning reveals local microenvironmental effects” and “EMT diversity in single cell data”). The landscape appears to be much more complex at this level of granularity, and we comment on this extensively in these sections and the discussion (lines 453-466|):

“Our spatial transcriptomics and single cell analyses demonstrate a heterogeneous EMT landscape, delineating clear spatial effects of the continuum of EMT transformation within the tissue. Fibroblasts and cytotoxic T cells often surround more mesenchymal neoplastic areas, and these are occasionally accompanied by hypoxia. While some differential immune recognition is evidenced by co-localisation of MES with CD8/CD4+ T cell signals and hEMT with NK cell signals, partially or fully mesenchymal cells appear to interact less with the microenvironment, potentially due to evasion caused by neoantigen presentation in these more mutated cells⁵⁸. The tumour cells in different EMT states are generally well distinguished from immune cells and the stroma in single cell datasets, with a minority of cells requiring improvement in discrimination methods. While this analysis is limited by our ability to capture a broad spectrum along the EMT transformation as the data are only sourced from early stage cancers, it does lay out a framework for future studies in this space. These should ideally integrate spatial and single cell transcriptomics for a better comprehension of the complex interplay between EMT and the tumour microenvironment.”

What the authors fail to discuss is the main conclusion of Tyler and Tirosh, who state that the hybrid EMT program is highly context-specific. Tyler and Tirosh reach this conclusion with a careful analysis of CAF vs malignant cell components of EMT-related genes. They add that their work “serves as a warning against using any single EMT signature for all cancer types.” Hence the authors’ work appears to directly contradict the findings of Tyler and Tirosh. These contradictions need to be more directly addressed.

The reviewer makes a very important point here that we have not sufficiently clarified. Indeed, Tyler and Tirosh demonstrate that applying any EMT signature in bulk data is context specific by looking at defined EMT signatures. However, we are not applying any specific EMT signature in our analysis; instead, we are using as a reference programme a clean transcriptional system of multiple time points of spontaneous EMT transformation. As highlighted in the introduction of this letter and point 2 below as well, we have now refined this reference to consist of a consensus trajectory across multiple non-transformed and cancer cell lines from different tissues, and have further validated it. Please see point 2 below for a detailed description.

2. Second, the authors derive their EMT macro-state signature from a single cell analysis of MCF10 breast cancer cells that have been profiled at different times during the epithelial to mesenchymal transition in vitro and apply it in a pancancer context. This is not a convincing model system for an analysis of bulk human tumors. What is the rationale that the genomic background of the MCF10 breast cancer cell line serves as a framework for a pancancer analysis? Only in the last section of their paper do they acknowledge a prior study [reference 49] that finds EMT signatures derived from cell lines to be highly context specific. The authors need to have a stronger reason for a pancancer analysis based on the MCF10-derived signature. What would happen if they derived the genomic drivers starting from another cell line?

The Reviewer is right in questioning the validity of the MCF10 single cell dataset as a template for all cancer types. Following this comment and those of Reviewer 2, we have now revised our strategy to reconstructing an EMT timeline pan-cancer. Specifically, we have built a consensus EMT reference from both the non-transformed epithelial cell line MCF10 as well as several cancer cell lines where EMT has been induced via different stimuli, including breast, lung, prostate and ovarian cancer (Methods lines 522-529 and Results lines 72-79):

Results: “Inspired by McFaline-Figueroa et al⁹, we quantified the levels of EMT in these bulk tumours against a consensus reference single cell RNA sequencing (scRNA-seq) dataset derived from non-transformed epithelial cells as well as cancer cells from multiple tissues that have been profiled at different times during the epithelial to mesenchymal transition *in vitro*. These data allowed us to reconstruct a generic “pseudotimeline” of spontaneous EMT transformation onto which we projected the bulk sequenced samples from TCGA, positioning them within the continuum of EMT states (Figure 1a).”

Methods: “Overall, 10 different scRNA-seq datasets were used including A549, MCF7, DU145 and OVCA420 cell lines treated with TGFB1 or TNF. A spontaneous, as well as TGFB1 driven EMT model in MCF10 cell lines was also used. The procedure described above was repeated with each of the 10 reference datasets as input along with the TCGA bulk expression data. This resulted in 10 separate pseudotime estimates for each TCGA bulk-sequenced sample, one based each one of the reference single-cell datasets. The average of the 10 pseudotimes was used to obtain the final pseudotime estimate.”

This has allowed us to have a reference that is much less biased in terms of tissue of origin, and indeed the associations between the newly identified hEMT and MES states with clinical outcomes (Figures 7a-b), as well as validations of metastatic potential in cancer cell lines (see Figure 1e, where we see hEMT correlates with a weakly metastatic phenotype, when previously it did not) have improved. Furthermore, we have validated our methodology in three longitudinal datasets where EMT transformation has been induced and tracked over multiple time points in various cell lines: GSE17708, a time course experiment of A549 lung adenocarcinoma lines treated with TGF-beta; GSE84135, a time course EMT transition experiment in hSAEC airway epithelial cells; and GSE75487, a 7 day EMT transformation experiment in H358 non-small cell lung cancer cells under doxycycline treatment to induce Zeb1. These are ideal time course scenarios where we would expect that our EMT estimates would reflect an increase in the pseudotimeline as the experiment progresses. We have reconstructed the EMT pseudotimeline in these datasets using the same approach as in the rest of the study, and indeed we find it is very well correlated with the timeline of experiment progression through EMT (Supplementary Figure S1d and lines 102-104):

“Furthermore, the reconstructed pseudotimeline closely matched increasing levels of EMT transformation in independent cell line experiments from a variety of systems (Supplementary Figure S1d), further validating our approach experimentally.”

While these cell line experiments fall short from recapitulating the EMT transformation in human cancers, they do suggest that our EMT methodology is generalisable to a range of systems and is able to capture expected EMT transitions.

In terms of tissue-specificity, we have compared the EMT reconstruction performance when using a pan-cancer (consensus) reference versus tissue-specific references and we see very good correlations between pseudotimeline and EMT score estimates (Supplementary Figure S1c and lines 97-101):

“Importantly, when analysing cancer types individually by aligning against breast, lung and prostate reference cell lines rather than to a consensus reference, the pseudotime reconstruction and EMT scores obtained were strongly correlated with those from the pan-cancer analysis (Supplementary Figure S1c), thereby demonstrating that the pan-cancer methodology can broadly recapitulate phenotypes identified in individual cancers.”

3. Third, the authors do not put their work in the context of increasing amount of multi-cancer single cell RNAseq datasets in order to provide a more convincing analysis that their analysis of bulk profiles is truly specific to malignant cells derived from bulk tumors.

Following this reviewer's suggestion and that of Reviewer #2, we have now analysed single cell data from multiple cancer tissues:

- (1) We analysed the dataset from Chung et al, where matched bulk and single cell RNA-sequencing was available for multiple breast tumours. This has allowed us to further ascertain that the estimated EMT pseudotime in bulk tissue is similar to that inferred in single cell data, although we acknowledge the limited sample size here: see Figure 4a and lines 245-252:

“To further investigate this, we employed matched bulk and single cell data from the same cancer patients to test whether the EMT profiles estimated in bulk tissue might capture similar states as those seen at single cell level. Using breast cancer data from Chung et al³⁶, we were able to confirm a good correlation between the estimated EMT pseudotime in bulk and the average EMT signal captured from single cell data (Figure 4a). This provides some further reassurance that the bulk estimates, while fairly generic, do approximate the average signal across the tumour.”

- (2) We performed further single cell analyses in four different cancer types using the dataset from Qian et al. We identify multiple interesting differences in the interactions established between cancer cells in different EMT states and other cells in the immune microenvironment, which are reported in Figure 4 and the entire Results section “EMT diversity in single cell data” (lines 241-273).

In addition, we would like to highlight that at the current time certain analyses are only feasible in bulk data, due to the wide availability of such data and the matched DNA and RNA sequencing which allows us to correlate genomic events with EMT states. This is not currently feasible in single cell data due to the scarcity of such matched DNA and RNA-seq datasets, which makes them underpowered for large-scale analyses. Thanks to the reviewers comments, we believe we now present a much improved manuscript that investigates EMT from multiple angles using orthogonal bulk, single cell and spatial transcriptomics datasets and allows us to uncover a highly diverse landscape with local dependencies that should be further investigated in an integrated manner in the future. We further highlight these limitations and opportunities in the discussion (lines 451-466):

“Undoubtedly, the hEMT state can be further subdivided into sub-states, as shown by Goetz et al⁴ and Brown et al⁵⁷. The true number of EMT intermediate states is just beginning to be explored. However, the noisy bulk sequencing data are limiting our ability to capture them, highlighting the need to complement these studies with spatially-resolved and single cell data.

Our spatial transcriptomics and single cell analyses demonstrate a heterogeneous EMT landscape, delineating clear spatial effects of the continuum of EMT transformation within the tissue. Fibroblasts and cytotoxic T cells often surround more mesenchymal neoplastic areas, and these are occasionally accompanied by hypoxia. While some differential immune recognition is evidenced by co-localisation of MES with CD8/CD4+ T cell signals and hEMT with NK cell signals, partially or fully mesenchymal cells appear to interact less with the microenvironment, potentially due to evasion caused by neoantigen presentation in these more mutated cells⁵⁸. The tumour cells in different EMT states are generally well distinguished from immune cells and the stroma in single cell datasets, with a minority of cells requiring improvement in discrimination methods. While this analysis is limited by our ability to capture a broad spectrum along the EMT transformation as the data are only sourced from early stage cancers, it does lay out a framework for future studies in this space. These should

ideally integrate spatial and single cell transcriptomics for a better comprehension of the complex interplay between EMT and the tumour microenvironment.”

REVIEWER COMMENTS

Reviewer #1 (Remarks to the Author):

All my questions have been addressed.

Reviewer #2 (Remarks to the Author):

I would like to congratulate the authors for the effort they have put into addressing the comments. From my perspective, this is a very much improved manuscript, containing useful information for the community. Specifically, the new consensus EMT pseudotime allows a better organization of the three EMT macrostates proposed. The clarification of signals coming from tumour and non-tumour cells is also appreciated. Finally, the validation approaches using both previously published datasets on EMT in vitro and new information on spatial transcriptomics are very valuable. Having said all this, there are still a few issues that need attention.

The new data emerging now for fibroblasts and endothelial cells do not seem to fit with previous data, revealing inconsistencies in the manuscript in its current form. While fibroblast seems to be more abundant in bulk pan-cancer hEMT (Figure 2b), the spatial analysis suggests the depletion of fibroblasts in the hEMT context (Figure 3g). Also, the latter is not compatible with the intermediate score of the interaction between stromal cells and hEMT from single cell data analysis (Figure 4b). In the case of endothelial cells, spatial analysis suggests low enrichment in MES compared to EPI (Figure 3g) while in the bulk pan-cancer analysis, endothelial cells are enriched at the MES state. All this should be noted and clarified.

It is not clear how the new results help to understand how the tumor microenvironment (TME) impinges into the EMT trajectory. For instance, the results show heterogeneity of the local microenvironment associated with the three EMT macrostates, but there is no evidence of consistent TME changes occurring at specific macrostate transitions, and no functional roles of associations between particular TME and states have been assessed. Thus, this cannot be a prominent part of the conclusions and the

new title "Genomic and local microenvironment effects shaping epithelial-to mesenchymal trajectories in cancer" is not justified in this respect.

Similarly, and as mentioned by the authors, they have "observed that the hypoxia landscape across an entire tissue is much more varied depending on the sample and spatial distribution, and thus our original inferences may have been confounded by the bulk nature of the data analysed". Even though they have toned down the previous claim, the truth is that the analysis of hypoxia is not informative here, as it does not provide any relevant information to understand the EMT trajectories. Thus, this section should be much reduced and just clearly state this point.

Reducing/deleting not relevant or non-informative data and their discussion should also help the manuscript, which is still dense and not that easy to follow. In addition to the reduction in the sections mentioned above, and overall exercise of simplification in the presentation of the main data, and figures in particular, is warranted and would be appreciated by the readers.

Reviewer #3 (Remarks to the Author):

General Impression:

The revised manuscript provides a substantial number of new analyses, most as a response to the reviewers' comments from the first review. While the paper is framed around interesting ideas of EMT, its methodological rigor remains a concern. The additional analyses, while impressive in scope, are hard to assess. They even appear to be contradictory: the authors claim the EMT state of the cancer cells does not depend on the microenvironment and then claim the opposite. In general, the paper is even harder to follow than before and appears to have lost its focus on identifying genomic events contributing to EMT. It is not clear what is the take-home message of the paper and if that message has actually been properly validated.

Specific Feedback:

The authors address the limitations of using one cell line to determine the EMT trajectory and now build a consensus EMT trajectory based on 10 different scRNAseq datasets. They state that they built an EMT pseudotime for each of the 10 dataset and use the average to obtain the final pseudotime estimate. This is described in the methods but no explicit results are shown to assess this entire process. What is the variance across the 10 datasets? How much did the pseudotime estimate change from using the consensus approach vs using the MCF 10 data alone. How sensitive is the downstream analysis to this critical part of the analysis?

The authors address the previous limitation of not correcting for the fibroblast-signature in bulk expression but do not provide any data, even supplementary, to assess the effect of this adjustment.

The authors apply HMM approach used to estimate the EMT states, but provide little detail on how this is done. There is no way to understand the robustness of this finding. It seems that EPI and hEMT states interconvert at roughly the same rates — does that make sense to the authors — if yes, why? It also seems that the a quarter of the MES cells can convert to EPI cells — which seems high and left unjustified. I would expect that these transition probabilities could be a function of the mutational background (a premise of the paper) but that is not considered in the HMM analysis. Perhaps this is the reason there was not mutagen event significantly associated with EMT transformation (lines 304-306)?

The authors claim that the EMT states are not influenced by the microenvironment: “The three EMT macro-states we have described within TCGA cancers displayed no significant difference in tumour purity, confirming that non-tumour cell content did not play a significant part in assigning these states.” This is a surprising result, particularly for early stage disease as others have demonstrated that myofibroblasts can alter the EMT state of cancer cells. This is a point that the authors acknowledge as ambiguous.

When the authors look at spatial transcriptomic data they observe significant EMT spatial heterogeneity, yet the rest of their paper rests on the idea that a bulk tumor specimen can have an EMT state. They do not address this contradiction, and instead from the spatial transcriptomics analysis they claim the importance of local EMT effects in tissues. When analyzing the single cell RNAseq data, they acknowledge the role of the microenvironment on the EMT of the cancer cells needs to be better understood.

It is unclear if the analysis of analysis of the publicly available siRNA screen data statistically validates any new insights derived from this study. The authors claim that 31 of 61 targets results in morphological changes that can be linked to EMT. What are the 61 targets? Is enrichment of 31 of 61 targets significant via a hypergeometric analysis, sampling from the Koedoot et al siRNA screening data?

Revision for manuscript:

“Genomic and local microenvironment effects shaping epithelial-to-mesenchymal trajectories in cancer”
(Malagoli-Tagliazzuchi, Wiecek et al)

Summary of manuscript changes:

We are very grateful for the Reviewers' suggestions, which have helped us strengthen our findings. Here is a summary of the key modifications we have made to the analysis and manuscript following these suggestions:

1. Performed additional comparisons between EMT pseudotime trajectories obtained from individual single cell references and the consensus one, to demonstrate the robustness of the latter.
2. Highlighted the effects of the tumour purity correction step before EMT trajectory reconstruction.
3. Performed further analyses to assess the robustness of the HMM state determination model.
4. Employed new methods to infer cell-cell interactions in spatial transcriptomics data in order to consolidate and more clearly delineate the relations between tumour cells in different EMT states and the tumour microenvironment. This also allowed us to observe how frequently tumour cells in these different macro-states interact with each other.
5. Performed a hypergeometric test to check for a significant enrichment of new EMT genomic targets in siRNA screens.
6. Restructured and expanded the spatial transcriptomics and single cell analyses with further discussion and interpretation of the results obtained; the discrepancies as well as key messages were further emphasised in the Discussion section.
7. Reordered the manuscript sections such that the genomic analyses are presented before the tumour microenvironment findings, which confers a more logical, uninterrupted flow from bulk to spatial/single cell investigations.
8. Simplified the manuscript figures by reorganising Figures 1 and 2 so that the methodology is separated from the pan-cancer TCGA landscape, and redesigned Figures 3 and 6 by removing technical details and keeping key panels only, to make it easier for the readers to grasp the main messages of the manuscript.

A detailed, point-by-point description of the changes implemented in order to address the Reviewers' comments is included below. The resubmitted manuscript contains tracked changes with updated Figures 1, 2, 3 and 6, updated Supplementary Figures S1, S2, S4, S6, S9, including additional panels with new analyses in Figure 1b, 6k-l, Supplementary Figures S1b,f, S2c, S9c,d, and a new Supplementary Table S5. We have also rewritten the abstract to reduce the claim on local microenvironment effects and changed the title to “Genomic and microenvironmental heterogeneity shaping the epithelial-to-mesenchymal trajectories in cancer”.

Reviewer #2:

I would like to congratulate the authors for the effort they have put into addressing the comments. From my perspective, this is a very much improved manuscript, containing useful information for the community. Specifically, the new consensus EMT pseudotime allows a better organization of the three EMT macrostates proposed. The clarification of signals coming from tumour and non-tumour cells is also appreciated. Finally, the validation approaches using both previously published datasets on EMT

in vitro and new information on spatial transcriptomics are very valuable. Having said all this, there are still a few issues that need attention.

1. The new data emerging now for fibroblasts and endothelial cells do not seem to fit with previous data, revealing inconsistencies in the manuscript in its current form. While fibroblast seems to be more abundant in bulk pan-cancer hEMT (Figure 2b), the spatial analysis suggests the depletion of fibroblasts in the hEMT context (Figure 3g). Also, the latter is not compatible with the intermediate score of the interaction between stromal cells and hEMT from single cell data analysis (Figure 4b). In the case of endothelial cells, spatial analysis suggests low enrichment in MES compared to EPI (Figure 3g) while in the bulk pan-cancer analysis, endothelial cells are enriched at the MES state. All this should be noted and clarified.

The Reviewer is right in pointing out the discrepancies between some of the bulk, spatial and single cell analyses. Given the very different way of sampling the tumour for each technique, this is not entirely surprising. However, we acknowledge that the overall message can become confused and we need to make more effort to reconcile the results from these parallel analyses. We have now reorganised the spatial transcriptomics figure so that a clearer message is conveyed by moving unnecessary panels (e.g. clustering details) to the supplementary material – see the new Figure 7 and Supplementary Figure S9. The analyses shown previously were only looking at enrichment of different immune/stromal cells within spots where there was a dominant EPI, hEMT or MES state. We have now performed additional analyses where we infer cell-cell interaction networks from the spatial transcriptomics data and show that the fraction of interactions with fibroblasts generally increases as the EMT stage advances in the Visium dataset (Figure 7k), but can also be highest in hEMT in the ST2K dataset (Supplementary Figure S9c). This is broadly in line with the bulk data observations. The strongest associations are observed between cells in a mesenchymal state and CD8+ T cells or fibroblasts across bulk and spatial transcriptomics datasets, while the hEMT associations tend indeed to be more variable. In single cell data, hEMT cells seem to interact less with non-tumour cells in most datasets, which may be because of the volatility of the interactions they establish spatially or the incapacity of the immune system to recognise the cancer cells once they have switched to an hEMT state. We believe all these analyses suggest that EMT transformation is linked with immune evasion, as stipulated in other studies too, and potentially hEMT cells may be even more effective at avoiding immune detection, although this requires further investigation. We also acknowledge that the spatial transcriptomics datasets are limited in number and only sample early stage cancer, which may limit our ability to capture a broad range of EMT phenotypes. Indeed, other cells such as endothelial cells show more variable patterns across different profiling technologies and we now more clearly state this. We have considerably expanded the spatial transcriptomics, single cell and discussion sections to highlight, comment on and speculate the reasons for the observed discrepancies in bulk, spatial and single cell sequencing datasets as follows:

Observations on bulk, Results section lines 419-425: “We observed that cytotoxic, γ δ T and endothelial cells were progressively enriched with increased stages of EMT transformation (Figure 5a-b, Supplementary Figure S7a), suggesting that the fully mesenchymal state is most often associated with “immune hot” tumours. In line with this hypothesis, these tumours also showed the highest exhaustion levels (Figure 5c). This links well with our previous observations of increased clonal mutations in the MES tumours (Figure 3a), which could generate higher neoantigen loads thereby triggering immune evasion⁵⁰.”

Observations on spatial transcriptomics:

>Results lines 512-515: “The most striking spatial pattern was that of fibroblasts, which surrounded the epithelial neoplastic areas proportionally to the increase in EMT state, appearing more strongly linked with highly transformed areas of the tumour (Figures 6b,e,h).”

>Results lines 523-530: “The hEMT state shows a relative depletion of fibroblasts compared to the MES areas in the Visium slides, which contrasts the signals seen in bulk data. However, within a larger dataset of multi-region spatial transcriptomics slides from multiple patients profiled using ST2K, we found that intermediate levels of transformation (hEMT) uniquely associated both with MSC/iCAF-like and

myCAF-like cells, whereas the EPI state was only linked with the former and the MES state with the latter (Figure 6j). Thus, the heterogeneity of hEMT-CAF associations may be explained by different subtypes of CAFs present in the context of hEMT and MES samples."

> **Results lines 532-571:** "It is worth noting that other cell types, such as endothelial cells, showed more heterogeneous associations from sample to sample and recapitulated to a lesser extent the observations obtained in bulk tissue.

Beyond inspecting enrichment of cell populations within the spatial transcriptomics spots, we also inferred cell-cell interactions within the spatially profiled slides based on signal co-localization. We find that cells in the fully mesenchymal state tend to interact more frequently with fibroblasts, CD8+ and CD4+ T cells compared to the ones in a hybrid or epithelial state, the latter two showing no marked differences in the fraction of such interactions established (Figure 7k, Supplementary Figure S9c). This further confirms the associations between transformed tumour cells and an immunogenic environment, which may be suppressed via CAFs.

Finally, we also inspected how the tumour cells themselves interact with each other (Figure 7l). Overall, we see that cancer cells at the extremes of the EMT transformation (either epithelial or fully mesenchymal) tend to interact more with cells in the same state. In contrast, there was no marked difference between the fraction of interactions established between hEMT cells and tumour cells in any state – implying that the hybrid state may be more mobile or more easily reached from any other state, as our HMM model had also suggested. The patterns were highly similar in the Visium and ST2K profiled slides (Figure 7l, Supplementary Figure S9d). "

Observations on single cell data:

>**Results lines 619-621:** "Within this dataset, the number of interactions with non-tumour cells increased with increasing EMT transformation, closely reflecting the observations in bulk tumours and spatial transcriptomics, particularly with fibroblasts and T cells."

>**Results lines 637-643:** "This stark contrast to the observations in bulk and, to a lesser extent, also in the spatially profiled datasets could potentially imply that while immune recognition may initially happen at more advanced stages of EMT within localised areas of the tissue that become gradually transformed, the physical interactions established with these immune cells are rather volatile and the transformed cells might quickly find ways to escape immune detection, possibly with the help of CAFs for the cells in a mesenchymal state and via other mechanisms for hybrid cells."

Finally, we have expanded the Discussion section as follows (lines 742-773):

"Our spatial transcriptomics analysis demonstrates a heterogeneous EMT landscape, delineating clear spatial effects of the continuum of EMT transformation within the tissue. Thus, we gain a new appreciation of the diversity of EMT states within a single tumour, which needs to be accounted for in future studies. However, this discovery does not detract from our analyses in bulk tumours: while considering a single sample as having a unique state was clearly a simplification, the signal captured from the tumour still reflects the average state of the cells, likely captures well the extremes of the EMT distribution when scanned across thousands of TCGA samples and is the only setting that makes genomic associations possible currently. In the future, it will be important to verify these findings in matched DNA/RNA datasets from spatially or single cell sequenced samples.

In the spatially profiled samples that we analysed, fibroblasts and cytotoxic T cells often surrounded more mesenchymal neoplastic areas, and more frequent interactions with these cells were observed in this context. There was evidence for initial immune recognition as suggested by the co-localisation of MES with CD8/CD4+ T cell signals and hEMT with NK cell signals, and this was further backed up by the increased mutational burden of tumours in hEMT and MES states, which could lead to higher neoantigen presentation and subsequent exhaustion of T cells⁵⁰. Thus, these data consolidate evidence for a link between EMT progression and immune evasion which was already well documented in the literature^{62,63}. Nevertheless, this analysis is limited by the small sample size and our ability to capture a broad spectrum along the EMT transformation as the data are only sourced from early stage cancers. Larger spatial datasets and a finer grained resolution of EMT states, especially hybrid ones, will be

required in order to understand the more complex relationships established at intermediate EMT stages, which are less clear than for the fully mesenchymal states. The combined data from spatial and single cell sequencing seem to indicate that hEMT cells are even more successful in avoiding immune detection than MES cells, as suggested by previous studies^{64,65}, but this hypothesis needs further investigation. This will also require new methods to identify localised, context-specific effects within the tissue which may not generalise throughout the tumour. Despite these limitations, our analyses do serve as a proof of concept for the ability to survey EMT spatially and lay out a framework for future studies in this space. These should ideally integrate spatial and single cell transcriptomics for a better comprehension of the complex interplay between EMT and the tumour microenvironment. “

Methods description of cell-cell interaction inference in spatial transcriptomics data (lines 1144-1150):

“Inference of interaction networks

The ScanPy⁸⁷ (Single-Cell Analysis in Python) and SquidPy⁸⁸ (Spatial Single Cell Analysis in Python) packages were used for graph analysis on the Visium spatial slides. This included graph visualisation and graph metric algorithms. The STUtility⁵⁶ package was adapted and used to create graphs from the ST2K spatial slides. The deconvolved spot results were used to assign node labels. Edges were assigned based on the spot neighbours. NetworkX⁸⁹ was used for querying and further analysis of the networks.”

2. It is not clear how the new results help to understand how the tumor microenvironment (TME) impinges into the EMT trajectory. For instance, the results show heterogeneity of the local microenvironment associated with the three EMT macrostates, but there is no evidence of consistent TME changes occurring at specific macrostate transitions, and no functional roles of associations between particular TME and states have been assessed. Thus, this cannot be a prominent part of the conclusions and the new title "Genomic and local microenvironment effects shaping epithelial-to mesenchymal trajectories in cancer" is not justified in this respect.

In light of the new analyses, the most consistent TME changes appear to be between the MES state and CD8+ T cells or fibroblasts. However, we acknowledge there are limitations to these claims given the broad heterogeneity observed in spatial and single cell datasets. As presented in the ample discussion highlighted above in the previous question, we have made an effort to reconcile the results across the bulk, spatial and single cell datasets by highlighting common trends as well as discrepancies and limitations. We have toned down any claims beyond the ones that are backed up by the literature and have changed both the spatial transcriptomics subsection title (now “Spatial transcriptomics reveals heterogeneous EMT patterning within the tissue”) as well as the manuscript title (“Genomic and microenvironmental heterogeneity shaping epithelial-to-mesenchymal trajectories in cancer”). We have also altered the abstract to remove any claims about local effects on EMT, which indeed have not been demonstrated and require more complex analyses (line 19): “We further employed spatial transcriptomics and single cell datasets to show extensive spatial heterogeneity of EMT transformation and distinct interaction patterns with cytotoxic, NK cells and fibroblasts in the tumour microenvironment.”

3. Similarly, and as mentioned by the authors, they have "observed that the hypoxia landscape across an entire tissue is much more varied depending on the sample and spatial distribution, and thus our original inferences may have been confounded by the bulk nature of the data analysed". Even though they have toned down the previous claim, the truth is that the analysis of hypoxia is not informative here, as it does not provide any relevant information to understand the EMT trajectories. Thus, this section should be much reduced and just clearly state this point.

The Reviewer is right that the associations are too sparse and heterogeneous to warrant such a large space in the paper. As suggested, we have now removed any analysis of hypoxia from the spatial transcriptomics data and have moved the additional analyses in prostate multi-region datasets and the 3D model of migration in the section investigating hypoxia associations in bulk. We have also altered the text to highlight that these relationships are likely tissue-specific (Results section lines 459-477):

“Samples with a transformed phenotype (MES, hEMT) presented significantly elevated hypoxia levels in

several cancer types (Figure 5d). Hypoxia has previously been shown to promote EMT by modulating stemness properties⁵¹, but our analyses indicate this may be tissue and context specific. Indeed, when examining distinct tissue sections from prostate tumours from Berglund et al⁵², only one of two tissue sections showed a marked correlation between hypoxia and hEMT within the cancer areas, and not in normal or prostatic intraepithelial neoplasia (PIN) (Supplementary Figure S8a-b). Furthermore, hypoxia was more strongly associated with high rather than moderate levels of EMT transformation in a 3D micro-tumour breast model where collective migration had been induced⁵³ (Supplementary Figure S8c). Thus, while hypoxia may create a favourable environment for EMT transformation in certain contexts, the analysed datasets provide no evidence for it being an obligatory condition.[...] Thus, the interplay between hypoxia and stemness may play a greater role during the intermediate rather than advanced stages of EMT transformation, although this is likely tissue specific and requires further mechanistic investigation. “

4. Reducing/deleting not relevant or non-informative data and their discussion should also help the manuscript, which is still dense and not that easy to follow. In addition to the reduction in the sections mentioned above, and overall exercise of simplification in the presentation of the main data, and figures in particular, is warranted and would be appreciated by the readers.

We thank the Reviewer for this suggestion. We have now reorganised the manuscript sections such that there is a clearer rationale and validation for our EMT methodology, which is a key aspect of the paper which needed to be emphasised. We are also presenting the genomic associations before the tumour microenvironment ones. We believe this helps improve the flow and provides a more logical, uninterrupted flow from bulk to single cell analyses. We have also extensively reorganised and simplified the following figures:

- Figure 1 – now focused only on the EMT methodology and its application in various bulk datasets and cell line validation sets; panels (a), (e), (f), (g) and (h) have been modified with more explicit labels and additional annotations to make the figures easier to read
- Figure 2 – a new simple figure which presents the distribution of EMT states across cancer types and stages
- Figure 3 – the genomic associations figure has been highly simplified by moving the lesser informative panels with the mutational signature and cancer cell fraction analyses to the supplementary material; we have also reordered the figure panels in 3(a) such that the mutational burden measurements (total, clonal, subclonal) are grouped together; for the genomic drivers associated with different EMT states, we amply annotate Fig.3(c) to make the figure more comprehensible and easier to read.
- Figure 4 – the validation figure employing the siRNA screens has been simplified by moving the bottom panel which presented a different siRNA screen to the supplementary material
- Figure 6 – we have moved the clustering details and UMAP previously included to the supplementary material, as these were technical aspects that were not key to understanding the message; we have also reduced the number of spatial transcriptomic profiles to only the ones that show tangible pattern
- Figure 8 – improved annotation for the drug response data

Reviewer #3:

General Impression:

The revised manuscript provides a substantial number of new analyses, most as a response to the reviewers' comments from the first review. While the paper is framed around interesting ideas of EMT, its methodological rigor remains a concern. The additional analyses, while impressive in scope, are hard to assess. They even appear to be contradictory: the authors claim the EMT state of the cancer cells does not depend on the microenvironment and then claim the opposite. In general, the paper is even harder to follow than before and appears to have lost its focus on identifying genomic events contributing to EMT.

It is not clear what is the take-home message of the paper and if that message has actually been properly validated.

We thank the Reviewer for these important observations. Indeed, the revised manuscript had a much stronger focus on the tumour microenvironment associations than the originally submitted one, which could detract from the genomic analyses. To address these concerns, we have decided to reorganize the paper such that the analyses of intrinsic (genomic) features of EMT are presented before any tumour microenvironment findings in bulk, spatial transcriptomics and single cell datasets. We believe this provides a more logical, uninterrupted flow from intrinsic to extrinsic features, and from bulk to spatial and single cell profiling. We have also placed much more emphasis on assessing the robustness of the EMT scoring methodology following the comments below (points 1 and 4), and highlight this more clearly as a resource/tool that can be applied in a variety of biological contexts to study EMT.

With regards to the tumour microenvironment, we would like to highlight that we had never claimed that the EMT state of the cancer cells does not have any association with the microenvironment – our phrasing may have been unclear, but we simply aimed to remove any confounding signals that may prevent an accurate assessment of EMT within the cancer cells themselves. Indeed, associations between EMT progression and the microenvironment have been well documented in the literature and we find similar trends in our analysis. This is further explained in question 8 below.

To make the take-home messages clearer, we have extensively reorganized some of the figures and parts of the text, particularly with respect to the EMT trajectory determination, and the results from the bulk, spatial transcriptomics and single cell analyses. We have attempted to reconcile some of the discrepancies between bulk, spatial and single cell profiling by adding new cell-cell interaction analyses on the spatial transcriptomics datasets and further interpretation of the results, although we acknowledge that some differences remain that are likely due to different technologies being used in the data generation process. We comment on this more extensively in the discussion and have limited our claims about any local microenvironment effects on EMT, all of which we have described in detail in questions 8 and 9 below.

Here is a brief summary of key changes we have made in the analysis and interpretation of the tumour microenvironment interactions:

- The spatial transcriptomics figure has been simplified so that a clearer message is conveyed by moving unnecessary panels (e.g. clustering details) to the supplementary material – see the new Figure 7 and Supplementary Figure S9.
- The analyses shown previously were only looking at enrichment of different immune/stromal cells within spots where there was a dominant EPI, hEMT or MES state. We have now performed additional analyses where we infer cell-cell interaction networks from the spatial transcriptomics data and show that the fraction of interactions with fibroblasts generally increases as the EMT stage advances in the Visium dataset (Figure 7k), but can also be highest in hEMT in the ST2K dataset (Supplementary Figure S9c). This is broadly in line with the bulk data observations.
- The strongest associations are observed between cells in a mesenchymal state and CD8+ T cells or fibroblasts across bulk and spatial transcriptomics datasets, while the hEMT associations tend indeed to be more variable. In single cell data, hEMT cells seem to interact less with non-tumour cells in most datasets, which may be because of the volatility of the interactions they establish spatially or the incapacity of the immune system to recognise the cancer cells once they have switched to an hEMT state.
- We believe all these analyses suggest that EMT transformation is linked with immune evasion, as stipulated in other studies too, and potentially hEMT cells may be even more effective at avoiding immune detection, although this requires further investigation.
- We also acknowledge that the spatial transcriptomics datasets are limited in number and only sample early stage cancer, which may limit our ability to capture a broad range of EMT phenotypes.

We have considerably expanded the spatial transcriptomics, single cell and discussion sections to highlight, comment on and speculate the reasons for the observed discrepancies in bulk, spatial and single cell sequencing datasets as follows:

Observations on bulk, Results section lines 419-425: “We observed that cytotoxic, $\gamma \delta$ T and endothelial cells were progressively enriched with increased stages of EMT transformation (Figure 5a-b, Supplementary Figure S7a), suggesting that the fully mesenchymal state is most often associated with “immune hot” tumours. In line with this hypothesis, these tumours also showed the highest exhaustion levels (Figure 5c). This links well with our previous observations of increased clonal mutations in the MES tumours (Figure 3a), which could generate higher neoantigen loads thereby triggering immune evasion⁵⁰.”

Observations on spatial transcriptomics:

>**Results lines 512-515:** “The most striking spatial pattern was that of fibroblasts, which surrounded the epithelial neoplastic areas proportionally to the increase in EMT state, appearing more strongly linked with highly transformed areas of the tumour (Figures 6b,e,h). “

>**Results lines 523-530:** “The hEMT state shows a relative depletion of fibroblasts compared to the MES areas in the Visium slides, which contrasts the signals seen in bulk data. However, within a larger dataset of multi-region spatial transcriptomics slides from multiple patients profiled using ST2K, we found that intermediate levels of transformation (hEMT) uniquely associated both with MSC/iCAF-like and myCAF-like cells, whereas the EPI state was only linked with the former and the MES state with the latter (Figure 6j). Thus, the heterogeneity of hEMT-CAF associations may be explained by different subtypes of CAFs present in the context of hEMT and MES samples.”

> **Results lines 532-571:** “It is worth noting that other cell types, such as endothelial cells, showed more heterogeneous associations from sample to sample and recapitulated to a lesser extent the observations obtained in bulk tissue.

Beyond inspecting enrichment of cell populations within the spatial transcriptomics spots, we also inferred cell-cell interactions within the spatially profiled slides based on signal co-localization. We find that cells in the fully mesenchymal state tend to interact more frequently with fibroblasts, CD8+ and CD4+ T cells compared to the ones in a hybrid or epithelial state, the latter two showing no marked differences in the fraction of such interactions established (Figure 7k, Supplementary Figure S9c). This further confirms the associations between transformed tumour cells and an immunogenic environment, which may be suppressed via CAFs.

Finally, we also inspected how the tumour cells themselves interact with each other (Figure 7l). Overall, we see that cancer cells at the extremes of the EMT transformation (either epithelial or fully mesenchymal) tend to interact more with cells in the same state. In contrast, there was no marked difference between the fraction of interactions established between hEMT cells and tumour cells in any state – implying that the hybrid state may be more mobile or more easily reached from any other state, as our HMM model had also suggested. The patterns were highly similar in the Visium and ST2K profiled slides (Figure 7l, Supplementary Figure S9d). “

Observations on single cell data:

>**Results lines 619-621:** “Within this dataset, the number of interactions with non-tumour cells increased with increasing EMT transformation, closely reflecting the observations in bulk tumours and spatial transcriptomics, particularly with fibroblasts and T cells.”

>**Results lines 637-643:** “This stark contrast to the observations in bulk and, to a lesser extent, also in the spatially profiled datasets could potentially imply that while immune recognition may initially happen at more advanced stages of EMT within localised areas of the tissue that become gradually transformed, the physical interactions established with these immune cells are rather volatile and the transformed cells might quickly find ways to escape immune detection, possibly with the help of CAFs for the cells in a mesenchymal state and via other mechanisms for hybrid cells.”

Finally, we have expanded the Discussion section as follows (lines 742-773):

“Our spatial transcriptomics analysis demonstrates a heterogeneous EMT landscape, delineating clear spatial effects of the continuum of EMT transformation within the tissue. Thus, we gain a new

appreciation of the diversity of EMT states within a single tumour, which needs to be accounted for in future studies. However, this discovery does not detract from our analyses in bulk tumours: while considering a single sample as having a unique state was clearly a simplification, the signal captured from the tumour still reflects the average state of the cells, likely captures well the extremes of the EMT distribution when scanned across thousands of TCGA samples and is the only setting that makes genomic associations possible currently. In the future, it will be important to verify these findings in matched DNA/RNA datasets from spatially or single cell sequenced samples.

In the spatially profiled samples that we analysed, fibroblasts and cytotoxic T cells often surrounded more mesenchymal neoplastic areas, and more frequent interactions with these cells were observed in this context. There was evidence for initial immune recognition as suggested by the co-localisation of MES with CD8/CD4+ T cell signals and hEMT with NK cell signals, and this was further backed up by the increased mutational burden of tumours in hEMT and MES states, which could lead to higher neoantigen presentation and subsequent exhaustion of T cells⁵⁰. Thus, these data consolidate evidence for a link between EMT progression and immune evasion which was already well documented in the literature^{62,63}. Nevertheless, this analysis is limited by the small sample size and our ability to capture a broad spectrum along the EMT transformation as the data are only sourced from early stage cancers. Larger spatial datasets and a finer grained resolution of EMT states, especially hybrid ones, will be required in order to understand the more complex relationships established at intermediate EMT stages, which are less clear than for the fully mesenchymal states. The combined data from spatial and single cell sequencing seem to indicate that hEMT cells are even more successful in avoiding immune detection than MES cells, as suggested by previous studies^{64,65}, but this hypothesis needs further investigation. This will also require new methods to identify localised, context-specific effects within the tissue which may not generalise throughout the tumour. Despite these limitations, our analyses do serve as a proof of concept for the ability to survey EMT spatially and lay out a framework for future studies in this space. These should ideally integrate spatial and single cell transcriptomics for a better comprehension of the complex interplay between EMT and the tumour microenvironment. “

Methods description of cell-cell interaction inference in spatial transcriptomics data (lines 1144-1150):

“Inference of interaction networks

The ScanPy⁸⁷ (Single-Cell Analysis in Python) and SquidPy⁸⁸ (Spatial Single Cell Analysis in Python) packages were used for graph analysis on the Visium spatial slides. This included graph visualisation and graph metric algorithms. The STUtility⁵⁶ package was adapted and used to create graphs from the ST2K spatial slides. The deconvolved spot results were used to assign node labels. Edges were assigned based on the spot neighbours. NetworkX⁸⁹ was used for querying and further analysis of the networks.”

Specific Feedback:

1. The authors address the limitations of using one cell line to determine the EMT trajectory and now build a consensus EMT trajectory based on 10 different scRNAseq datasets. They state that they built an EMT pseudotime for each of the 10 dataset and use the average to obtain the final pseudotime estimate. This is described in the methods but no explicit results are shown to assess this entire process. What is the variance across the 10 datasets? How much did the pseudotime estimate change from using the consensus approach vs using the MCF 10 data alone. How sensitive is the downstream analysis to this critical part of the analysis?

We thank the Reviewer for the suggestion of providing further details on the various single cell references and the consequences on building EMT pseudotimelines from each of them. We had included some comparisons of the pseudotime and EMT scores derived using the consensus reference versus the tissue-specific references already in Supplementary Figure 1, but we agree with the Reviewer that some parts of the process are not fully transparent and further checks are required as this is a critical step that all downstream analyses rely upon. We have now reorganised Figure 1 and

Supplementary Figure 1 such that the focus falls on thoroughly assessing and validating the EMT methodology rather than mixing this with further information on the EMT landscape in TCGA.

To further address the Reviewer's question about the variance across the 10 datasets, we have added a new panel in Figure 1b where we show the EMT pseudotime distribution for each single cell reference employed, along with the one derived from the combined consensus reference. This makes the process more transparent and allows a clearer assessment of differences between single cell templates and their variance. We show that the consensus template captures a spread of the EMT timeline that is broadly centered on mid and higher levels of transformation, thus reconciling some of the skewed ranges of other templates and being more in line with what would be expected to be observed in bulk cancer samples from multiple disease stages (Results lines 82-106):

"Inspired by McFaline-Figueroa et al⁹, we surveyed multiple single cell RNA sequencing (scRNA-seq) datasets where immortalized non-malignant epithelial cells or cancer cells from various tissues have been profiled at different times during the epithelial to mesenchymal transition *in vitro* (see Methods). These single cell profiles can act as a reference for EMT transformation in a healthy or cancer setting, and in the following tissues: breast, lung, ovary and prostate. Hence, these data allowed us to reconstruct tissue-specific "pseudo-timelines" of spontaneous EMT transformation (Figure 1b). These pseudo-timelines captured a broader or narrower range of EMT transformation, with lesser transformed (i.e. closer to 0) or more transformed (closer to 100) states depending on the tissue. Given the inter-tissue variation of the EMT programme and the limited tissue diversity in the available single cell datasets profiled in this manner, we reasoned that a consensus reference timeline of EMT combining all these datasets would reflect a broader and more generalisable range of EMT phenotypes that we could then use as a baseline for quantifying EMT in any cancer sample (see Methods). The resulting consensus EMT reference indeed yielded a pseudo-timeline that was more specifically centred on mid-transformed and higher-transformed EMT states (Figure 1b), which would be more often expected in random cancer samples across various stages of the disease."

To compare the effects of using the consensus reference versus a tissue specific reference on the pseudotime reconstruction and the EMT scoring, we had already provided the current Supplementary Figure 1e. Furthermore, we have followed the Reviewer's suggestion and provide a scatter plot comparing the pseudotime values derived from the consensus template versus the MCF10 data (mock i.e. spontaneous, TGFB-stimulated, or both), shown in Supplementary Figure 1f. We observe that there is a good correlation between the individual tissue-specific pseudotimelines and the consensus one, suggesting that the downstream analysis is unlikely to be affected by large changes in EMT phenotype when estimated using different references. All of this is now documented in lines 139-143:

"Importantly, when analysing cancer types individually by aligning against breast, lung and prostate reference cell lines rather than to a consensus reference, the pseudotime reconstruction and EMT scores obtained were strongly correlated with those from the pan-cancer analysis (Supplementary Figure S1e), even in the case of the non-transformed breast cell line MCF10 (Supplementary Figure S1f)."

Moreover, we have validated our consensus methodology experimentally in three independent *in vitro* experiments which we had originally included in Supplementary Figure 1. We have now moved one of the panels to Figure 1c to stress this point further as it demonstrates that the consensus template is applicable in a range of cell systems and tissues (lines 106-109):

"Furthermore, the reconstructed pseudotime closely matched increasing levels of EMT transformation in independent cell line experiments from a variety of systems (Figure 1c, Supplementary Figure S1a), validating our approach experimentally."

2. The authors address the previous limitation of not correcting for the fibroblast-signature in bulk expression but do not provide any data, even supplementary, to assess the effect of this adjustment.

We apologise for not being clear enough in the text regarding the consequences of this adjustment, we have done this for brevity reasons. We had indeed checked that the tumour purity correction has had the desired result, i.e. that there would be no differences in overall purity between samples in the three macro-states EPI, hEMT and MES. We had shown this in Supplementary Figure 3 originally at the start of the tumour microenvironment analysis. However, we acknowledge that the text may have been ambiguous in that context, and this check should have been presented earlier in the manuscript. We have now corrected this and present this analysis in the paragraph where the three EMT macro-states are introduced (Supplementary Figure 2b), lines 153-155:

“These states were robust to varying levels of gene expression noise (Supplementary Figure S2a), and were not influenced by sample purity, as expected due to the correction step before pseudotime reconstruction (Supplementary Figure S2b). “

Furthermore, we acknowledge the need to assess the consequences of this adjustment even earlier at the moment of the pseudotime reconstruction. We have now included a new Supplementary Figure 1b where we show that following the purity correction there is no notable correlation observed between the EMT pseudotime and the tumour purity or the fibroblast content of the tumour. This is reassuring, as it suggests that the EMT trajectories inferred are independent of any potential biases caused by microenvironmental effects. We clarify this aspect in the Results section, lines 117-119:

“Indeed, after this correction there was no notable correlation observed between the reconstructed EMT pseudotime and the sample purity or the amount of fibroblast infiltration in the tumour (Supplementary Figure S1b).”

3. The authors apply HMM approach used to estimate the EMT states, but provide little detail on how this is done.

We have explained the HMM approach in the Methods section “Segmentation of the EMT trajectory and robustness evaluation”, which we have now slightly expanded to make the logic behind the process clearer (lines 906-922):

“We used a Hidden Markov Model approach to identify of a discrete number of EMT states, with the assumption that the observed expression profiles of samples that are ordered along an EMT timeline underlie multiple unobserved states. The input of this analysis was an expression matrix (M) where the rows were the TCGA samples (N) and the columns the gene markers (G) of EMT (see the section “Computation of the EMT scores” below for the list of genes). The original N columns were sorted for the t values of the pseudotime (P). This matrix and P were provided as input for a lasso penalized regression. P was used as response variable, the genes as the independent variables. The non-zero coefficients obtained from this analysis were selected to create a sub-matrix of M that was used as input for a Hidden Markov Model.

Different HMM models were tested while changing the number of states. After this tuning, and through manual inspection, we determined that 3 states were most in line with biological expectations. Each HMM state was assigned to a “biological group” (i.e. epithelial, hybrid EMT, mesenchymal) by exploring the expression levels of known epithelial and mesenchymal markers in each HMM state. The selection of the coefficients was performed with the R package *glmnet*. The identification of the EMT states was done using the *depmixS4* R package.“

4. There is no way to understand the robustness of this finding.

We have already presented the results of simulations demonstrating that adding increasing amounts of gene expression noise to the data does not affect the EMT macro-state assignment significantly unless the values exceed 500 FPKM, which is an unrealistically large amount of noise (Supplementary Figure S2a). Furthermore, we had used the data from MetMap, which measured the metastatic potential of 500 cancer cell lines, as validation for our EMT scoring methodology and the HMM macro-state identification, showing that cell lines with hEMT phenotypes have weak metastatic potential whereas those with

mesenchymal phenotypes have a strong metastatic potential (Figure 1g-h). We acknowledge that the results originally presented in Figure 1 may not have been in the clearest form possible, so we have now altered the labels and added annotations to the corresponding panels in order to make the representation clearer, see Figures 1g-h.

In addition to these previous results and in light of the previous questions from the Reviewer, the Reviewer may also be asking about the robustness of the HMM macro-state assignment when a consensus reference is used compared to using tissue-specific references. We have now generated a plot that shows the agreement and differences in EMT state assignment when using the consensus versus tissue-specific references (see Supplementary Figure S2c). The majority of tissue-specific references show an overwhelming agreement with the consensus EMT assignments. Where discrepancies are observed, these are usually involving a small state jump e.g. from EPI to hEMT or from hEMT to MES. A disagreement where one approach classifies a sample as EPI and another as MES is very rarely observed. Some discrepancies are expected given the continuous nature of this phenotype, but it is important to note that none of these involve large changes in phenotype, suggesting that the state assignments are fairly robustly determined.

We have included a discussion of this in the Results lines 173-180: “Furthermore, the EMT macro-state assignment employing the consensus template was similar to that obtained when using most of the tissue-specific templates, with expected small disagreements in phenotype (i.e. from EPI to hEMT, or hEMT to MES) occasionally observed while large disagreements (EPI to MES) were rare (Supplementary Figure S2c). The discrepancies were mostly observed against the MCF10 reference, which may be explained by the fact that these cells generally show a rather intermediate phenotype between epithelial and mesenchymal at baseline²², and hence this template may not capture the entire EMT spectrum. This analysis confirmed that the uncovered macro-states are relatively stable.”

5. It seems that EPI and hEMT states interconvert at roughly the same rates — does that make sense to the authors — if yes, why?

This is an intriguing observation which we should have highlighted in the manuscript. One would possibly expect the EPI to hEMT state conversion probability to be higher, but there is evidence in the literature that cells do revert to an epithelial-like phenotype upon removal of specific stimuli like TGF β . Thus, these conversion estimates could simply reflect the dynamic cytokine and chemokine environment across more than 7,000 tumours that have been profiled, although this requires further validation. We have expanded the Results section to comment on this in lines 186-192:

“Intriguingly, the EPI and hEMT states appear to convert at roughly the same rates, which could possibly reflect the dynamic effects of specific cytokines or chemokines either starting or ceasing to be expressed in the tumour environment. For instance, Bidarra et al²⁵ have shown that removal of TGF β 1 from the medium of a 3D *in vitro* mesenchymal cell culture leads to a reversion to an epithelial-like state. Whether stimuli that drive EMT switches are as likely to disappear as they are to appear during the course of cancer progression is difficult to assess here and should be validated in other model systems.”

6. It also seems that a quarter of the MES cells can convert to EPI cells — which seems high and left unjustified.

Indeed, the MES to EPI conversion rate may seem surprisingly high at first sight. However, the opposite process of mesenchymal to epithelial transition (MET) is well documented in the literature. Indeed, we also see evidence for EPI states in the MET500 cohort, which is only comprised of metastatic samples (Figure 1f). We have now included additional sentences in the Results to make this point more clear and to further comment on the possible reasons why this may happen (lines 197-205):

“Interestingly, we also observed possible cases of a reversion to an epithelial state in metastatic samples, which is to be expected to happen when colonizing a new environmental niche via the mesenchymal to epithelial transition programme (MET), as amply described in the literature²⁶. This could

also explain the relatively high conversion rate (26%) from MES to EPI predicted by our HMM model (Figure 1e).”

7. I would expect that these transition probabilities could be a function of the mutational background (a premise of the paper) but that is not considered in the HMM analysis. Perhaps this is the reason there was not mutagen event significantly associated with EMT transformation (lines 304-306)?

The idea of EMT transitions occurring as a function of the mutational background is a plausible one. We do not see any evidence for this when analysing the mutational signatures of every sample. However, certain mutational processes are historical, meaning that they acted sometime in the past and left a genomic scar which still persists even after the process stops. On the other hand, other mutational processes are continuous and persist in the tumour at the moment when it is sampled. As we cannot easily distinguish between the two cases, this may impede the discovery of any associations with EMT switches. Nevertheless, we find multiple associations between individual genomic events and EMT states, suggesting that certain oncogenic events are more likely to be required for the EMT transition than an entire mutational process. The latter also tend to be highly tissue-specific, meaning that if the activation of a mutational process were a requirement for EMT transition then some tissue would never experience EMT transformation, which is unlikely. On the other hand, multiple oncogenic events can have similar effects and this could more plausibly explain the ubiquitous nature of EMT and also why we see so many associations between EMT states and mutations or copy number changes in the genome. While we would like to be able to model these genomic changes together with the gene expression profiles within the HMM model, this is unlikely to be feasible due to the size of the resulting model (too many and too heterogeneous genomic changes) and the possible bias towards stable genomic states rather than expression states.

We have briefly highlighted the limitations in our understanding of the frequency and context of these switches in the Discussion (lines 715-717):

“The true number of EMT intermediate states is just beginning to be explored, and the frequency and context of such switches needs to be better understood. However, the noisy bulk sequencing data are limiting our ability to capture them, highlighting the need to complement these studies with spatially-resolved and single cell data. “

8. The authors claim that the EMT states are not influenced by the microenvironment: “The three EMT macro-states we have described within TCGA cancers displayed no significant difference in tumour purity, confirming that non-tumour cell content did not play a significant part in assigning these states.” This is a surprising result, particularly for early stage disease as others have demonstrated that myofibroblasts can alter the EMT state of cancer cells. This is a point that the authors acknowledge as ambiguous.

We apologise for the ambiguous formulation of this sentence, which has led to misinterpretation. We did not want to claim that the EMT states are not influenced by the microenvironment. Indeed, the correlations we find in bulk, spatial and single cell analysis suggest the contrary is likely true, and there is a vast body of literature demonstrating exactly this, some of which we have cited in the manuscript. We simply wanted to state that the tumour purity correction has had the desired effect and that the expression profiles have been recalibrated such that the EMT state assignment is not influenced by any non-tumour cells (given that stromal signals can be similar to hEMT signals and this can confound the assessment of EMT within the cancer cells themselves). We have now rephrased this to convey the message more clearly and moved it earlier in the context of the HMM state determination to avoid any confusion with microenvironment associations (lines 151-155):

“[...] uncovered three macro-states: epithelial (EPI), hybrid EMT (hEMT) and mesenchymal (MES) (Figure 1e-f, Supplementary Figure S1d). These states were robust to varying levels of gene expression noise (Supplementary Figure S2a), and were not influenced by sample purity, as expected due to the

correction step before pseudotime reconstruction (Supplementary Figure S2b).”

9. When the authors look at spatial transcriptomic data they observe significant EMT spatial heterogeneity, yet the rest of their paper rests on the idea that a bulk tumor specimen can have an EMT state. They do not address this contradiction, and instead from the spatial transcriptomics analysis they claim the importance of local EMT effects in tissues. When analyzing the single cell RNAseq data, they acknowledge the role of the microenvironment on the EMT of the cancer cells needs to be better understood.

Whether it is valid that a bulk tumour specimen should have an EMT state is an important point and we acknowledge we should have addressed it directly. We have done so now in the Discussion as follows (lines 742-752):

“Our spatial transcriptomics analysis demonstrates a heterogeneous EMT landscape, delineating clear spatial effects of the continuum of EMT transformation within the tissue. Thus, we gain a new appreciation of the diversity of EMT states within a single tumour, which needs to be accounted for in future studies. However, this discovery does not detract from our analyses in bulk tumours: while considering a single sample as having a unique state was clearly a simplification, the signal captured from the tumour still reflects the average state of the cells, likely captures well the extremes of the EMT distribution when scanned across thousands of TCGA samples and is the only setting that makes genomic associations possible currently. In the future, it will be important to verify these findings in matched DNA/RNA datasets from spatially or single cell sequenced samples.”

We have also removed any strong claims about local microenvironment effects (including in the abstract and title, see summary of changes), as indeed these are not clearly determined and new methods would need to be developed to demonstrate local effects. We highlight this and other limitation in lines 761-773, but also opportunities for future research:

“Nevertheless, this analysis is limited by the small sample size and our ability to capture a broad spectrum along the EMT transformation as the data are only sourced from early stage cancers. Larger spatial datasets and a finer grained resolution of EMT states, especially hybrid ones, will be required in order to understand the more complex relationships established at intermediate EMT stages, which are less clear than for the fully mesenchymal states. The combined data from spatial and single cell sequencing seem to indicate that hEMT cells are even more successful in avoiding immune detection than MES cells, as suggested by previous studies^{64,65}, but this hypothesis needs further investigation. This will also require new methods to identify localised, context-specific effects within the tissue which may not generalise throughout the tumour. Despite these limitations, our analyses do serve as a proof of concept for the ability to survey EMT spatially and lay out a framework for future studies in this space. These should ideally integrate spatial and single cell transcriptomics for a better comprehension of the complex interplay between EMT and the tumour microenvironment.”

10. It is unclear if the analysis of analysis of the publicly available siRNA screen data statistically validates any new insights derived from this study. The authors claim that 31 of 61 targets results in morphological changes that can be linked to EMT. What are the 61 targets? Is enrichment of 31 of 61 targets significant via a hypergeometric analysis, sampling from the Koedoot et al siRNA screening data?

We apologise for the omission to make the 61 targets available. We have now included a new table (Supplementary Table S5) summarising these targets which have been derived using lasso modelling as potential genomic events impacting the transition between EPI, hEMT or MES states. Only state-specific gene candidates (i.e. those which came up in a single model out of the three) were considered for these analyses. We have performed the hypergeometric test as suggested by the Reviewer and we find a significant enrichment of these targets ($p=3.33e-16$). We have updated the text with the new results:

Results lines 364-366: “There was a significant enrichment of our candidate genes among the siRNA hits from Koedoot et al⁴ (hypergeometric test $p=3.33e-16$). Specifically, we found that knocking down 31 of the 61 targets (Supplementary Table S5) resulted in significant changes in the surface area, perimeter

and elongation/roundness of the cells in Hs578T and MDA-MBA-231 breast cancer cell lines, suggesting either an impairment or an enhancement of migratory properties (Figure 4).”

Methods lines 1062-1066: “The hypergeometric test performed to assess enrichment of the 31 out of 61 targets in the screen was performed using the *dhyper* function in R by calculating the probability of getting 31 or more targets among the 217 genes which showed a phenotype in the screen out of 3906 tested in total, given that we started with a total of 61 candidate genes derived from the various genomic models.”

REVIEWERS' COMMENTS

Reviewer #2 (Remarks to the Author):

The authors have acknowledged the validity of the comments from both reviewers and have changed the text accordingly in many instances. The problem I continue to face is that there are no productive explanations for the inconsistencies/contradictions detected by both reviewers, as the authors just provide ad hoc explanations. Indeed it is true that different analyses (for instance bulk vs single-cell) are very different, but the aim is to reach conclusions that hold true regardless of the method. Otherwise, what would the reader believe? I must admit that now I am more concerned than before

Point-by-point response to Reviewers' comments

Reviewer #2:

The authors have acknowledged the validity of the comments from both reviewers and have changed the text accordingly in many instances. The problem I continue to face is that there are no productive explanations for the inconsistencies/contradictions detected by both reviewers, as the authors just provide ad hoc explanations. Indeed it is true that different analyses (for instance bulk vs single-cell) are very different, but the aim is to reach conclusions that hold true regardless of the method. Otherwise, what would the reader believe? I must admit that now I am more concerned than before

We have thoroughly acknowledged that not all results are consistent across platforms and have tried our best to provide explanations for why this may be the case. The reality is that these are still early days for understanding relations between tumour cell states and their environment in single cell data, and very early days for exploring such relations in spatial transcriptomics. Therefore, we are limited both by the methods available to study such complex multidimensional landscapes as well as by the data availability. In fact, the latter is likely to be the main cause for any discrepancies in the results. We only have a few spatial transcriptomics samples from patients with breast cancer, and a few single cell patient samples from breast, lung, colon and ovarian cancers. Although the number of profiled spots and cells is large, there is limited sampling of the diversity of cancer tissues and subtypes. Inter- and intratumour heterogeneity are well known factors that could affect what we are seeing in such a limited number of samples, and a plausible reason for why not all EMT-TME relations look the same. As we break the EMT transition further down into multiple states, it is not surprising that we might observe more context-dependent effects rather than a unique, generalisable pattern. Furthermore, moving from bulk to better resolved datasets is bound to uncover more diversity, as was already shown very prevalently for cancer mutations, which reveal diverse clonal structure when spatially sampled rather than from a single sample. Transcriptomics data is undoubtedly similar in this respect. Another limitation which we have not stressed enough in this context is the fact that we have focused on a single hybrid state, but we acknowledge this is likely encompassing multiple such states which may or may not all be present in one tumour or another. This is another factor which could confound our analyses, although we have chosen the most conservative approach for the sake of simplicity and the awareness that the number of states may be highly cancer type-dependent. We agree with the Reviewer that further dissecting and resolving all these complexities is absolutely crucial in future studies, but this is beyond the scope of our manuscript. We have now more thoroughly acknowledged these additional limitations mentioned here in the Discussion (lines 656-662):

“Our study was limited by the availability of spatial and single cell datasets, which are only sourced from a few patients with breast, lung, ovarian and colon cancer, and hence it is not surprising that the TME associations are not always generalisable in these datasets. Furthermore, the hybrid state was treated as a single state, but we acknowledge that more than one such state can occur with different properties, which may confound the results. Intra- and intertumour heterogeneity are likely to create complex EMT-TME landscapes that are tissue and patient-specific, and resolving these is beyond the scope of our study.”

Finally, we would also like to point out that the main scope of our analysis was to provide a proof of concept for the feasibility of identifying EMT states and studying them in spatially resolved and single cell datasets and we hope future studies can build upon our work, which we also point out in the discussion:

“Despite these limitations, our analyses do serve as a proof of concept for the ability to survey EMT spatially and lay out a framework for future studies in this space. These should ideally integrate spatial and single cell transcriptomics for a better comprehension of the complex interplay between EMT and the tumour microenvironment.”